# FASTER ADAPTIVE MOMENTUM-BASED FEDERATED METHODS FOR DISTRIBUTED COMPOSITION OPTIMIZATION

## ABSTRACT

Federated learning is a popular distributed learning paradigm in machine learning. Meanwhile, composition optimization is an effective hierarchical learning model, which appears in many machine learning applications such as meta learning and robust learning. More recently, although a few federated composition optimization algorithms have been proposed, they still suffer from high sample and communication complexities. In the paper, thus, we propose a class of faster adaptive federated compositional optimization algorithms (i.e., MFCGD and AdaMFCGD) to solve the nonconvex distributed composition problems, which builds on the momentum-based variance reduced and local-SGD techniques. In particular, our adaptive algorithm (i.e., AdaMFCGD) uses a unified adaptive matrix to flexibly incorporate various adaptive learning rates. Moreover, we provide a solid theoretical analysis for our algorithms under non-i.i.d. setting, and prove our algorithms obtain a lower sample and communication complexities simultaneously than the existing federated composition optimization algorithms. Specifically, our algorithms obtain lower sample complexity of $\tilde{O}(\epsilon^{-3})$ with lower communication complexity of $\tilde{O}(\epsilon^{-2})$ in finding an $\epsilon$-stationary solution. We conduct numerical experiments on robust federated learning and distributed meta learning tasks to demonstrate the efficiency of our algorithms.

## 1 INTRODUCTION

Composition optimization is an effective hierarchical model in machine learning, which is widely used to many applications such as reinforcement learning Wang et al. (2017b); Huo et al. (2018), meta learning Wang et al. (2021), robust federated learning Huang et al. (2021a) and deep AUC maximization Yuan et al. (2022). In the paper, we study the following distributed composition optimization problem:

$$\min_{x \in \mathbb{R}^d} \ \frac{1}{M} \sum_{m=1}^{M} \mathbb{E}_{\xi^m} \left[ f^m \Big( \mathbb{E}_{\zeta^m} \big[ g^m(x; \zeta^m) \big]; \xi^m \Big) \right], \tag{1}$$

where $F(x) := \frac{1}{M} \sum_{m=1}^{M} f^m(x)$ and $f^m(x) = \mathbb{E}_{\xi^m} \left[ f^m \Big( \mathbb{E}_{\zeta^m} \big[ g^m(x; \zeta^m) \big]; \xi^m \Big) \right]$. Here $y^m = g^m(x) = \mathbb{E}_{\zeta^m \sim \mathcal{S}^m} \big[ g^m(x; \zeta^m) \big]$ and $f^m(y^m) = \mathbb{E}_{\xi^m \sim \mathcal{D}^m} \big[ f^m(y^m; \xi^m) \big]$ for any $m \in [M]$ denote the inner and outer objective functions respectively in $m$-th client. Here $\xi^m$ and $\zeta^m$ for any $m \in [M]$ are independent random variables follow unknown distributions $\mathcal{D}^m$ and $\mathcal{S}^m$ respectively. For any $m, j \in [M]$ possibly $\mathcal{D}^m \neq \mathcal{D}^j$, $\mathcal{S}^m \neq \mathcal{S}^j$ and $\mathcal{D}^m \neq \mathcal{S}^j$. Applications of Problem (1) involve many machine learning problems with a compositional structure, which include model-agnostic meta learning Tutunov et al. (2020); Chen et al. (2020b); Wang et al. (2021), reinforcement learning Wang et al. (2017b); Huo et al. (2018) and sparse additive models Wang et al. (2017a). In the following, we give two specific applications that can be formulated as the distributed composition Problem (1).

**1). Task-Distributed Meta Learning.** Meta learning is to learn some properties in the optimal model to improve model performances with more experiences, i.e., learning to learn Andrychowicz et al. (2016). Model-Agnostic Meta Learning (MAML) Finn et al. (2017) is a class of popular meta

Table 1: **Sample** and **Communication** complexities comparison of the representative **federated compositional optimization** algorithms in finding an $\epsilon$-stationary point of the distributed composition optimization problem (1), i.e., $\mathbb{E}\|\nabla F(x)\| \leq \epsilon$ or its equivalent variants. **ALR** denotes adaptive learning rate.

| Algorithm | Reference | Sample Complexity | Communication Complexity | ALR |
|-----------|-----------|-------------------|--------------------------|-----|
| ComFedL | Huang et al. (2021a) | $O(\epsilon^{-8})$ | $O(\epsilon^{-4})$ | |
| Local-MOML | Wang et al. (2021) | $O(\epsilon^{-5})$ | $O(\epsilon^{-3})$ | |
| FEDNEST | Tarzanagh et al. (2022) | $\tilde{O}(\epsilon^{-4})$ | $\tilde{O}(\epsilon^{-4})$ | |
| Local-SCGDM | Gao et al. (2022) | $O(\epsilon^{-4})$ | $O(\epsilon^{-3})$ | |
| MFCGD | Ours | $\tilde{O}(\epsilon^{-3})$ | $\tilde{O}(\epsilon^{-2})$ | |
| AdaMFCGD | Ours | $\tilde{O}(\epsilon^{-3})$ | $\tilde{O}(\epsilon^{-2})$ | ✓ |

learning methods, which is to find a common initialization that can adapt to a desired model for a set of new tasks after taking several gradient descent steps. In the paper, we consider a class of task-distributed MAMLs, where a set of tasks $\{\mathcal{T}_m\}_{m=1}^M$ are drawn from a certain task distribution and each task is assigned in each client. Specifically, we solve the following task-distributed MAML problem:

$$\min_{x \in \mathbb{R}^d} \frac{1}{M} \sum_{m=1}^M f^m\big(x - \eta \nabla f^m(x)\big), \tag{2}$$

where $f^m(x) = \mathbb{E}_{\xi^m \sim \mathcal{D}^m}[f(x; \xi^m)]$, and random variable $\xi^m$ follows the unknown distribution $\mathcal{D}^m$, and $\eta > 0$ is a learning rate. Let $f^m(y^m) = f^m(g^m(x))$ and $y^m = g^m(x) = x - \eta \nabla f^m(x)$, the above problem (2) is a special case of the above composition problem (1).

**2). Distributionally Robust Federated Learning.** Federated learning (FL) McMahan et al. (2017); Kairouz et al. (2019); Li et al. (2021a) is a distributed and privacy preserving machine learning method to learn a global model collaboratively from decentralized data distributed over a network of devices. To tackle the data heterogeneity from different devices, some robust FL algorithms Mohri et al. (2019); Reisizadeh et al. (2020); Deng et al. (2020b) have been studied. In the paper, as in Huang et al. (2021a), we consider solving the following distributed composition problem to reach distributionally robust FL, defined as

$$\min_{x \in \mathbb{R}^d} \frac{1}{M} \sum_{m=1}^M f\Big(\mathbb{E}\big[g^m(x; \xi^m)\big]\Big), \tag{3}$$

where $g^m(x) = \mathbb{E}\big[g^m(x; \xi^m)\big]$ denotes the loss function in the $m$-th client, and $f(\cdot)$ is a monotonically increasing function. Clearly, the problem (3) is a special case of the above problem (1).

Although recently many compositional gradient algorithms have been proposed to solve the composition problems, few distributed algorithms focus on solving the distributed composition optimization problems. More recently, Huang et al. (2021a); Wang et al. (2021); Gao et al. (2022); Tarzanagh et al. (2022) proposed some federated compositional gradient algorithms for the distributed stochastic composition problems. However, few adaptive algorithm focuses on the composition optimization problems under the distributed setting. Meanwhile, these existing federated composition optimization methods suffer from large sample and communication complexities (Please see Table 1). Then there exists a natural question:

> **Could we develop faster and adaptive federated learning methods to solve the distributed composition optimization problem (1) ?**

In the paper, we provide an affirmative answer to the above question and propose a class of faster momentum-based federated compositional gradient descent algorithms (i.e., MFCGD and AdaMFCGD) to solve Problem (1), which build on the local Stochastic Gradient Descent (SGD) and momentum-based variance reduced techniques to obtain lower sample and communication complexities simultaneously. Our main contributions are as follows:

(1) We propose a class of faster adaptive momentum-based federated compositional gradient descent algorithms (i.e., MFCGD and AdaMFCGD) to solve the nonconvex distributed

composition problems, which build on the momentum-based variance reduced and local-SGD techniques. In particular, our adaptive algorithm (i.e., AdaMFCGD) uses a unified adaptive matrix to flexibly incorporate various adaptive learning rates.

(2) We provide a solid convergence analysis framework for our algorithms under non-i.i.d. setting, and prove that our algorithms obtain simultaneously lower sample complexity of $\tilde{O}(\epsilon^{-3})$ and lower communication complexity of $\tilde{O}(\epsilon^{-2})$ than the existing federated composition methods for finding an $\epsilon$-stationary solution (Please see Table 1).

(3) Experimental results demonstrate efficiency of our algorithms on the task-distributed meta learning and robust federated learning tasks.

## 2 RELATED WORKS

In this section, we overview some representative composition optimization, federated optimization and adaptive optimization methods, respectively.

### 2.1 COMPOSITION OPTIMIZATION

Composition optimization has been widely applied to many applications such as reinforcement learning Wang et al. (2017b), model-agnostic meta Learning Tutunov et al. (2020) and risk management Huo et al. (2018). Recently, many compositional gradient-based methods have recently been proposed to solve these composition optimization problems. For example, stochastic compositional gradient methods Wang et al. (2017a;b); Ghadimi et al. (2020) have been proposed to solve these problems. Subsequently, some variance-reduced compositional algorithms Huo et al. (2018); Lin et al. (2018); Zhang & Xiao (2019) have been proposed for composition optimization. Tutunov et al. (2020); Chen et al. (2020b) presented a class of momentum-based compositional gradient methods for stochastic composition optimization. More recently, Jiang et al. (2022) proposed a class of efficient momentum-based variance reduced methods for non-convex stochastic composition optimization. Huang & Gao (2022) studied the stochastic composition optimization on Riemannian manifolds.

For the distributed setting, Huang et al. (2021a) firstly studied federated learning algorithm for the general distributed composition optimization. Meanwhile, Wang et al. (2021) studied personalized federated learning algorithm based on the composition optimization. Subsequently, Gao et al. (2022); Tarzanagh et al. (2022) proposed some accelerated federated learning algorithms for the distributed composition optimization.

### 2.2 FEDERATED LEARNING

Federated Learning (FL) McMahan et al. (2017); Li et al. (2021a); Zhang et al. (2022) is a promising distributed machine learning framework for collaboratively training the global model without sharing the local data to obtain the privacy-preserving learning solutions, and is widely used in many applications such as healthcare informatics Xu et al. (2021) and automatic diagnosis of COVID-19 Yang et al. (2021). McMahan et al. (2017) first studied FL and proposed the FedAvg algorithm for FL based on local-SGD algorithms Stich (2019), where each client conducts multiple steps of SGD with its local data and then sends the learned model to the server for averaging. Subsequently, Li et al. (2019); Karimireddy et al. (2019); Deng & Mahdavi (2021) have studied the convergence properties of the local-SGD and FedAvg algorithms or their variations. To accelerate the vanilla local-SGD and FedAvg algorithms, various accelerated FL algorithms Yuan & Ma (2020); Karimireddy et al. (2020); Khanduri et al. (2021); Chen et al. (2020a) have been developed and studied. For example, Karimireddy et al. (2020) proposed a stochastic controlled averaging algorithm for FL by adopting the variance-reduced technique of SARAH Nguyen et al. (2017)/SPIDER Fang et al. (2018). Subsequently, Khanduri et al. (2021) proposed a faster federated algorithm based on momentum-based variance reduced technique of STORM Cutkosky & Orabona (2019) and ProxHSGD Tran-Dinh et al. (2022), which obtains lower sample and communication complexities simultaneously.

To solve the data heterogeneity in FL, Mohri et al. (2019); Deng et al. (2020b) proposed some effective robust FL algorithms by learning the worst-case loss based on the minimax optimization problems. To further incorporate personalization in FL, some personalized federated learning models Fallah et al. (2020); Deng et al. (2020a); Li et al. (2021b) have been developed and studied. For example, Li et al. (2021b) proposed an effective and efficient personalized FL algorithm (i.e., Ditto) by learning a regularized local model for each client.

### 2.3 Adaptive Optimization Methods

Adaptive optimization methods Duchi et al. (2011); Kingma & Ba (2014) are a class of efficient optimization methods due to using adaptive learning rates in machine learning, and they have been widely studied in machine learning community. For example, AdaGrad Duchi et al. (2011) is the first adaptive gradient method. Adam Kingma & Ba (2014) is a popular variation of AdaGrad algorithm based on the momentum technique, which is the default optimization algorithm for training large-scale machine learning models. Meanwhile, some variants of Adam algorithm Reddi et al. (2019); Chen et al. (2019) have been proposed to obtain a convergence guarantee under the nonconvex setting. To further improve the performance of Adam algorithm, recently some new its variants such as AdamW Loshchilov & Hutter (2018) have been developed. More recently, some accelerated adaptive gradient methods Cutkosky & Orabona (2019); Huang et al. (2021b) have been proposed based on the momentum-based variance reduced techniques. In parallel, some adaptive gradient methods Reddi et al. (2020); Chen et al. (2020c) are proposed for distributed optimization. For example, Reddi et al. (2020) proposed a class of adaptive federated algorithms for FL by using adaptive learning rates at the server side.

### Notations

Let $[M]$ denote the set $\{1, 2, \cdots, M\}$. $\| \cdot \|$ denotes the $\ell_2$ norm for vectors and Frobenius norm for matrices. $\langle x, y \rangle$ denotes the inner product of two vectors $x$ and $y$. For vectors $x$ and $y$, $x^r$ $(r > 0)$ denotes the element-wise power operation, $x/y$ denotes the element-wise division and $\max(x, y)$ denotes the element-wise maximum. $I_d$ denotes a $d$-dimensional identity matrix. $A \succ 0$ denotes that $A$ is a positive definite matrix. $a_t = O(b_t)$ denotes that $a_t \leq c b_t$ for some constant $c > 0$. The notation $\tilde{O}(\cdot)$ hides logarithmic terms. $\Pi_C[x] = \arg\min_{||w|| \leq C} ||x - w||^2$ denote a projection onto the ball with radius $C > 0$.

## 3 Federated Compositional Gradient Descent Algorithms

In this section, we propose a class of faster momentum-based federated compositional gradient descent algorithms (i.e., MFCGD and AdaMFCGD) to solve the problem (1), which builds on the local-SGD and momentum-based variance reduced techniques. Specifically, the local-SGD technique reduce the communication complexity and the momentum-based variance reduced technique reduce the sample complexity without relying on large batches. Meanwhile, our AdaMFCGD algorithm uses the unified adaptive matrix to flexibly incorporate various adaptive learning rates in updating variables. Specifically, Algorithm 1 provides a procedure framework of our MFCGD and AdaMFCGD algorithms.

In Algorithm 1, when $\mod(t, q) = 0$ (i.e., **synchronization** step), the server receives the local variables $\{x_t^m\}_{m=1}^M$ and local gradients $\{w_t^m\}_{m=1}^M$ from the clients, and then averages them to obtain the averaged variables $\{\bar{x}_t\}$ and averaged gradients $\{\bar{w}_t\}$. Based on these averaged gradients $\{\bar{w}_t\}$, we can generate some adaptive matrices $\{A_t\}_{t \geq 1}$ (i.e., adaptive learning rates). **Note that** for our non-adaptive MFCGD algorithm, we only set $\bar{A}_t = I_d$ for all $t \geq 1$ in Algorithm 1. Besides one example given at the line 6 of Algorithm 1, we can also generate many other adaptive matrices. For example, we can generate adaptive matrix $A_t$ as the norm-type of Adam, defined as

$$a_t = \vartheta_t a_{t-1} + (1 - \vartheta_t)\|\bar{w}_t\|, \quad A_t = \text{diag}(a_t + \rho), \tag{4}$$

where $0 < \vartheta_t \leq 1$. Note that we can directly choose $\alpha_t$, $\beta_t$ or $\varrho_t$ instead of $\vartheta_t$ to reduce the number of tuning parameters in our algorithm. Next, based on these adaptive matrices, we can update the variable $x$ in the server, then sent it to each client.

When $\mod(t, q) \neq 0$ (i.e., **asynchronization** step), the clients receive the updated variables $\{\bar{x}_{t+1}\}$ and the generated adaptive matrices $\{A_t\}$ from the server. Then the clients use the momentum-based variance reduced technique of STORM Cutkosky & Orabona (2019) and ProxHSGD Tran-Dinh et al. (2022) to update the stochastic gradients based on local data: for $m \in [M]$

$$h_{t+1}^m = g^m(x_{t+1}^m; \zeta_{t+1}^m) + (1 - \alpha_{t+1})\big(h_t^m - g^m(x_t^m; \zeta_{t+1}^m)\big)$$

$$u_{t+1}^m = \Pi_{C_g}\Big[\nabla g^m(x_{t+1}^m; \zeta_{t+1}^m) + (1 - \beta_{t+1})\big(u_t^m - \nabla g^m(x_t^m; \zeta_{t+1}^m)\big)\Big]$$

$$v_{t+1}^m = \Pi_{C_f}\Big[\nabla f^m(h_{t+1}^m; \xi_{t+1}^m) + (1 - \varrho_{t+1})\big(v_t^m - \nabla f(h_t^m; \xi_{t+1}^m)\big)\Big],$$

---

**Algorithm 1 MFCGD** and **AdaMFCGD** Algorithms

---

1: **Input:** $T, q$, tuning parameters $\{\gamma, \eta_t, \alpha_t, \beta_t, \varrho_t\}$ and initial input $x_1 \in \mathbb{R}^d$;

2: **initialize:** Set $x_1^m = x_1$ for $m \in [M]$, and draw $2q$ independent samples $\{\xi_{1,j}^m\}_{j=1}^q$ and $\{\zeta_{1,j}^m\}_{j=1}^q$, and then compute $h_1^m = \frac{1}{q}\sum_{j=1}^q g^m(x_1^m; \zeta_{1,j}^m)$, $u_1^m = \frac{1}{q}\sum_{j=1}^q \nabla g^m(x_1^m; \zeta_{1,j}^m)$ and $v_1^m = \frac{1}{q}\sum_{j=1}^q \nabla f(h_1^m; \xi_{1,j}^m)$ for all $m \in [M]$; Generate adaptive matrix $A_1 \in \mathbb{R}^{d \times d}$.

3: **for** $t = 1$ **to** $T$ **do**

4:   **if** $\mod(t, q) = 0$ **then**

5:       $\bar{w}_t = \frac{1}{M}\sum_{m=1}^M w_t^m$ and $\bar{x}_t = \frac{1}{M}\sum_{m=1}^M x_t^m$;

6:       Generate the adaptive matrix $A_t \in \mathbb{R}^{d \times d}$;

      One example of $A_t$ by using update rule ($a_0 = 0$, $0 < \vartheta_t < 1$, $\rho > 0$.)

      Compute $a_t = \vartheta_t a_{t-1} + (1 - \vartheta_t)\bar{w}_t^2$, $A_t = \text{diag}(\sqrt{a_t} + \rho)$;

7:       $x_{t+1}^m = \bar{x}_{t+1} = \arg\min_{x \in \mathbb{R}^d}\left\{\langle x, \bar{w}_t\rangle + \frac{1}{2\eta_t\gamma}(x - \bar{x}_t)^T A_t(x - \bar{x}_t)\right\}$; (Sent them to Clients)

8:   **else**

9:     **for** each client $m \in [M]$ **(in parallel) do**

10:        $w_t^m = (u_t^m)^T v_t^m$;

11:        $x_{t+1}^m = \arg\min_{x \in \mathbb{R}^d}\left\{\langle x, w_t^m\rangle + \frac{1}{2\eta_t\gamma}(x - x_t^m)^T A_t(x - x_t^m)\right\}$;

12:        $A_{t+1} = A_t$;

13:     **end for**

14:   **end if**

15:   **for** each client $m \in [M]$ **(in parallel) do**

16:     Draw two independent samples $\xi_{t+1}^m$ and $\zeta_{t+1}^m$;

17:     $h_{t+1}^m = g^m(x_{t+1}^m; \zeta_{t+1}^m) + (1 - \alpha_{t+1})(h_t^m - g^m(x_t^m; \zeta_{t+1}^m))$;

18:     $u_{t+1}^m = \Pi_{C_g}\left[\nabla g^m(x_{t+1}^m; \zeta_{t+1}^m) + (1 - \beta_{t+1})(u_t^m - \nabla g^m(x_t^m; \zeta_{t+1}^m))\right]$;

19:     $v_{t+1}^m = \Pi_{C_f}\left[\nabla f^m(h_{t+1}^m; \xi_{t+1}^m) + (1 - \varrho_{t+1})(v_t^m - \nabla f(h_t^m; \xi_{t+1}^m))\right]$;

20:     $w_{t+1}^m = (u_{t+1}^m)^T v_{t+1}^m$;

21:   **end for**

22: **end for**

23: **Output:** Chosen uniformly random from $\{\bar{x}_t\}_{t=1}^T$, where $\bar{x}_t = \frac{1}{M}\sum_{m=1}^M x_t^m$.

---

where $\alpha_{t+1} \in (0, 1)$, $\beta_{t+1} \in (0, 1)$ and $\varrho_{t+1} \in (0, 1)$. Here the projection functions $\Pi_{C_g}[\cdot]$ and $\Pi_{C_f}[\cdot]$ ensure that the estimated stochastic gradients $u_{t+1}^m$ and $v_{t+1}^m$ are bounded, i.e., $\|u_{t+1}^m\| \leq C_g$ and $\|v_{t+1}^m\| \leq C_f$ for any $t \geq 1$. Based on the estimated stochastic gradients and adaptive matrices, the clients update the variables $\{x_t^m\}_{m=1}^M$, defined as

$$x_{t+1}^m = x_t^m - \gamma\eta_t A_t^{-1} w_t^m = \arg\min_{x \in \mathbb{R}^d}\left\{\langle x, w_t^m\rangle + \frac{1}{2\eta_t\gamma}(x - x_t^m)^T A_t(x - x_t^m)\right\}, \quad (5)$$

where $\gamma > 0$ and $\eta_t > 0$. In our algorithms, all clients use the same adaptive matrix generated from the server as in Chen et al. (2020c). **Note that** the existing adaptive FL algorithms such as local-AMSGrad Chen et al. (2020c) only builds on some specific adaptive learning rates such as AMSGrad Reddi et al. (2019). However, our algorithms can use the unified adaptive matrix to flexibly incorporate various adaptive learning rates.

## 4 CONVERGENCE ANALYSIS

In this section, we study the convergence properties of our MFCGD and AdaMFCGD algorithms under some mild assumptions. All related proofs are provided in the Appendix A. We first review some useful lemmas and assumptions.

**Assumption 1.** *(Lipschitz Gradients) For any $m \in [M]$, there exist constants $L_f$ and $L_g$ for $\nabla f^m(y; \xi^m)$, $\nabla g^m(x; \zeta^m)$ respectively satisfying*

$$\|\nabla g^m(x_1, \zeta^m) - \nabla g^m(x_2, \zeta^m)\| \leq L_g\|x_1 - x_2\|, \ \forall x_1, x_2 \in \mathbb{R}^d,$$
$$\|\nabla f^m(y_1; \xi^m) - \nabla f^m(y_2; \xi^m)\| \leq L_f\|y_1 - y_2\|, \ \forall y_1, y_2 \in \mathbb{R}^p.$$

**Assumption 2.** *(Bounded Gradients) For any $m \in [M]$, gradient $\nabla g^m(x;\zeta^m)$ and Jacobian matrix $\nabla f^m(y;\xi^m)$ have the upper bounds $C_g$ and $C_f$ respectively, i.e.,*

$$\|\nabla g^m(x;\zeta^m)\| \leq C_g, \ \|\nabla f^m(y;\xi^m)\| \leq C_f, \ \forall x \in \mathbb{R}^d, \ y \in \mathbb{R}^p.$$

**Assumption 3.** *(Bounded Variances) For any $m \in [M]$, functions $f^m(y;\xi^m)$ and $g^m(x;\zeta^m)$ and its gradients are unbiased and the bounded variances, i.e., we have $\mathbb{E}[g^m(x;\zeta^m)] = g^m(x)$, $\mathbb{E}[\nabla g^m(x;\zeta^m)] = \nabla g^m(x)$, $\mathbb{E}[\nabla f^m(y;\xi^m)] = \nabla f^m(y)$ and*

$$\mathbb{E}\|g^m(x;\zeta^m) - g^m(x)\|^2 \leq \sigma^2, \ \mathbb{E}\|\nabla g^m(x;\zeta^m) - \nabla g^m(x)\|^2 \leq \sigma^2,$$

$$\mathbb{E}\|\nabla f^m(y;\xi^m) - \nabla f^m(y)\|^2 \leq \sigma^2, \quad \forall x \in \mathbb{R}^d, \ y \in \mathbb{R}^p$$

*where $\sigma > 0$.*

**Assumption 4.** *$F(x)$ has a lower bound, i.e., $F^* = \inf_{x \in \mathbb{R}^d} F(x)$.*

**Assumption 5.** *In our algorithms, the adaptive matrix $A_t$ for all $t \geq 1$ satisfies $A_t \succeq \rho I_d$, where $\rho > 0$ is an appropriate positive number.*

**Assumption 6.** *For any $m, j \in [M]$, $x \in \mathbb{R}^d$ and $y \in \mathbb{R}^p$, we have $\|\nabla f^m(y) - \nabla f^j(y)\| \leq \delta_f$, $\|\nabla g^m(x) - \nabla g^j(x)\| \leq \delta_g$ and $\|g^m(x) - g^j(x)\| \leq \delta_g$, where $\delta_f > 0$ and $\delta_g > 0$ are constants.*

Assumptions 1 ensures the smoothness of functions $f^m(y;\xi^m)$, $g^m(x;\zeta^m)$ for any $m \in [M]$, Assumption 2 ensures the bounded gradients (or Jacobian matrix) of functions $f^m(y;\xi^m)$ and $g^m(x;\zeta^m)$ for any $m \in [M]$. Assumption 3 ensures the bounded variances of stochastic gradient or value of functions $f^m(y;\xi^m)$ and $g^m(x;\zeta^m)$ for any $m \in [M]$. Assumption 4 guarantees the feasibility of the problem (1). Assumptions 1-4 have been commonly used in the convergence analysis of the stochastic composition algorithms Wang et al. (2017a;b). Assumption 5 has been commonly used in the existing adaptive methods Huang et al. (2021b). Assumption 6 is the standard condition constrained the data heterogeneity in non-i.i.d FL setting Li et al. (2019). In fact, we can obtain the part results of Assumption 6 based on Assumptions 1-2. For example, we have $\|\nabla f^m(y) - \nabla f^j(y)\| \leq 2\sigma + 2C_f$, where the last inequality holds by Assumptions 1-2. Similarly, we have $\|\nabla g^m(y) - \nabla g^j(y)\| \leq 2\sigma + 2C_g$ based on Assumptions 1-2.

### 4.1 Convergence Properties of AdaMFCGD Algorithm

In this subsection, we provide the convergence properties of our **AdaMFCGD** algorithm.

**Theorem 1.** *Assume the sequence $\{\bar{x}_t\}_{t=1}^T$ be generated from **AdaMFCGD** algorithm. Under the above Assumptions, and let $\eta_t = \frac{k}{(n+t)^{1/3}}$ for all $t \geq 0$, $\alpha_{t+1} = c_1 \eta_t^2$, $\beta_{t+1} = c_2 \eta_t^2$, $\varrho_{t+1} = c_3 \eta_t^2$, $n \geq \max\left(2, k^3, (c_1 k)^3, (c_2 k)^3, (c_3 k)^3, \frac{(24 k \gamma q L_{fg} C_{fg})^3}{\rho^3}\right)$, $k > 0$, $c_1 \geq \frac{2}{3k^3} + B$, $c_2 \geq \frac{2}{3k^3} + 5C_f^2$, $c_1^2 + c_2^2 \leq \frac{(24)^4 q^2 \gamma^4 L_{fg}^4 C_{fg}^4}{9\rho^4}$, $c_3 \geq \frac{2}{3k^3} + 5C_g^2$, $\frac{\rho(c_1^2+c_3^2)^{1/4}}{12\sqrt{5q}L_{fg}C_{fg}} \leq \gamma \leq \min\left(\frac{3\rho q L_{fg} C_{fg}}{4(C_g^2+L_g^2+2L_f^2 C_g^2)}, \frac{n^{1/3}\rho}{2Lk}\right)$, $B \geq 20 C_g^2 L_f^2 + \frac{c_2^2 C_g^2 L_f^2}{216 q^3 \gamma^3 L_{fg}^3 C_{fg}^3} + \frac{\Theta \rho^2 (c_1^2+c_3^2)}{30 q^2 \gamma^4 C_{fg}^2 L_{fg}^2 C_g^2}$, $\Theta = \left(5C_f^2 L_g^2 + \frac{c_2^2 C_g^2 L_f^2}{864 q^3 \gamma^3 L_{fg}^3 C_{fg}^3}\right) \frac{\rho^2}{(24)^2 L_{fg}^2 C_{fg}^2} + \frac{\gamma \rho}{6q L_{fg} C_{fg}}\left(C_g^2 + L_g^2 + 2L_f^2 C_g^2\right)$ and $\Theta + \frac{BC_g^2 \rho^2}{(24)^2 L_{fg}^2 C_{fg}^2} \leq \frac{5\rho^2}{48}$, we have*

$$\frac{1}{T}\sum_{t=1}^T \mathbb{E}\|\nabla F(\bar{x}_t)\| \leq \left(\frac{\sqrt{2G}n^{1/6}}{T^{1/2}} + \frac{\sqrt{2G}}{T^{1/3}}\right)\sqrt{\frac{1}{T}\sum_{t=1}^T \mathbb{E}\|A_t\|^2}, \tag{6}$$

*where $C_{fg}^2 = \max(C_f^2, C_g^2)$, $L_{fg}^2 = L_f^2 C_g^2 + L_g^2$, $G = \frac{4(F(\bar{x}_1)-F^*)}{k\rho\gamma} + \frac{12n^{1/3}\sigma^2}{qk^2\rho^2} + 4k^2\left(\frac{\hat{\delta}^2}{4\gamma^2 L_{fg}^2} + \frac{(c_1^2+c_2^2+c_3^2)\sigma^2}{3\rho\gamma q L_{fg} C_{fg}}\right)\ln(n+T)$ and $\hat{\delta}^2 = 2c_1^2 L_f^2 \sigma^2 + c_3^2 \sigma^2 + 4c_3^2 \delta_f^2 + 4c_3^2 L_f^2 \delta_g^2 + c_2^2 \sigma^2 + 3c_2^2 \delta_g^2$.*

**Remark 1.** *Under the above Assumption 2, we have $\left\|\frac{1}{M}\sum_{m=1}^M \left(\nabla g^m(\bar{x}_t)\right)^T \nabla f^m(g^m(\bar{x}_t))\right\| \leq C_f C_g$. When the adaptive matrix $A_t$ be generated from the line 6 of Algorithm 1, we have $\sqrt{\frac{1}{T}\sum_{t=1}^T \mathbb{E}\|A_t\|^2} \leq 2(C_f^2 C_g^2 + \rho)$. Without loss of generality, let $k = O(1)$, $\rho = O(1)$, $c_1 = O(1)$, $c_2 = O(1)$, $c_3 = O(1)$ and $n = O(q^3)$, we have and $G = \tilde{O}(1)$. Let $q = T^{1/3}$ and*

$$\frac{1}{T}\sum_{t=1}^T \mathbb{E}\|\nabla F(\bar{x}_t)\| \leq \tilde{O}\left(\frac{\sqrt{q}}{\sqrt{T}} + \frac{1}{T^{1/3}}\right) = \tilde{O}\left(\frac{1}{T^{1/3}}\right) \leq \epsilon, \tag{7}$$

*then we have $T = \tilde{O}(\epsilon^{-3})$. Since our AdaMFCGD algorithm requires $2$ samples at each iteration expect for the first iteration requires $2q$ samples, it has a sample complexity of $2q + 2T = \tilde{O}(\epsilon^{-3})$. Thus, our AdaMFCGD algorithm requires $\tilde{O}(\epsilon^{-3})$ sample (or gradient) complexity and $\frac{T}{q} = T^{2/3} = \tilde{O}(\epsilon^{-2})$ communication complexity to find an $\epsilon$-stationary point of the Problem (1).*

**Remark 2.** *From Theorem 1, our AdaMFCGD algorithm simultaneously have lower sample and communication complexities than the existing federated compositional optimization algorithms (Please see Table 1). Moreover, our AdaMFCGD algorithm simultaneously have lower sample and communication complexities than the existing adaptive single-level FL algorithms such as the local-AMSGrad Chen et al. (2020c) algorithm that needs sample complexity of $O(\epsilon^{-4})$ and communication complexity of $O(\epsilon^{-3})$ for finding an $\epsilon$-stationary point of the distributed single-level optimization problem, i.e., the above problem (1) with $g^m(x) = x$ for all $m \in [M]$.*

## 4.2 CONVERGENCE PROPERTIES OF MFCGD ALGORITHM

In this subsection, we provide the convergence properties of our non-adaptive **MFCGD** algorithm, i.e., set $A_t = I_d$ for all $t \geq 1$.

**Theorem 2.** *Assume the sequence $\{\bar{x}_t\}_{t=1}^T$ be generated from **MFCGD** algorithm, i.e., $A_t = I_d$ for all $t \geq 1$ in Algorithm 1. Under the above Assumptions, and let $\eta_t = \frac{k}{(n+t)^{1/3}}$ for all $t \geq 0$, $\alpha_{t+1} = c_1\eta_t^2$, $\beta_{t+1} = c_2\eta_t^2$, $\varrho_{t+1} = c_3\eta_t^2$, $n \geq \max\left(2, k^3, (c_1 k)^3, (c_2 k)^3, (c_3 k)^3, (24k\gamma q L_{fg} C_{fg})^3\right)$, $k > 0$, $c_1 \geq \frac{2}{3k^3} + B$, $c_2 \geq \frac{2}{3k^3} + 5C_f^2$, $c_1^2 + c_2^2 \leq \frac{(24)^4 q^2 \gamma^4 L_{fg}^4 C_{fg}^4}{9}$, $c_3 \geq \frac{2}{3k^3} + 5C_g^2$, $\frac{(c_1^2+c_3^2)^{1/4}}{12\sqrt{5q}L_{fg}C_{fg}} \leq \gamma \leq \min\left(\frac{3qL_{fg}C_{fg}}{4(C_g^2+L_f^2+2L_f^2C_g^2)}, \frac{n^{1/3}}{2Lk}\right)$, $B \geq 20C_g^2 L_f^2 + \frac{c_2^2 C_g^2 L_f^2}{216q^3 \gamma^3 L_{fg}^3 C_{fg}^3} + \frac{\Theta(c_1^2+c_3^2)}{30q^2 \gamma^4 C_{fg}^2 L_{fg}^2 C_g^2}$, $\Theta = \left(5C_f^2 L_g^2 + \frac{c_2^2 C_g^2 L_f^2}{864q^3 \gamma^3 L_{fg}^3 C_{fg}^3}\right)\frac{1}{(24)^2 L_{fg}^2 C_{fg}^2} + \frac{\gamma}{6qL_{fg}C_{fg}}\left(C_g^2 + L_g^2 + 2L_f^2 C_g^2\right)$ and $\Theta + \frac{BC_g^2}{(24)^2 L_{fg}^2 C_{fg}^2} \leq \frac{5}{48}$, we have*

$$\frac{1}{T}\sum_{t=1}^T \mathbb{E}\|\nabla F(\bar{x}_t)\| \leq \frac{\sqrt{2G}n^{1/6}}{T^{1/2}} + \frac{\sqrt{2G}}{T^{1/3}}, \tag{8}$$

*where $C_{fg}^2 = \max(C_f^2, C_g^2)$, $L_{fg}^2 = L_f^2 C_g^2 + L_g^2$, $G = \frac{4(F(\bar{x}_1)-F^*)}{k\gamma} + \frac{12n^{1/3}\sigma^2}{qk^2} + 4k^2\left(\frac{\hat{\delta}^2}{4\gamma^2 L_{fg}^2} + \frac{(c_1^2+c_2^2+c_3^2)\sigma^2}{3\gamma q L_{fg}C_{fg}}\right)\ln(n+T)$ and $\hat{\delta}^2 = 2c_1^2 L_f^2 \sigma^2 + c_3^2 \sigma^2 + 4c_3^2 \delta_f^2 + 4c_3^2 L_f^2 \delta_g^2 + c_2^2 \sigma^2 + 3c_2^2 \delta_g^2$.*

**Remark 3.** *The proof of Theorem 2 can totally follow the proofs of the above Theorem 1 with the parameter $\rho = 1$. Without loss of generality, let $k = O(1)$, $c_1 = O(1)$, $c_2 = O(1)$, $c_3 = O(1)$ and $n = O(q^3)$, we have and $G = \tilde{O}(1)$. Let $q = T^{1/3}$ and*

$$\frac{1}{T}\sum_{t=1}^T \mathbb{E}\|\nabla F(\bar{x}_t)\| \leq \tilde{O}\left(\frac{\sqrt{q}}{\sqrt{T}} + \frac{1}{T^{1/3}}\right) = \tilde{O}\left(\frac{1}{T^{1/3}}\right) \leq \epsilon, \tag{9}$$

*then we have $T = \tilde{O}(\epsilon^{-3})$. Since our MFCGD algorithm requires $2$ samples at each iteration expect for the first iteration requires $2q$ samples, it has a sample complexity of $2q + 2T = \tilde{O}(\epsilon^{-3})$. As the above AdaMFCGD algorithm, our MFCGD algorithm also obtain lower sample complexity of $\tilde{O}(\epsilon^{-3})$ and communication complexity of $\tilde{O}(\epsilon^{-2})$ in finding an $\epsilon$-stationary solution of the Problem (1).*

## 5 NUMERICAL EXPERIMENTS

In this section, we apply some numerical experiments to demonstrate efficiency of our MFCGD and AdaMFCGD algorithms on the robust federated learning and distributed meta learning tasks. Note that the experiment on distributed meta learning task is given in the Appendix B. In the experiments, we compare our algorithms with the existing federated composition optimization algorithms in Table 1 for solving distributed composition optimization problems.

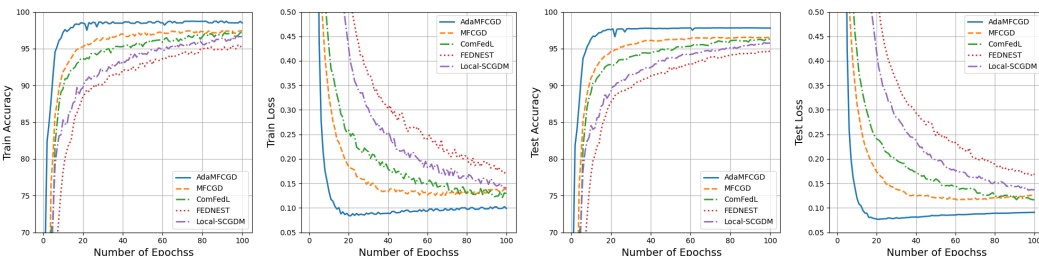

Figure 1: The performances of various FCO methods are evaluated on a synthetic imbalanced dataset based on MNIST, with a focus on addressing the distributed problem (10). The results are visualized using four plots: the first two show the accuracy(%) and loss on the training set, while the last two show the accuracy(%) and loss on the test set. These plots provide insights into how the FL methods perform on imbalanced data and their effectiveness in improving model performance while ensuring fairness across different clients.

## 5.1 ROBUST FEDERATED LEARNING

In this subsection, we evaluate the efficacy of our algorithms by performing a distributionally robust federated learning task defined in (3). Specifically, this robust federated learning problem can be rewritten as into a distributed composition optimization problem as in Huang et al. (2021a),

$$\min_{x \in \mathbb{R}^d} \frac{1}{M} \sum_{m=1}^{M} f\Big(g^m(x)/\lambda\Big), \tag{10}$$

where $f(\cdot) = \exp(\cdot/\lambda)$ and $\lambda > 0$ is a regularization parameter. In fact, we can also use some other monotonically increasing functions instead of $f(\cdot)$.

In the experiments, we tackle a multi-class classification problem on the MNISTLeCun et al. (2010) dataset with a 3-layer Convolutional Neural Network (CNN) which is widely used in this task. The experiments are conducted on a network comprising 10 clients and 1 server. To introduce data imbalance across clients, we randomly select one client to have a larger dataset of 5000 images, while the remaining clients have a significantly smaller dataset of only 20 images. This unequal distribution of data aims to create a challenging scenario where the algorithm must focus on the hardest and most important task, namely the client with the dominant number of images, to achieve good performance. In this way, we can test the algorithm's adaptability and ability to handle this highly imbalanced dataset. In these experiments, we performed a grid search to identify the optimal hyperparameters for each method, which we discussed in detail in subsection B.2. We set the learning rate to 0.01 for all methods and used in our Algorithm 1 for adaptive matrix generation. We fixed the total number of training steps to 500 and set the asynchronization step $q$ to 5 if not specified.

From Figure 1, we can find that our MFCGD and AdaMFCGD algorithms have a faster convergence rate and more stable optimization processes compared to the other composition federated optimization algorithms, partly contributes to momentum-based variance reduced strategy. Specifically, the experimental results show that our algorithms outperformed the existing composition federated optimization approaches, such as ComFedL Huang et al. (2021a), FEDNEST Tarzanagh et al. (2022), and Local-SCGDM Gao et al. (2022), in terms of both accuracy rates and cross-entropy losses. Moreover, the comparison between MFCGD and AdaMFCGD methods highlighted the advantage of the unified adaptive matrix, which flexibly incorporates various adaptive learning rates. Meanwhile, Figure 2 demonstrates the robustness of our AdaMFCGD algorithm by varying the asynchronization step $q$. It is worth noting that our AdaMFCGD achieves its optimal performance when $q = 1$, as this implies that an adaptive matrix is calculated in each iteration, allowing the momentum-based variance reduction technique to fully showcase its potential.

Figure 2 shows the robustness of our AdaMFCGD algorithm by varying the regularization parameter $\lambda$. The results indicate that our AdaMFCGD algorithm achieves good test accuracy and low test loss with different $\lambda$. Moreover, when decreasing $\lambda$, our AdaMFCGD algorithm converges much faster, especially when running multiple local epochs. As in Huang et al. (2021a), $\lambda$ is a penalty parameter

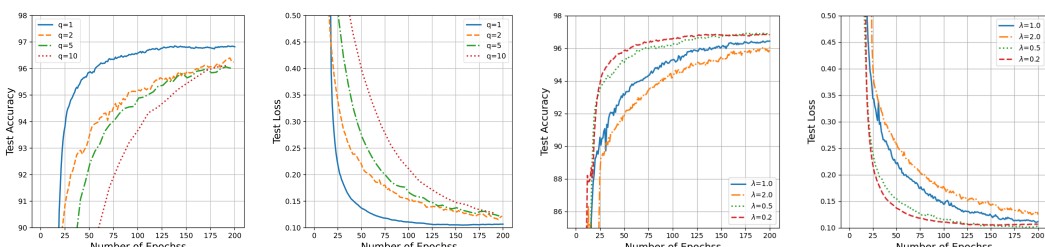

Figure 2: Comparing the accuracy(%) (left) and cross-entropy loss (right) on different synchronization step $q$ and different regularization parameter $\lambda$, respectively, on our AdaMFCGD algorithm

in distributionally robust FL that penalizes divergence between $r = (r_1, \cdots, r_M)$ and $\mathbf{1}/M = (1/M, \cdots, 1/M)$, where $r_m \in (0, 1)$ for any $m \in [M]$ denotes the proportion of $m$-th client in the entire mode. A larger $\lambda$ means places more emphasis on bringing the original proportion parameter $r_m$ closer to the average weight $1/M$. Since the optimal weights in our dataset are far from $1/M$, we should select $\lambda$ relatively small. The results in Figure 2 also confirm that our AdaMFCGD algorithm converges much faster with smaller $\lambda$. Specifically, our AdaMFCGD algorithm converges much faster, when choosing $\lambda = 0.2$ or $\lambda = 0.5$ compared to that $\lambda = 2$ in terms of both accuracy rate, cross-entropy loss on true labels and convergence speed.

## 6 CONCLUSION

In the paper, we proposed a class of faster adaptive momentum-based federated compositional gradient descent methods to solve the nonconvex distributed composition problems based on the local-SGD and momentum-based variance reduced techniques. In particular, our adaptive algorithm (i.e., AdaMFCGD) uses a unified adaptive matrix to flexibly incorporate various adaptive learning rates further accelerate algorithm. Moreover, we established a solid convergence analysis framework for our methods, and proved that they obtain lower sample and communication complexities simultaneously than the existing federated composition optimization methods.

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

## A CONVERGENCE ANALYSIS

In this section, we provide the detailed convergence analysis of our algorithms.

We first introduce some useful notations: $\bar{w}_t = \frac{1}{M} \sum_{m=1}^{M} w_t^m$, $\bar{x}_t = \frac{1}{M} \sum_{m=1}^{M} x_t^m$,

$$F(x) = \frac{1}{M} \sum_{m=1}^{M} f^m(g^m(x)), \quad \nabla F(x) = \frac{1}{M} \sum_{m=1}^{M} \left(\nabla g^m(x)\right)^T \nabla f^m(g^m(x)).$$

Next, we review and provide some useful lemmas.

**Lemma 1.** *Given $M$ vectors $\{u^m\}_{m=1}^{M}$, the following inequalities satisfy: $||u^m + u^j||^2 \leq (1 + c)||u^m||^2 + (1 + \frac{1}{c})||u^j||^2$ for any $c > 0$, and $||\sum_{m=1}^{M} u^m||^2 \leq M \sum_{m=1}^{M} ||u^m||^2$.*

**Lemma 2.** *Given a finite sequence $\{u^m\}_{m=1}^{M}$, and $\bar{u} = \frac{1}{M} \sum_{m=1}^{M} u^m$, the following inequality satisfies $\sum_{m=1}^{M} ||u^m - \bar{u}||^2 \leq \sum_{m=1}^{M} ||u^m||^2$.*

Given a $\rho$-strongly convex function $\varphi(x)$, we define a prox-function (Bregman distance) Censor & Lent (1981); Censor & Zenios (1992) associated with $\varphi(x)$ as follows:

$$D(z, x) = \varphi(z) - \left[\varphi(x) + \langle \nabla \varphi(x), z - x \rangle\right]. \tag{11}$$

Then we define a generalized projection problem as in Ghadimi et al. (2016):

$$x^+ = \arg\min_{z \in \mathcal{X}} \left\{ \langle z, w \rangle + \frac{1}{\gamma} D(z, x) + h(z) \right\}, \tag{12}$$

where $\mathcal{X} \subseteq \mathbb{R}^d$, $w \in \mathbb{R}^d$ and $\gamma > 0$. In the paper, we consider $h(x) = 0$. Meanwhile, we also define a generalized projected gradient (a.k.a., gradient mapping):

$$\mathcal{G}_{\mathcal{X}}(x, w, \gamma) = \frac{x - x^+}{\gamma}. \tag{13}$$

**Lemma 3.** *(Lemma 1 in Ghadimi et al. (2016)) Let $x^+$ be given in (12). Then, for any $x \in \mathcal{X}$, $w \in \mathbb{R}^d$ and $\gamma > 0$, we have*

$$\langle w, \mathcal{G}_{\mathcal{X}}(x, w, \gamma) \rangle \geq \rho \|\mathcal{G}_{\mathcal{X}}(x, w, \gamma)\|^2 + \frac{1}{\gamma} \left[h(x^+) - h(x)\right], \tag{14}$$

*where $\rho > 0$ depends on $\rho$-strongly convex function $\varphi(x)$.*

When $h(x) = 0$, in the above lemma 3, we have

$$\langle w, \mathcal{G}_{\mathcal{X}}(x, w, \gamma) \rangle \geq \rho \|\mathcal{G}_{\mathcal{X}}(x, w, \gamma)\|^2. \tag{15}$$

**Lemma 4.** *(Restatement of Lemma 1) Given the above Assumptions 1-2, the function $F(x)$ is $L$-smooth, i.e., for any $x_1, x_2 \in \mathbb{R}^d$, we have*

$$\|\nabla F(x_1) - \nabla F(x_2)\|^2 \leq L^2 \|x_1 - x_2\|^2, \tag{16}$$

*where $L = \sqrt{2C_f^2 L_g^2 + 2C_g^4 L_f^2}$.*

*Proof.* Based on Assumptions 1-2, the deterministic functions $f^m(y) = \mathbb{E}\left[f^m(y; \xi^m)\right]$ $g^m(x) = \mathbb{E}\left[f^m(x; \zeta^m)\right]$ and its gradients also satisfy the Lipschitz gradients and bounded gradients. For example, for any $y_1, y_2 \in \mathbb{R}^n$

$$\begin{aligned}
\left\|\nabla f^m(y_1) - \nabla f^m(y_1)\right\| &= \left\|\mathbb{E}\left[\nabla f^m(y_1; \xi^m) - \nabla f^m(y_1; \xi^m)\right]\right\| \\
&\leq \mathbb{E}\left\|\nabla f^m(y_1; \xi^m) - \nabla f^m(y_1; \xi^m)\right\| \leq L_f \|y_1 - y_2\|,
\end{aligned} \tag{17}$$

where the first inequality holds by Jensen's inequality, and the last inequality holds by Assumption 1.

Since $F(x) = \frac{1}{M} \sum_{m=1}^{M} f^m(g^m(x))$, we have

$$\|\nabla F(x_1) - \nabla F(x_2)\|^2$$

$$= \Big\| \frac{1}{M} \sum_{m=1}^{M} \left(\nabla g^m(x_1)\right)^T \nabla f^m(g^m(x_1)) - \frac{1}{M} \sum_{m=1}^{M} \left(\nabla g^m(x_2)\right)^T \nabla f^m(g^m(x_2)) \Big\|^2$$

$$\leq \frac{1}{M} \sum_{m=1}^{M} \left\| \left(\nabla g^m(x_1)\right)^T \nabla f^m(g^m(x_1)) - \left(\nabla g^m(x_2)\right)^T \nabla f^m(g^m(x_2)) \right\|^2$$

$$= \frac{1}{M} \sum_{m=1}^{M} \Big\| \left(\nabla g^m(x_1)\right)^T \nabla f^m(g^m(x_1)) - \left(\nabla g^m(x_2)\right)^T \nabla f^m(g^m(x_1)) + \left(\nabla g^m(x_2)\right)^T \nabla f^m(g^m(x_1))$$

$$- \left(\nabla g^m(x_2)\right)^T \nabla f^m(g^m(x_2)) \Big\|^2$$

$$\leq \frac{1}{M} \sum_{m=1}^{M} 2C_f^2 \left\| \nabla g^m(x_1) - \nabla g^m(x_2) \right\|^2 + \frac{1}{M} \sum_{m=1}^{M} 2C_g^2 \left\| \nabla f^m(g^m(x_1)) - \nabla f^m(g^m(x_2)) \right\|^2$$

$$\leq 2C_f^2 L_g^2 \|x_1 - x_2\|^2 + 2C_g^4 L_f^2 \|x_1 - x_2\|^2 = \left(2C_f^2 L_g^2 + 2C_g^4 L_f^2\right) \|x_1 - x_2\|^2, \tag{18}$$

where the second last and the last inequalities hold by Assumptions 1-2.

$\square$

**Lemma 5.** *(Restatement of Lemma 2) Assume the gradient estimator $\{\bar{w}_t\}_{t=1}^T$ generated from Algorithm 1, where $w_t = \frac{1}{M} \sum_{m=1}^{M} w_t^m$, we have*

$$\|\bar{w}_t - \nabla F(\bar{x}_t)\|^2 \leq \frac{1}{M} \sum_{m=1}^{M} \Big( 2C_f^2 \|u_t^m - \nabla g^m(\bar{x}_t)\|^2 + 4C_g^2 \|v_t^m - \nabla f^m(h_t^m)\|^2 + 4C_g^2 L_f^2 \|h_t^m - g^m(\bar{x}_t)\|^2 \Big).$$

$$\tag{19}$$

*Proof.* Since $\bar{w}_t = \frac{1}{M} \sum_{m=1}^{M} (u_t^m)^T v_t^m$, we have

$$\|\bar{w}_t - \nabla F(\bar{x}_t)\|^2$$

$$= \Big\| \frac{1}{M} \sum_{m=1}^{M} (u_t^m)^T v_t^m - \frac{1}{M} \sum_{m=1}^{M} \left(\nabla g^m(\bar{x}_t)\right)^T \nabla f^m(g^m(\bar{x}_t)) \Big\|^2$$

$$= \Big\| \frac{1}{M} \sum_{m=1}^{M} (u_t^m)^T v_t^m - \frac{1}{M} \sum_{m=1}^{M} \left(\nabla g^m(\bar{x}_t)\right)^T v_t^m + \frac{1}{M} \sum_{m=1}^{M} \left(\nabla g^m(\bar{x}_t)\right)^T v_t^m - \frac{1}{M} \sum_{m=1}^{M} \left(\nabla g^m(\bar{x}_t)\right)^T \nabla f^m(g^m(\bar{x}_t)) \Big\|^2$$

$$\leq \frac{1}{M} \sum_{m=1}^{M} 2C_f^2 \|u_t^m - \nabla g^m(\bar{x}_t)\|^2 + \frac{1}{M} \sum_{m=1}^{M} 2C_g^2 \|v_t^m - \nabla f^m(g^m(\bar{x}_t))\|^2$$

$$= \frac{2C_f^2}{M} \sum_{m=1}^{M} \|u_t^m - \nabla g^m(\bar{x}_t)\|^2 + \frac{2C_g^2}{M} \sum_{m=1}^{M} \|v_t^m - \nabla f^m(h_t^m) + \nabla f^m(h_t^m) - \nabla f^m(g^m(\bar{x}_t))\|^2$$

$$\leq \frac{2C_f^2}{M} \sum_{m=1}^{M} \|u_t^m - \nabla g^m(\bar{x}_t)\|^2 + \frac{4C_g^2}{M} \sum_{m=1}^{M} \|v_t^m - \nabla f^m(h_t^m)\|^2 + \frac{4C_g^2 L_f^2}{M} \sum_{m=1}^{M} \|h_t^m - g^m(\bar{x}_t)\|^2, \tag{20}$$

where the first inequality is due to Assumptions 1-2 and the above Lemma 1.

$\square$

**Lemma 6.** *Suppose that the sequence $\{\bar{x}_t\}_{t=1}^{T}$ be generated from Algorithm 1, where $\bar{x}_t = \frac{1}{M}\sum_{m=1}^{M} x_t^m$. Let $0 < \gamma \leq \frac{\rho}{2L\eta_t}$, then we have*

$$
F(\bar{x}_{t+1}) \leq F(\bar{x}_t) + \frac{1}{M}\sum_{m=1}^{M}\Big(\frac{2C_f^2\eta_t\gamma}{\rho}\|u_t^m - \nabla g^m(\bar{x}_t)\|^2 + \frac{4C_g^2\eta_t\gamma}{\rho}\|v_t^m - \nabla f^m(h_t^m)\|^2
$$

$$
+ \frac{4C_g^2 L_f^2\eta_t\gamma}{\rho}\|h_t^m - g^m(\bar{x}_t)\|^2\Big) - \frac{\rho}{2\eta_t\gamma}\|\bar{x}_{t+1} - \bar{x}_t\|^2. \tag{21}
$$

*Proof.* According to the above Lemma 4, the function $F(x)$ is $L$-smooth. Thus we have

$$
F(\bar{x}_{t+1}) \leq F(\bar{x}_t) + \langle\nabla F(\bar{x}_t), \bar{x}_{t+1} - \bar{x}_t\rangle + \frac{L}{2}\|\bar{x}_{t+1} - \bar{x}_t\|^2 \tag{22}
$$

$$
= F(\bar{x}_t) + \underbrace{\langle\bar{w}_t, \bar{x}_{t+1} - \bar{x}_t\rangle}_{=T_1} + \underbrace{\langle\nabla F(\bar{x}_t) - \bar{w}_t, \bar{x}_{t+1} - \bar{x}_t\rangle}_{=T_2} + \frac{L}{2}\|\bar{x}_{t+1} - \bar{x}_t\|^2.
$$

According to Assumption 5, i.e., $A_t \succ \rho I_d$ for any $t \geq 1$, the mirror function $\varphi_t(x) = \frac{1}{2}x^T A_t x$ is $\rho$-strongly convex, then we can define a Bregman distance as in Ghadimi et al. (2016),

$$
D_t(x, \bar{x}_t) = \varphi_t(x) - \big[\varphi_t(\bar{x}_t) + \langle\nabla\varphi_t(\bar{x}_t), x - \bar{x}_t\rangle\big] = \frac{1}{2}(x - \bar{x}_t)^T A_t(x - \bar{x}_t). \tag{23}
$$

When $t = s_t = q\lfloor t/q\rfloor + 1$, according to the line 7 of Algorithm 1, we have $\bar{x}_{t+1} = \arg\min_{x\in\mathbb{R}^d}\big\{\langle\bar{w}_t, x\rangle + \frac{1}{2\eta_t\gamma}(x - \bar{x}_t)^T A_t(x - \bar{x}_t)\big\}$. By using Lemma 1 in Ghadimi et al. (2016) to the problem $\bar{x}_{t+1} = \arg\min_{x\in\mathbb{R}^d}\big\{\langle\bar{w}_t, x\rangle + \frac{1}{2\eta_t\gamma}(x - \bar{x}_t)^T A_t(x - \bar{x}_t)\big\}$, we can obtain

$$
\langle\bar{w}_t, \frac{1}{\eta_t\gamma}(\bar{x}_t - \bar{x}_{t+1})\rangle \geq \rho\|\frac{1}{\eta_t\gamma}(\bar{x}_t - \bar{x}_{t+1})\|^2. \tag{24}
$$

When $t \in (s_t, s_t + q)$, according to the line 11 of Algorithm 1, we have $x_{t+1}^m = \arg\min_{x\in\mathbb{R}^d}\big\{\langle w_t^m, x\rangle + \frac{1}{2\eta_t\gamma}(x - x_t^m)^T A_t(x - x_t^m)\big\}$. Similarly, we have

$$
\langle w_t^m, \frac{1}{\eta_t\gamma}(x_t^m - x_{t+1}^m)\rangle \geq \rho\|\frac{1}{\eta_t\gamma}(x_t^m - x_{t+1}^m)\|^2. \tag{25}
$$

Then we have

$$
\frac{1}{M}\sum_{m=1}^{M}\langle w_t^m, \frac{1}{\eta_t\gamma}(x_t^m - x_{t+1}^m)\rangle \geq \rho\frac{1}{M}\sum_{m=1}^{M}\|\frac{1}{\eta_t\gamma}(x_t^m - x_{t+1}^m)\|^2
$$

$$
\geq \rho\|\frac{1}{\eta_t\gamma}\frac{1}{M}\sum_{m=1}^{M}(x_t^m - x_{t+1}^m)\|^2 = \rho\|\frac{1}{\eta_t\gamma}(\bar{x}_t - \bar{x}_{t+1})\|^2. \tag{26}
$$

Thus we have

$$
\langle w_t^m, \frac{1}{\eta_t\gamma}(\bar{x}_t - \bar{x}_{t+1})\rangle \geq \rho\|\frac{1}{\eta_t\gamma}(\bar{x}_t - \bar{x}_{t+1})\|^2. \tag{27}
$$

Since $\bar{w}_t = \frac{1}{M}\sum_{m=1}^{M} w_t^m$, averaging the above inequality (27) from $m = 1$ to $M$, we can obtain

$$
\langle\bar{w}_t, \frac{1}{\eta_t\gamma}(\bar{x}_t - \bar{x}_{t+1})\rangle = \frac{1}{M}\sum_{m=1}^{M}\langle w_t^m, \frac{1}{\eta_t\gamma}(\bar{x}_t - \bar{x}_{t+1})\rangle
$$

$$
\geq \rho\frac{1}{M}\sum_{m=1}^{M}\|\frac{1}{\eta_t\gamma}(\bar{x}_t - \bar{x}_{t+1})\|^2 = \rho\|\frac{1}{\eta_t\gamma}(\bar{x}_t - \bar{x}_{t+1})\|^2. \tag{28}
$$

Then we have for any $t \in [s_t, s_t + q)$,

$$
T_1 = \langle\bar{w}_t, \bar{x}_{t+1} - \bar{x}_t\rangle \leq -\frac{\rho}{\eta_t\gamma}\|\bar{x}_{t+1} - \bar{x}_t\|^2. \tag{29}
$$

Since $s_t = q\lfloor t/q \rfloor + 1$ and all $t \in [s_t, s_t + q)$, clearly, we have, for all $t \geq 1$

$$T_1 = \langle \bar{w}_t, \bar{x}_{t+1} - \bar{x}_t \rangle \leq -\frac{\rho}{\eta_t \gamma} \|\bar{x}_{t+1} - \bar{x}_t\|^2. \tag{30}$$

Next, consider the bound of the term $T_2$, we have

$$
\begin{aligned}
T_2 &= \langle \nabla F(\bar{x}_t) - \bar{w}_t, \bar{x}_{t+1} - \bar{x}_t \rangle \\
&\leq \|\nabla F(\bar{x}_t) - \bar{w}_t\| \cdot \|\bar{x}_{t+1} - \bar{x}_t\| \\
&\leq \frac{\eta_t \gamma}{\rho} \|\nabla F(\bar{x}_t) - \bar{w}_t\|^2 + \frac{\rho}{4\eta_t \gamma} \|\bar{x}_{t+1} - \bar{x}_t\|^2,
\end{aligned}
\tag{31}
$$

where the first inequality is due to the Cauchy-Schwarz inequality and the last is due to Young's inequality. By combining the above inequalities (22), (30) with (31), we obtain

$$
\begin{aligned}
F(\bar{x}_{t+1}) &\leq F(\bar{x}_t) + \langle \nabla F(\bar{x}_t) - \bar{w}_t, \bar{x}_{t+1} - \bar{x}_t \rangle + \langle \bar{w}_t, \bar{x}_{t+1} - \bar{x}_t \rangle + \frac{L}{2} \|\bar{x}_{t+1} - \bar{x}_t\|^2 \\
&\leq F(\bar{x}_t) + \frac{\eta_t \gamma}{\rho} \|\nabla F(\bar{x}_t) - \bar{w}_t\|^2 + \frac{\rho}{4\eta_t \gamma} \|\bar{x}_{t+1} - \bar{x}_t\|^2 - \frac{\rho}{\eta_t \gamma} \|\bar{x}_{t+1} - \bar{x}_t\|^2 + \frac{L}{2} \|\bar{x}_{t+1} - \bar{x}_t\|^2 \\
&= F(\bar{x}_t) + \frac{\eta_t \gamma}{\rho} \|\nabla F(\bar{x}_t) - \bar{w}_t\|^2 - \frac{\rho}{2\eta_t \gamma} \|\bar{x}_{t+1} - \bar{x}_t\|^2 - \left(\frac{\rho}{4\eta_t \gamma} - \frac{L}{2}\right) \|\bar{x}_{t+1} - \bar{x}_t\|^2 \\
&\leq F(\bar{x}_t) + \frac{\eta_t \gamma}{\rho} \|\nabla F(\bar{x}_t) - \bar{w}_t\|^2 - \frac{\rho}{2\eta_t \gamma} \|\bar{x}_{t+1} - \bar{x}_t\|^2 \\
&\leq F(\bar{x}_t) + \frac{1}{M} \sum_{m=1}^{M} \left( \frac{2C_f^2 \eta_t \gamma}{\rho} \|u_t^m - \nabla g^m(\bar{x}_t)\|^2 + \frac{4C_g^2 \eta_t \gamma}{\rho} \|v_t^m - \nabla f^m(h_t^m)\|^2 \right. \\
&\quad \left. + \frac{4C_g^2 L_f^2 \eta_t \gamma}{\rho} \|h_t^m - g^m(\bar{x}_t)\|^2 \right) - \frac{\rho}{2\eta_t \gamma} \|\bar{x}_{t+1} - \bar{x}_t\|^2,
\end{aligned}
\tag{32}
$$

where the second last inequality is due to $0 < \gamma \leq \frac{\rho}{2L\eta_t}$, and the last inequality holds by Lemma 5.

$\square$

**Lemma 7.** *Under the above assumptions, and assume the stochastic gradient estimators $\{h_t^m, u_t^m, v_t^m\}_{t=1}^T$ be generated from Algorithm 1, we have, for any $m \in [M]$*

$$
\begin{aligned}
\mathbb{E}\|h_{t+1}^m - g^m(x_{t+1}^m)\|^2 &\leq (1 - \alpha_{t+1})\mathbb{E}\|h_t^m - g^m(x_t^m)\|^2 + 2\alpha_{t+1}^2 \sigma^2 \\
&\quad + 2C_g^2 \mathbb{E}\|x_{t+1}^m - x_t^m\|^2,
\end{aligned}
\tag{33}
$$

$$
\begin{aligned}
\mathbb{E}\|u_{t+1}^m - \nabla g^m(x_{t+1}^m)\|^2 &\leq (1 - \beta_{t+1})\mathbb{E}\|u_t^m - \nabla g^m(x_t^m)\|^2 + 2\beta_{t+1}^2 \sigma^2 \\
&\quad + 2L_g^2 \mathbb{E}\|x_{t+1}^m - x_t^m\|^2.
\end{aligned}
\tag{34}
$$

$$
\begin{aligned}
\mathbb{E}\|v_{t+1}^m - \nabla f^m(h_{t+1}^m)\|^2 &\leq (1 - \varrho_{t+1})\mathbb{E}\|v_t^m - \nabla f^m(h_t^m)\|^2 + 4L_f^2 C_g^2 \mathbb{E}\|x_{t+1}^m - x_t^m\|^2 \\
&\quad + 2\varrho_{t+1}^2 \sigma^2 + 8\alpha_{t+1}^2 L_f^2 \mathbb{E}\|h_t^m - g^m(x_t^m)\|^2 + 8L_f^2 \alpha_{t+1}^2 \sigma^2,
\end{aligned}
\tag{35}
$$

*Proof.* Without loss of generality, we only prove the above inequality (35), and it is similar to the other inequalities. Since $v_{t+1}^m = \Pi_{C_f} \left[ \nabla f^m(h_{t+1}^m; \xi_{t+1}^m) + (1 - \varrho_{t+1})\left(v_t^m - \nabla f^m(h_t^m; \xi_{t+1}^m)\right) \right]$, we

have

$$\mathbb{E}\|v_{t+1}^m - \nabla f^m(h_{t+1}^m)\|^2$$

$$= \mathbb{E}\big\|\Pi_{C_f}\big[\nabla f^m(h_{t+1}^m; \xi_{t+1}^m) + (1-\varrho_{t+1})\big(v_t^m - \nabla f^m(h_t^m; \xi_{t+1}^m)\big)\big] - \Pi_{C_f}\big[\nabla f^m(h_{t+1}^m)\big]\big\|^2$$

$$\leq \mathbb{E}\|\nabla f^m(h_{t+1}^m; \xi_{t+1}^m) + (1-\varrho_{t+1})\big(v_t^m - \nabla f^m(h_t^m; \xi_{t+1}^m)\big) - \nabla f^m(h_{t+1}^m)\|^2$$

$$= \mathbb{E}\big\|(1-\varrho_{t+1})(v_t^m - \nabla f^m(h_t^m)) - \varrho_{t+1}(\nabla f^m(h_{t+1}^m) - \nabla f^m(h_{t+1}^m; \xi_{t+1}^m))$$

$$\quad + (1-\varrho_{t+1})\big(\nabla f^m(h_{t+1}^m; \xi_{t+1}^m) - \nabla f^m(h_t^m; \xi_{t+1}^m) - \nabla f^m(h_{t+1}^m) + \nabla f^m(h_t^m)\big)\big\|^2$$

$$= (1-\varrho_{t+1})^2\mathbb{E}\|v_t^m - \nabla f^m(h_t^m)\|^2 + \mathbb{E}\big\|\varrho_{t+1}(\nabla f^m(h_{t+1}^m) - \nabla f^m(h_{t+1}^m; \xi_{t+1}^m))$$

$$\quad - (1-\varrho_{t+1})\big(\nabla f^m(h_{t+1}^m; \xi_{t+1}^m) - \nabla f^m(h_t^m; \xi_{t+1}^m) - \nabla f^m(h_{t+1}^m) + \nabla f^m(h_t^m)\big)\big\|^2$$

$$\leq (1-\varrho_{t+1})^2\mathbb{E}\|v_t^m - \nabla f^m(h_t^m)\|^2 + 2\varrho_{t+1}^2\mathbb{E}\big\|\nabla f^m(h_{t+1}^m) - \nabla f^m(h_{t+1}^m; \xi_{t+1}^m)\big\|^2$$

$$\quad + 2(1-\varrho_{t+1})^2\big\|\nabla f^m(h_{t+1}^m; \xi_{t+1}^m) - \nabla f^m(h_t^m; \xi_{t+1}^m) - \nabla f^m(h_{t+1}^m) + \nabla f^m(h_t^m)\big\|^2$$

$$\leq (1-\varrho_{t+1})^2\mathbb{E}\|v_t^m - \nabla f^m(h_t^m)\|^2 + 2\varrho_{t+1}^2\sigma^2 + 2(1-\varrho_{t+1})^2\big\|\nabla f^m(h_{t+1}^m; \xi_{t+1}^m) - \nabla f^m(h_t^m; \xi_{t+1}^m)\big\|^2$$

$$\leq (1-\varrho_{t+1})^2\mathbb{E}\|v_t^m - \nabla f^m(h_t^m)\|^2 + 2\varrho_{t+1}^2\sigma^2 + 2(1-\varrho_{t+1})^2 L_f^2\mathbb{E}\|h_{t+1}^m - h_t^m\|^2, \tag{36}$$

where the third equality holds by the following fact:

$$\mathbb{E}_{\xi_{t+1}^m}\big[\varrho_{t+1}(\nabla f^m(h_{t+1}^m) - \nabla f^m(h_{t+1}^m; \xi_{t+1}^m)) - (1-\varrho_{t+1})\big(\nabla f^m(h_{t+1}^m; \xi_{t+1}^m) - \nabla f^m(h_t^m; \xi_{t+1}^m)$$

$$\quad - \nabla f^m(h_{t+1}^m) + \nabla f^m(h_t^m)\big)\big] = 0,$$

and the second last inequality holds by the inequality $\mathbb{E}\|\zeta - \mathbb{E}[\zeta]\|^2 \leq \mathbb{E}\|\zeta\|^2$ and Assumption 3; the last inequality is due to Assumption 1.

Since $h_{t+1}^m = g^m(x_{t+1}^m; \zeta_{t+1}^m) + (1-\alpha_{t+1})\big(h_t^m - g^m(x_t^m; \zeta_{t+1}^m)\big)$, we have

$$\mathbb{E}\|h_{t+1}^m - h_t^m\|^2 = \mathbb{E}\|g^m(x_{t+1}^m; \zeta_{t+1}^m) - g^m(x_t^m; \zeta_{t+1}^m) - \alpha_{t+1}\big(h_t^m - g^m(x_t^m; \zeta_{t+1}^m)\big)\|^2$$

$$\leq 2\mathbb{E}\|g^m(x_{t+1}^m; \zeta_{t+1}^m) - g^m(x_t^m; \zeta_{t+1}^m)\|^2 + 2\alpha_{t+1}^2\mathbb{E}\|h_t^m - g^m(x_t^m; \zeta_{t+1}^m)\|^2$$

$$\leq 2C_g^2\|x_{t+1}^m - x_t^m\|^2 + 2\alpha_{t+1}^2\mathbb{E}\|h_t^m - g^m(x_t^m; \zeta_{t+1}^m)\|^2$$

$$= 2C_g^2\|x_{t+1}^m - x_t^m\|^2 + 2\alpha_{t+1}^2\mathbb{E}\|h_t^m - g^m(x_t^m; \zeta_{t+1}^m) + g^m(x_t^m) - g^m(x_t^m)\|^2$$

$$\leq 2C_g^2\|x_{t+1}^m - x_t^m\|^2 + 4\alpha_{t+1}^2\mathbb{E}\|h_t^m - g^m(x_t^m)\|^2 + 4\alpha_{t+1}^2\mathbb{E}\|g^m(x_t^m; \zeta_{t+1}^m) + g^m(x_t^m)\|^2$$

$$\leq 2C_g^2\|x_{t+1}^m - x_t^m\|^2 + 4\alpha_{t+1}^2\mathbb{E}\|h_t^m - g^m(x_t^m)\|^2 + 4\alpha_{t+1}^2\sigma^2, \tag{37}$$

where the second inequality holds by Assumption , .

Combining the above inequalities (36) with (37), we have

$$\mathbb{E}\|v_{t+1}^m - \nabla f^m(h_{t+1}^m)\|^2$$

$$\leq (1-\varrho_{t+1})^2\mathbb{E}\|v_t^m - \nabla f^m(h_t^m)\|^2 + 2\varrho_{t+1}^2\sigma^2 + 2(1-\varrho_{t+1})^2 L_f^2\mathbb{E}\|h_{t+1}^m - h_t^m\|^2$$

$$\leq (1-\varrho_{t+1})\mathbb{E}\|v_t^m - \nabla f^m(h_t^m)\|^2 + 2\varrho_{t+1}^2\sigma^2 + 4L_f^2 C_g^2\|x_{t+1}^m - x_t^m\|^2$$

$$\quad + 8\alpha_{t+1}^2 L_f^2\mathbb{E}\|h_t^m - g^m(x_t^m)\|^2 + 8L_f^2\alpha_{t+1}^2\sigma^2,$$

where the last inequality holds by $0 < \varrho_{t+1} \leq 1$.

$$\square$$

**Lemma 8.** *Based on the above Assumptions 1-2 and 6, we have*

$$\sum_{m=1}^M \mathbb{E}\big\|\nabla f^m(h_t^m) - \frac{1}{M}\sum_{j=1}^M \nabla f^j(h_t^j)\big\|^2 \leq 8L_f^2\sum_{m=1}^M \mathbb{E}\|h_t^m - g^m(\bar{x}_t)\|^2 + 4M\delta_f^2 + 4ML_f^2\delta_g^2,$$

$$\sum_{m=1}^M \mathbb{E}\big\|\nabla g^m(x_t^m) - \frac{1}{M}\sum_{j=1}^M \nabla g^j(x_t^j)\big\|^2 \leq 6L_g^2\sum_{m=1}^M \mathbb{E}\|x_t^m - \bar{x}_t\|^2 + 3M\delta_g^2$$

$$\sum_{m=1}^M \mathbb{E}\big\|g^m(x_t^m) - \frac{1}{M}\sum_{j=1}^M g^j(x_t^j)\big\|^2 \leq 6C_g^2\sum_{m=1}^M \mathbb{E}\|x_t^m - \bar{x}_t\|^2 + 3M\delta_g^2.$$

*Proof.* Consider the term $\sum_{m=1}^{M} \mathbb{E} \big\| \nabla f^m(h_t^m) - \frac{1}{M} \sum_{j=1}^{M} \nabla f^j(h_t^j) \big\|^2$, we have

$$\sum_{m=1}^{M} \mathbb{E} \big\| \nabla f^m(h_t^m) - \frac{1}{M} \sum_{j=1}^{M} \nabla f^j(h_t^j) \big\|^2$$

$$= \sum_{m=1}^{M} \mathbb{E} \big\| \nabla f^m(h_t^m) - \nabla f^m(g^m(\bar{x}_t)) + \nabla f^m(g^m(\bar{x}_t)) - \frac{1}{M} \sum_{j=1}^{M} \nabla f^j(g^m(\bar{x}_t)) + \frac{1}{M} \sum_{j=1}^{M} \nabla f^j(g^m(\bar{x}_t))$$

$$- \frac{1}{M} \sum_{j=1}^{M} \nabla f^j(g^j(\bar{x}_t)) + \frac{1}{M} \sum_{j=1}^{M} \nabla f^j(g^j(\bar{x}_t)) - \frac{1}{M} \sum_{j=1}^{M} \nabla f^j(h_t^j) \big\|^2$$

$$\leq \sum_{m=1}^{M} 4\mathbb{E} \big\| \nabla f^m(h_t^m) - \nabla f^m(g^m(\bar{x}_t)) \big\|^2 + \sum_{m=1}^{M} 4\mathbb{E} \big\| \nabla f^m(g^m(\bar{x}_t)) - \frac{1}{M} \sum_{j=1}^{M} \nabla f^j(g^m(\bar{x}_t)) \big\|^2$$

$$+ \sum_{m=1}^{M} 4\mathbb{E} \big\| \frac{1}{M} \sum_{j=1}^{M} \nabla f^j(g^m(\bar{x}_t)) - \frac{1}{M} \sum_{j=1}^{M} \nabla f^j(g^j(\bar{x}_t)) \big\|^2 + \sum_{m=1}^{M} 4\mathbb{E} \big\| \frac{1}{M} \sum_{j=1}^{M} \nabla f^j(g^j(\bar{x}_t)) - \frac{1}{M} \sum_{j=1}^{M} \nabla f^j(h_t^j) \big\|^2$$

$$\leq 4L_f^2 \sum_{m=1}^{M} \mathbb{E} \| h_t^m - g^m(\bar{x}_t) \|^2 + 4 \sum_{m=1}^{M} \frac{1}{M} \sum_{j=1}^{M} \mathbb{E} \| \nabla f^m(g^m(\bar{x}_t)) - \nabla f^j(g^m(\bar{x}_t)) \|^2$$

$$+ 4L_f^2 \sum_{m=1}^{M} \frac{1}{M} \sum_{j=1}^{M} \big\| g^m(\bar{x}_t) - g^j(\bar{x}_t) \big\|^2 + 4L_f^2 \sum_{j=1}^{M} \frac{1}{M} \sum_{m=1}^{M} \mathbb{E} \big\| g^m(\bar{x}_t) - h_t^m \big\|^2$$

$$\leq 8L_f^2 \sum_{m=1}^{M} \mathbb{E} \| h_t^m - g^m(\bar{x}_t) \|^2 + 4M\delta_f^2 + 4ML_f^2 \delta_g^2, \tag{38}$$

where the last inequality holds by Assumption 6.

Next, we have

$$\sum_{m=1}^{M} \mathbb{E} \big\| \nabla g^m(x_t^m) - \frac{1}{M} \sum_{j=1}^{M} \nabla g^j(x_t^j) \big\|^2$$

$$= \sum_{m=1}^{M} \mathbb{E} \big\| \nabla g^m(x_t^m) - \nabla g^m(\bar{x}_t) + \nabla g^m(\bar{x}_t) - \frac{1}{M} \sum_{j=1}^{M} \nabla g^j(\bar{x}_t) + \frac{1}{M} \sum_{j=1}^{M} \nabla g^j(\bar{x}_t) - \frac{1}{M} \sum_{j=1}^{M} \nabla g^j(x_t^j) \big\|^2$$

$$\leq \sum_{m=1}^{M} 3\mathbb{E} \big\| \nabla g^m(x_t^m) - \nabla g^m(\bar{x}_t) \big\|^2 + \sum_{m=1}^{M} 3\mathbb{E} \big\| \nabla g^m(\bar{x}_t) - \frac{1}{M} \sum_{j=1}^{M} \nabla g^j(\bar{x}_t) \big\|^2$$

$$+ \sum_{m=1}^{M} 3\mathbb{E} \big\| \frac{1}{M} \sum_{j=1}^{M} \nabla g^j(\bar{x}_t) - \frac{1}{M} \sum_{j=1}^{M} \nabla g^j(x_t^j) \big\|^2$$

$$\leq 3L_g^2 \sum_{m=1}^{M} \mathbb{E} \| x_t^m - \bar{x}_t \|^2 + 3 \sum_{m=1}^{M} \frac{1}{M} \sum_{j=1}^{M} \mathbb{E} \| \nabla g^m(\bar{x}_t) - \nabla g^j(\bar{x}_t) \|^2 + 3 \sum_{m=1}^{M} \frac{1}{M} \sum_{j=1}^{M} \big\| \nabla g^j(\bar{x}_t) - \nabla g^j(x_t^j) \big\|^2$$

$$\leq 6L_g^2 \sum_{m=1}^{M} \mathbb{E} \| x_t^m - \bar{x}_t \|^2 + 3M\delta_g^2, \tag{39}$$

where the last inequality is due to the above Assumption 6.

Similarly, we can obtain

$$\sum_{m=1}^{M} \mathbb{E} \big\| g^m(x_t^m) - \frac{1}{M} \sum_{j=1}^{M} g^j(x_t^j) \big\|^2 \leq 6C_g^2 \sum_{m=1}^{M} \mathbb{E} \| x_t^m - \bar{x}_t \|^2 + 3M\delta_g^2. \tag{40}$$

$\square$

**Lemma 9.** *Suppose the iterates $\{x_t^m\}_{t=1}^T$, for all $m \in [M]$ generated from Algorithm 1 satisfy:*

$$\sum_{m=1}^{M} \mathbb{E}\|x_t^m - \bar{x}_t\|^2 \leq (q-1) \sum_{l=s_t}^{t-1} \gamma^2 \eta_l^2 \sum_{m=1}^{M} \mathbb{E}\|d_l^m - \bar{d}_l\|^2, \tag{41}$$

*where $\bar{x}_t = \frac{1}{M} \sum_{m=1}^{M} x^m$, $d_t^m = \frac{x_t^m - x_{t+1}^m}{\gamma \eta_t}$ and $\bar{d}_t = \frac{\bar{x}_t - \bar{x}_{t+1}}{\eta_t \gamma}$.*

*Proof.* According to the lines 7 and 11 of Algorithm 1, we have

$$x_{t+1}^m = x_t^m - \gamma \eta_t A_t^{-1} w_t^m = \arg\min_{x \in \mathbb{R}^d} \left\{ \langle w_t^m, x \rangle + \frac{1}{2\eta_t \gamma} (x - x_t^m)^T A_t (x - x_t^m) \right\},$$

$$\bar{x}_{t+1} = \bar{x}_t - \gamma \eta_t A_t^{-1} \bar{w}_t = \arg\min_{x \in \mathbb{R}^d} \left\{ \langle \bar{w}_t, x \rangle + \frac{1}{2\eta_t \gamma} (x - \bar{x}_t)^T A_t (x - \bar{x}_t) \right\},$$

and then we define the gradient mappings as in the above (13): $d_t^m = \frac{x_t^m - x_{t+1}^m}{\gamma \eta_t} = A_t^{-1} w_t^m$ and $\bar{d}_t = \frac{\bar{x}_t - \bar{x}_{t+1}}{\eta_t \gamma} = A_t^{-1} \bar{w}_t = \frac{1}{M} \sum_{m=1}^{M} d_t^m$ for any $m \in [M]$ and $t \geq 1$.

From the line 7 of Algorithm 1, when $t = s_t = q\lfloor t/q \rfloor + 1$, we have $x_t^m = \bar{x}_t = \frac{1}{M} \sum_{m=1}^{M} x_t^m$ for any $m \in [M]$, so the about inequality in the lemma holds trivially.

When $t \in (s_t, s_t + q)$, we have

$$x_t^m = x_{s_t}^m - \sum_{l=s_t}^{t-1} \gamma \eta_l d_l^m, \quad \text{and} \quad \bar{x}_t = \bar{x}_{s_t} - \sum_{l=s_t}^{t-1} \gamma \eta_l \bar{d}_l.$$

Thus we have

$$\sum_{m=1}^{M} \mathbb{E}\|x_t^m - \bar{x}_t\|^2 = \sum_{m=1}^{M} \mathbb{E}\left\| x_{s_t}^m - \bar{x}_{s_t} - \left( \sum_{l=s_t}^{t-1} \gamma \eta_l d_l^m - \sum_{l=s_t}^{t-1} \gamma \eta_l \bar{d}_l \right) \right\|^2$$

$$= \sum_{m=1}^{M} \mathbb{E}\left\| \left( \sum_{l=s_t}^{t-1} \gamma \eta_l d_l^m - \sum_{l=s_t}^{t-1} \gamma \eta_l \bar{d}_l \right) \right\|^2 \leq (q-1) \sum_{l=s_t}^{t-1} \gamma^2 \eta_l^2 \sum_{m=1}^{M} \mathbb{E}\|d_l^m - \bar{d}_l\|^2,$$

where the above inequality is due to $t - s_t \leq q - 1$. $\qquad\square$

**Lemma 10.** *Let $C_{fg}^2 = \max(C_f^2, C_g^2)$, $L_{fg}^2 = L_f^2 C_g^2 + L_g^2$ and $\eta_t \leq \frac{\rho}{24\gamma q L_{fg} C_{fg}}$ for all $t \geq 0$. Further let $\alpha_{t+1} = c_1 \eta_t^2$, $\beta_{t+1} = c_2 \eta_t^2$ and $\varrho_{t+1} = c_3 \eta_t^2$, $c_1, c_2, c_3 > 0$ and $c_1^2 + c_2^2 \leq \frac{(24)^4 q^2 \gamma^4 L_{fg}^4 C_{fg}^4}{9\rho^4}$. Set $s_t = q\lfloor t/q \rfloor + 1$ and $t \in [s_t, s_t + q - 1]$, we have*

$$\sum_{t=s_t}^{s_t+q-1} \eta_t \sum_{m=1}^{M} \mathbb{E}\|d_t^m - \bar{d}_t\|^2$$

$$\leq \frac{6M}{5} \sum_{t=s_t}^{s_t+q-1} \eta_t \mathbb{E}\|\bar{d}_t\|^2 + \frac{\rho^2(c_1^2 + c_3^2)}{120q^2\gamma^4 C_{fg}^2 L_{fg}^2 C_g^2} \sum_{t=s_t}^{s_t+q-1} \eta_t \sum_{m=1}^{M} \mathbb{E}\|h_t^m - g^m(\bar{x}_t)\|^2 + \frac{3M\hat{\delta}^2}{5\gamma^2 L_{fg}^2} \sum_{t=s_t}^{s_t+q-1} \eta_t^3, \tag{42}$$

*where $\hat{\delta}^2 = 2c_1^2 L_f^2 \sigma^2 + c_3^2 \sigma^2 + 4c_3^2 \delta_f^2 + 4c_3^2 L_f^2 \delta_g^2 + c_2^2 \sigma^2 + 3c_2^2 \delta_g^2$.*

*Proof.* According to the lines 7 and 11 of Algorithm 1, we have

$$x_{t+1}^m = \arg\min_{x \in \mathbb{R}^d} \left\{ \langle w_t^m, x \rangle + \frac{1}{2\eta_t \gamma} (x - x_t^m)^T A_t (x - x_t^m) \right\},$$

$$\bar{x}_{t+1} = \arg\min_{x \in \mathbb{R}^d} \left\{ \langle \bar{w}_t, x \rangle + \frac{1}{2\eta_t \gamma} (x - \bar{x}_t)^T A_t (x - \bar{x}_t) \right\},$$

and then we define the gradient mappings as in the above (13): $d_t^m = \frac{x_t^m - x_{t+1}^m}{\gamma \eta_t} = A_t^{-1} w_t^m$ and $\bar{d}_t = \frac{\bar{x}_t - \bar{x}_{t+1}}{\eta_t \gamma} = A_t^{-1} \bar{w}_t = \frac{1}{M} \sum_{m=1}^{M} d_t^m$ for any $m \in [M]$ and $t \geq 1$. Then we have

$$\sum_{m=1}^{M} \mathbb{E}\|d_t^m - \bar{d}_t\|^2 = \sum_{m=1}^{M} \mathbb{E}\|A_t^{-1}(w_t^m - \bar{w}_t)\|^2 \leq \frac{1}{\rho^2} \sum_{m=1}^{M} \mathbb{E}\|w_t^m - \bar{w}_t\|^2 \tag{43}$$

$$= \frac{1}{\rho^2} \sum_{m=1}^{M} \mathbb{E}\|(u_t^m)^T - (u_t^m)^T \bar{v}_t + (u_t^m)^T \bar{v}_t - (\bar{u}_t)^T \bar{v}_t + (\bar{u}_t)^T \bar{v}_t - \frac{1}{M} \sum_{m=1}^{M} (u_t^m)^T v_t^m\|^2$$

$$\leq \frac{1}{\rho^2} \sum_{m=1}^{M} \left( 3C_g^2 \mathbb{E}\|v_t^m - \bar{v}_t\|^2 + 3C_f^2 \mathbb{E}\|u_t^m - \bar{u}_t\|^2 + 3\|(\bar{u}_t)^T \bar{v}_t - \frac{1}{M} \sum_{m=1}^{M} (u_t^m)^T v_t^m\|^2 \right),$$

where the first inequality holds by Assumption 5, i.e., $A_t \succeq \rho I_d$ for all $t \geq 1$, and the last inequality holds by $\|u_t^m\|^2 \leq C_g^2$ and $\|\bar{v}_t\|^2 \leq C_f^2$. Consider the term $\|(\bar{u}_t)^T \bar{v}_t - \frac{1}{M} \sum_{m=1}^{M} (u_t^m)^T v_t^m\|^2$, we have

$$\|(\bar{u}_t)^T \bar{v}_t - \frac{1}{M} \sum_{m=1}^{M} (u_t^m)^T v_t^m\|^2 = \|(\bar{u}_t)^T \bar{v}_t - \frac{1}{M} \sum_{m=1}^{M} (u_t^m)^T v_t^m\|^2$$

$$\leq \frac{1}{M} \sum_{m=1}^{M} \|(\bar{u}_t)^T \bar{v}_t - (u_t^m)^T v_t^m\|^2$$

$$= \frac{1}{M} \sum_{m=1}^{M} \|(\bar{u}_t)^T \bar{v}_t - (\bar{u}_t)^T v_t^m + (\bar{u}_t)^T v_t^m - (u_t^m)^T v_t^m\|^2$$

$$\leq \frac{1}{M} \sum_{m=1}^{M} \left( 2C_g^2 \|v_t^m - \bar{v}_t\|^2 + 2C_f^2 \|u_t^m - \bar{u}_t\|^2 \right). \tag{44}$$

By combining the above inequalities (43) and (44), we have

$$\sum_{m=1}^{M} \mathbb{E}\|d_t^m - \bar{d}_t\|^2 \leq \frac{9C_g^2}{\rho^2} \sum_{m=1}^{M} \mathbb{E}\|v_t^m - \bar{v}_t\|^2 + \frac{9C_f^2}{\rho^2} \sum_{m=1}^{M} \mathbb{E}\|u_t^m - \bar{u}_t\|^2. \tag{45}$$

Let $t = s_t = q\lfloor t/q \rfloor + 1$. When $t = s_t$, we have $v_t^m = \bar{v}_t$ and $u_t^m = \bar{u}_t$ for any $m \in [M]$, so we have $\sum_{m=1}^{M} \mathbb{E}\|v_t^m - \bar{v}_t\|^2 = 0$ and $\sum_{m=1}^{M} \mathbb{E}\|u_t^m - \bar{u}_t\|^2 = 0$. According to the above inequality (45), when $t = s_t$, we have $\sum_{m=1}^{M} \mathbb{E}\|d_t^m - \bar{d}_t\|^2 = 0$. Clearly, the about inequality (42) in the lemma holds trivially.

When $t \in (s_t, s_t + q)$, we first consider the term $\sum_{m=1}^{M} \mathbb{E}\|v_t^m - \bar{v}_t\|^2$ as follows:

$$\sum_{m=1}^{M} \mathbb{E}\|v_t^m - \bar{v}_t\|^2 \tag{46}$$

$$= \sum_{m=1}^{M} \mathbb{E}\left\|v_t^m - \frac{1}{M}\sum_{m=1}^{M} v_t^m\right\|^2$$

$$= \sum_{m=1}^{M} \mathbb{E}\left\|\Pi_{C_f}\left[\nabla f^m(h_t^m; \xi_t^m) + (1 - \varrho_t)\left(v_{t-1}^m - \nabla f^m(h_{t-1}^m; \xi_t^m)\right)\right] - \frac{1}{M}\sum_{m=1}^{M}\Pi_{C_f}\left[\nabla f^m(h_t^m; \xi_t^m)\right.\right.$$

$$\left.\left. + (1 - \varrho_t)\left(v_{t-1}^m - \nabla f^m(h_{t-1}^m; \xi_t^m)\right)\right]\right\|^2$$

$$\leq \sum_{m=1}^{M} \mathbb{E}\left\|\nabla f^m(h_t^m; \xi_t^m) + (1 - \varrho_t)\left(v_{t-1}^m - \nabla f^m(h_{t-1}^m; \xi_t^m)\right) - \frac{1}{M}\sum_{m=1}^{M}\left(\nabla f^m(h_t^m; \xi_t^m)\right.\right.$$

$$\left.\left. + (1 - \varrho_t)\left(v_{t-1}^m - \nabla f^m(h_{t-1}^m; \xi_t^m)\right)\right)\right\|^2$$

$$\leq (1 + \nu)(1 - \varrho_t)^2 \sum_{m=1}^{M} \mathbb{E}\|v_{t-1}^m - \bar{v}_{t-1}\|^2 + (1 + \frac{1}{\nu})\sum_{m=1}^{M} \mathbb{E}\left\|\nabla f^m(h_t^m; \xi_t^m)\right.$$

$$\left. - \frac{1}{M}\sum_{m=1}^{M}\nabla f^m(h_t^m; \xi_t^m) - (1 - \varrho_t)\left(\nabla f^m(h_{t-1}^m; \xi_t^m) - \frac{1}{M}\sum_{m=1}^{M}\nabla f^m(h_{t-1}^m; \xi_t^m)\right)\right\|^2.$$

Then, we consider the last term of (46):

$$\sum_{m=1}^{M} \mathbb{E}\left\|\nabla f^m(h_t^m; \xi_t^m) - \frac{1}{M}\sum_{m=1}^{M}\nabla f^m(h_t^m; \xi_t^m) - (1 - \varrho_t)\left(\nabla f^m(h_{t-1}^m; \xi_t^m) - \frac{1}{M}\sum_{m=1}^{M}\nabla f^m(h_{t-1}^m; \xi_t^m)\right)\right\|^2$$

$$= \sum_{m=1}^{M} \mathbb{E}\left\|\nabla f^m(h_t^m; \xi_t^m) - \nabla f^m(h_{t-1}^m; \xi_t^m) - \frac{1}{M}\sum_{m=1}^{M}\left(\nabla f^m(h_t^m; \xi_t^m) - \nabla f^m(h_{t-1}^m; \xi_t^m)\right)\right.$$

$$\left. + \varrho_t\left(\nabla f^m(h_{t-1}^m; \xi_t^m) - \frac{1}{M}\sum_{m=1}^{M}\nabla f^m(h_{t-1}^m; \xi_t^m)\right)\right\|^2$$

$$\leq 2\sum_{m=1}^{M} \mathbb{E}\|\nabla f^m(h_t^m; \xi_t^m) - \nabla f^m(h_{t-1}^m; \xi_t^m)\|^2 + 2\varrho_t^2\sum_{m=1}^{M} \mathbb{E}\left\|\nabla f^m(h_{t-1}^m; \xi_t^m) - \frac{1}{M}\sum_{m=1}^{M}\nabla f^m(h_{t-1}^m; \xi_t^m)\right\|^2$$

$$\leq 2L_f^2\sum_{m=1}^{M} \mathbb{E}\|h_t^m - h_{t-1}^m\|^2 + 2\varrho_t^2\sum_{m=1}^{M} \mathbb{E}\left\|\nabla f^m(h_{t-1}^m; \xi_t^m) - \frac{1}{M}\sum_{m=1}^{M}\nabla f^m(h_{t-1}^m; \xi_t^m)\right\|^2, \tag{47}$$

where the second last inequality is due to Young inequality and the above Lemma 2.

Consider the term $\sum_{m=1}^{M} \left\| \nabla f^m(h_{t-1}^m; \xi_t^m) - \frac{1}{M} \sum_{m=1}^{M} \nabla f^m(h_{t-1}^m; \xi_t^m) \right\|^2$, we have

$$\sum_{m=1}^{M} \left\| \nabla f^m(h_{t-1}^m; \xi_t^m) - \frac{1}{M} \sum_{m=1}^{M} \nabla f^m(h_{t-1}^m; \xi_t^m) \right\|^2$$

$$= \sum_{m=1}^{M} \left\| \nabla f^m(h_{t-1}^m; \xi_t^m) - \nabla f^m(h_{t-1}^m) - \frac{1}{M} \sum_{m=1}^{M} \left( \nabla f^m(h_{t-1}^m; \xi_t^m) - \nabla f^m(h_{t-1}^m) \right) \right.$$

$$\left. + \nabla f^m(h_{t-1}^m) - \frac{1}{M} \sum_{m=1}^{M} \nabla f^m(h_{t-1}^m) \right\|^2$$

$$\leq 2 \sum_{m=1}^{M} \left\| \nabla f^m(h_{t-1}^m; \xi_t^m) - \nabla f^m(h_{t-1}^m) - \frac{1}{M} \sum_{m=1}^{M} \left( \nabla f^m(h_{t-1}^m; \xi_t^m) - \nabla f^m(h_{t-1}^m) \right) \right\|$$

$$+ 2 \sum_{m=1}^{M} \left\| \nabla f^m(h_{t-1}^m) - \frac{1}{M} \sum_{m=1}^{M} \nabla f^m(h_{t-1}^m) \right\|^2$$

$$\leq 2 \sum_{m=1}^{M} \left\| \nabla f^m(h_{t-1}^m; \xi_t^m) - \nabla f^m(h_{t-1}^m) \right\|^2 + 2 \sum_{m=1}^{M} \left\| \nabla f^m(h_{t-1}^m) - \frac{1}{M} \sum_{m=1}^{M} \nabla f^m(h_{t-1}^m) \right\|^2$$

$$\leq 2M\sigma^2 + 16L_f^2 \sum_{m=1}^{M} \mathbb{E}\|h_{t-1}^m - g^m(\bar{x}_{t-1})\|^2 + 8M\delta_f^2 + 8ML_f^2\delta_g^2, \tag{48}$$

where the last inequality holds by the above Lemma 8.

Since $h_t^m = g^m(x_t^m; \zeta_t^m) + (1 - \alpha_t)\left( h_{t-1}^m - g^m(x_{t-1}^m; \zeta_t^m) \right)$, we have

$$\mathbb{E}\|h_t^m - h_{t-1}^m\|^2$$

$$= \mathbb{E}\|g^m(x_t^m; \zeta_t^m) - g^m(x_{t-1}^m; \zeta_t^m) - \alpha_t\left( h_{t-1}^m - g^m(x_{t-1}^m; \zeta_t^m) \right)\|^2$$

$$\leq 2\mathbb{E}\|g^m(x_t^m; \zeta_t^m) - g^m(x_{t-1}^m; \zeta_t^m)\|^2 + 2\alpha_t^2 \mathbb{E}\|h_{t-1}^m - g^m(x_{t-1}^m; \zeta_t^m)\|^2$$

$$\leq 2C_g^2 \|x_t^m - x_{t-1}^m\|^2 + 2\alpha_t^2 \mathbb{E}\|h_{t-1}^m - g^m(x_{t-1}^m; \zeta_t^m)\|^2$$

$$= 2C_g^2 \|x_t^m - x_{t-1}^m\|^2 + 2\alpha_t^2 \mathbb{E}\|h_{t-1}^m - g^m(x_{t-1}^m; \zeta_t^m) + g^m(x_{t-1}^m) - g^m(x_{t-1}^m)\|^2$$

$$\leq 2C_g^2 \|x_t^m - x_{t-1}^m\|^2 + 4\alpha_t^2 \mathbb{E}\|h_{t-1}^m - g^m(x_{t-1}^m)\|^2 + 4\alpha_t^2 \mathbb{E}\|g^m(x_{t-1}^m; \zeta_t^m) + g^m(x_{t-1}^m)\|^2$$

$$\leq 2C_g^2 \|x_t^m - x_{t-1}^m\|^2 + 4\alpha_t^2 \mathbb{E}\|h_{t-1}^m - g^m(x_{t-1}^m)\|^2 + 4\alpha_t^2 \sigma^2$$

$$= 2C_g^2 \|x_t^m - x_{t-1}^m\|^2 + 4\alpha_t^2 \mathbb{E}\|h_{t-1}^m - g^m(\bar{x}_{t-1}) + g^m(\bar{x}_{t-1}) - g^m(x_{t-1}^m)\|^2 + 4\alpha_t^2 \sigma^2$$

$$\leq 2C_g^2 \|x_t^m - x_{t-1}^m\|^2 + 8\alpha_t^2 \mathbb{E}\|h_{t-1}^m - g^m(\bar{x}_{t-1})\|^2 + 8\alpha_t^2 C_g^2 \|\bar{x}_{t-1} - x_{t-1}^m\|^2 + 4\alpha_t^2 \sigma^2, \tag{49}$$

where the second inequality holds by Assumption 3.

By combining the above inequalities (46), (47), (48) and (49), we have

$$
\sum_{m=1}^{M} \mathbb{E}\|v_t^m - \bar{v}_t\|^2 \tag{50}
$$

$$
\leq (1+\nu)(1-\varrho_t)^2 \sum_{m=1}^{M} \mathbb{E}\|v_{t-1}^m - \bar{v}_{t-1}\|^2 + (1+\frac{1}{\nu}) \sum_{m=1}^{M} \mathbb{E}\big\|\nabla f^m(h_t^m; \xi_t^m)
$$

$$
- \frac{1}{M} \sum_{m=1}^{M} \nabla f^m(h_t^m; \xi_t^m) - (1-\varrho_t)\big(\nabla f^m(h_{t-1}^m; \xi_t^m) - \frac{1}{M} \sum_{m=1}^{M} \nabla f^m(h_{t-1}^m; \xi_t^m)\big)\big\|^2
$$

$$
\leq (1+\nu)(1-\varrho_t)^2 \sum_{m=1}^{M} \mathbb{E}\|v_{t-1}^m - \bar{v}_{t-1}\|^2 + (1+\frac{1}{\nu})\Big(4L_f^2 C_g^2 \sum_{m=1}^{M} \mathbb{E}\|x_t^m - x_{t-1}^m\|^2
$$

$$
+ 16\alpha_t^2 L_f^2 \sum_{m=1}^{M} \mathbb{E}\|h_{t-1}^m - g^m(\bar{x}_{t-1})\|^2 + 16\alpha_t^2 L_f^2 C_g^2 \sum_{m=1}^{M} \mathbb{E}\|\bar{x}_{t-1} - x_{t-1}^m\|^2 + 8ML_f^2 \alpha_t^2 \sigma^2
$$

$$
+ 4M\varrho_t^2 \sigma^2 + 32\varrho_t^2 L_f^2 \sum_{m=1}^{M} \mathbb{E}\|h_{t-1}^m - g^m(\bar{x}_{t-1})\|^2 + 16M\varrho_t^2 \delta_f^2 + 16ML_f^2 \varrho_t^2 \delta_g^2\Big)
$$

$$
\leq (1+\nu)(1-\varrho_t)^2 \sum_{m=1}^{M} \mathbb{E}\|v_{t-1}^m - \bar{v}_{t-1}\|^2 + (1+\frac{1}{\nu})\Big(8L_f^2 C_g^2 \eta_{t-1}^2 \gamma^2 \sum_{m=1}^{M} \mathbb{E}\|d_{t-1}^m - \bar{d}_{t-1}\|^2
$$

$$
+ 8L_f^2 C_g^2 \eta_{t-1}^2 \gamma^2 \sum_{m=1}^{M} \mathbb{E}\|\bar{d}_{t-1}\|^2 + 16(q-1)L_f^2 C_g^2 \alpha_t^2 \sum_{l=s_t}^{t-1} \gamma^2 \eta_l^2 \sum_{m=1}^{M} \mathbb{E}\|d_l^m - \bar{d}_l\|^2 + 8ML_f^2 \alpha_t^2 \sigma^2
$$

$$
+ 4M\varrho_t^2 \sigma^2 + 32(\varrho_t^2 + \alpha_t^2)L_f^2 \sum_{m=1}^{M} \mathbb{E}\|h_{t-1}^m - g^m(\bar{x}_{t-1})\|^2 + 16M\varrho_t^2 \delta_f^2 + 16ML_f^2 \varrho_t^2 \delta_g^2\Big)
$$

$$
\leq (1+\nu)(1-\varrho_t)^2 \sum_{m=1}^{M} \mathbb{E}\|v_{t-1}^m - \bar{v}_{t-1}\|^2 + (1+\frac{1}{\nu})\Big(\frac{72L_f^2 C_g^4 \eta_{t-1}^2 \gamma^2}{\rho^2} \sum_{m=1}^{M} \mathbb{E}\|v_{t-1}^m - \bar{v}_{t-1}\|^2
$$

$$
+ \frac{72C_f^2 L_f^2 C_g^2 \eta_{t-1}^2 \gamma^2}{\rho^2} \sum_{m=1}^{M} \mathbb{E}\|u_{t-1}^m - \bar{u}_{t-1}\|^2 + 8L_f^2 C_g^2 \eta_{t-1}^2 \gamma^2 \sum_{m=1}^{M} \mathbb{E}\|\bar{d}_{t-1}\|^2
$$

$$
+ 16(q-1)L_f^2 C_g^2 \alpha_t^2 \sum_{l=s_t}^{t-1} \gamma^2 \eta_l^2 \Big(\frac{9C_g^2}{\rho^2} \sum_{m=1}^{M} \mathbb{E}\|v_l^m - \bar{v}_l\|^2 + \frac{9C_f^2}{\rho^2} \sum_{m=1}^{M} \mathbb{E}\|u_l^m - \bar{u}_l\|^2\Big)
$$

$$
+ 32(\varrho_t^2 + \alpha_t^2)L_f^2 \sum_{m=1}^{M} \mathbb{E}\|h_{t-1}^m - g^m(\bar{x}_{t-1})\|^2 + 8ML_f^2 \alpha_t^2 \sigma^2 + 4M\varrho_t^2 \sigma^2 + 16M\varrho_t^2 \delta_f^2 + 16ML_f^2 \varrho_t^2 \delta_g^2\Big),
$$

$$
\tag{51}
$$

where the second last inequality holds by the above Lemma 9 and the above inequality (45), and the last inequality holds by $d_{t-1}^m = \frac{x_t^m - x_{t-1}^m}{\eta_t \gamma}$, and the above inequality (45).

Next, we consider the term $\sum_{m=1}^M \mathbb{E}\|u_t^m - \bar{u}_t\|^2$ as follows:

$$\sum_{m=1}^M \mathbb{E}\|u_t^m - \bar{u}_t\|^2 \tag{52}$$

$$= \sum_{m=1}^M \mathbb{E}\left\|u_t^m - \frac{1}{M}\sum_{m=1}^M u_t^m\right\|^2$$

$$= \sum_{m=1}^M \mathbb{E}\left\|\Pi_{C_g}\left[\nabla g^m(x_t^m;\zeta_t^m) + (1-\beta_t)\left(u_{t-1}^m - \nabla g^m(x_{t-1}^m;\zeta_t^m)\right)\right] - \frac{1}{M}\sum_{m=1}^M \Pi_{C_g}\left[\nabla g^m(x_t^m;\zeta_t^m)\right.\right.$$

$$\left.\left. + (1-\beta_t)\left(u_{t-1}^m - \nabla g^m(x_{t-1}^m;\zeta_t^m)\right)\right]\right\|^2$$

$$\leq \sum_{m=1}^M \mathbb{E}\left\|\nabla g^m(x_t^m;\zeta_t^m) + (1-\beta_t)\left(u_{t-1}^m - \nabla g^m(x_{t-1}^m;\zeta_t^m)\right) - \frac{1}{M}\sum_{m=1}^M \left(\nabla g^m(x_t^m;\zeta_t^m)\right.\right.$$

$$\left.\left. + (1-\beta_t)\left(u_{t-1}^m - \nabla g^m(x_{t-1}^m;\zeta_t^m)\right)\right)\right\|^2$$

$$\leq (1+\nu)(1-\beta_t)^2 \sum_{m=1}^M \mathbb{E}\|u_{t-1}^m - \bar{u}_{t-1})\|^2 + (1+\frac{1}{\nu})\sum_{m=1}^M \mathbb{E}\left\|\nabla g^m(x_t^m;\zeta_t^m)\right.$$

$$\left. - \frac{1}{M}\sum_{m=1}^M \nabla g^m(x_t^m;\zeta_t^m) - (1-\beta_t)\left(\nabla g^m(x_{t-1}^m;\zeta_t^m) - \frac{1}{M}\sum_{m=1}^M \nabla g^m(x_{t-1}^m;\zeta_t^m)\right)\right\|^2. \tag{53}$$

Then, we consider the last term of (52):

$$\sum_{m=1}^M \mathbb{E}\left\|\nabla g^m(x_t^m;\zeta_t^m) - \frac{1}{M}\sum_{m=1}^M \nabla g^m(x_t^m;\zeta_t^m) - (1-\beta_t)\left(\nabla g^m(x_{t-1}^m;\zeta_t^m) - \frac{1}{M}\sum_{m=1}^M \nabla g^m(x_{t-1}^m;\zeta_t^m)\right)\right\|^2$$

$$= \sum_{m=1}^M \mathbb{E}\left\|\nabla g^m(x_t^m;\zeta_t^m) - \nabla g^m(x_{t-1}^m;\zeta_t^m) - \frac{1}{M}\sum_{m=1}^M \left(\nabla g^m(x_t^m;\zeta_t^m) - \nabla g^m(x_{t-1}^m;\zeta_t^m)\right)\right.$$

$$\left. + \beta_t\left(\nabla g^m(x_{t-1}^m;\zeta_t^m) - \frac{1}{M}\sum_{m=1}^M \nabla g^m(x_{t-1}^m;\zeta_t^m)\right)\right\|^2$$

$$\leq 2\sum_{m=1}^M \mathbb{E}\|\nabla g^m(x_t^m;\zeta_t^m) - \nabla g^m(x_{t-1}^m;\zeta_t^m)\|^2 + 2\beta_t^2 \sum_{m=1}^M \mathbb{E}\left\|\nabla g^m(x_{t-1}^m;\zeta_t^m) - \frac{1}{M}\sum_{m=1}^M \nabla g^m(x_{t-1}^m;\zeta_t^m)\right\|^2$$

$$\leq 2L_g^2 \sum_{m=1}^M \mathbb{E}\|x_t^m - x_{t-1}^m\|^2 + 2\beta_t^2 \sum_{m=1}^M \mathbb{E}\left\|\nabla g^m(x_{t-1}^m;\zeta_t^m) - \frac{1}{M}\sum_{m=1}^M \nabla g^m(x_{t-1}^m;\zeta_t^m)\right\|^2, \tag{54}$$

where the second last inequality is due to Young inequality and the above Lemma 2.

Consider the term $\sum_{m=1}^{M} \left\| \nabla g^m(x_{t-1}^m; \zeta_t^m) - \frac{1}{M} \sum_{m=1}^{M} \nabla g^m(x_{t-1}^m; \zeta_t^m) \right\|^2$, we have

$$\sum_{m=1}^{M} \left\| \nabla g^m(x_{t-1}^m; \zeta_t^m) - \frac{1}{M} \sum_{m=1}^{M} \nabla g^m(x_{t-1}^m; \zeta_t^m) \right\|^2$$

$$= \sum_{m=1}^{M} \left\| \nabla g^m(x_{t-1}^m; \zeta_t^m) - \nabla g^m(x_{t-1}^m) - \frac{1}{M} \sum_{m=1}^{M} \left( \nabla g^m(x_{t-1}^m; \zeta_t^m) - \nabla g^m(x_{t-1}^m) \right) \right.$$

$$\left. + \nabla g^m(x_{t-1}^m) - \frac{1}{M} \sum_{m=1}^{M} \nabla g^m(x_{t-1}^m) \right\|^2$$

$$\leq 2 \sum_{m=1}^{M} \left\| \nabla g^m(x_{t-1}^m; \zeta_t^m) - \nabla g^m(x_{t-1}^m) - \frac{1}{M} \sum_{m=1}^{M} \left( \nabla g^m(x_{t-1}^m; \zeta_t^m) - \nabla g^m(x_{t-1}^m) \right) \right\|$$

$$+ 2 \sum_{m=1}^{M} \left\| \nabla g^m(x_{t-1}^m) - \frac{1}{M} \sum_{m=1}^{M} \nabla g^m(x_{t-1}^m) \right\|^2$$

$$\leq 2 \sum_{m=1}^{M} \left\| \nabla g^m(x_{t-1}^m; \zeta_t^m) - \nabla g^m(x_{t-1}^m) \right\| + 2 \sum_{m=1}^{M} \left\| \nabla g^m(x_{t-1}^m) - \frac{1}{M} \sum_{m=1}^{M} \nabla g^m(x_{t-1}^m) \right\|^2$$

$$\leq 2M\sigma^2 + 12L_g^2 \sum_{m=1}^{M} \mathbb{E}\|x_{t-1}^m - \bar{x}_{t-1}\|^2 + 6M\delta_g^2, \tag{55}$$

where the last inequality holds by the above Lemma 8.

By combining the above inequalities (52), (54) and (55), we have

$$\sum_{m=1}^{M} \mathbb{E}\|u_t^m - \bar{u}_t\|^2 \tag{56}$$

$$\leq (1+\nu)(1-\beta_t)^2 \sum_{m=1}^{M} \mathbb{E}\|u_{t-1}^m - \bar{u}_{t-1})\|^2 + (1+\frac{1}{\nu}) \sum_{m=1}^{M} \mathbb{E}\big\| \nabla g^m(x_t^m; \zeta_t^m)$$

$$- \frac{1}{M} \sum_{m=1}^{M} \nabla g^m(x_t^m; \zeta_t^m) - (1-\beta_t)\big( \nabla g^m(x_{t-1}^m; \zeta_t^m) - \frac{1}{M} \sum_{m=1}^{M} \nabla g^m(x_{t-1}^m; \zeta_t^m) \big) \big\|^2$$

$$\leq (1+\nu)(1-\beta_t)^2 \sum_{m=1}^{M} \mathbb{E}\|u_{t-1}^m - \bar{u}_{t-1})\|^2 + (1+\frac{1}{\nu})\Big( 2L_g^2 \sum_{m=1}^{M} \mathbb{E}\|x_t^m - x_{t-1}^m\|^2$$

$$+ 4M\sigma^2\beta_t^2 + 24L_g^2\beta_t^2 \sum_{m=1}^{M} \mathbb{E}\|x_{t-1}^m - \bar{x}_{t-1}\|^2 + 12M\delta_g^2\beta_t^2 \Big)$$

$$\leq (1+\nu)(1-\beta_t)^2 \sum_{m=1}^{M} \mathbb{E}\|u_{t-1}^m - \bar{u}_{t-1})\|^2 + (1+\frac{1}{\nu})\Big( 4L_g^2\eta_{t-1}^2\gamma^2 \sum_{m=1}^{M} \mathbb{E}\|d_{t-1}^m - \bar{d}_{t-1}\|^2 + 4L_g^2\eta_{t-1}^2\gamma^2 \sum_{m=1}^{M} \mathbb{E}\|\bar{d}_{t-1}\|^2$$

$$+ 4M\sigma^2\beta_t^2 + 24(q-1)L_g^2\beta_t^2 \sum_{l=s_t}^{t-2} \gamma^2\eta_l^2 \sum_{m=1}^{M} \mathbb{E}\|d_l^m - \bar{d}_l\|^2 + 12M\delta_g^2\beta_t^2 \Big)$$

$$\leq (1+\nu)(1-\beta_t)^2 \sum_{m=1}^{M} \mathbb{E}\|u_{t-1}^m - \bar{u}_{t-1})\|^2 + (1+\frac{1}{\nu})\Big( \frac{36C_g^2 L_g^2\eta_{t-1}^2\gamma^2}{\rho^2} \sum_{m=1}^{M} \mathbb{E}\|v_{t-1}^m - \bar{v}_{t-1}\|^2$$

$$+ \frac{36C_f^2 L_g^2\eta_{t-1}^2\gamma^2}{\rho^2} \sum_{m=1}^{M} \mathbb{E}\|u_{t-1}^m - \bar{u}_{t-1}\|^2 + 4L_g^2\eta_{t-1}^2\gamma^2 \sum_{m=1}^{M} \mathbb{E}\|\bar{d}_{t-1}\|^2 + 4M\sigma^2\beta_t^2 + 12M\delta_g^2\beta_t^2$$

$$+ 24(q-1)L_g^2\beta_t^2 \sum_{l=s_t}^{t-2} \gamma^2\eta_l^2 \Big( \frac{9C_g^2}{\rho^2} \sum_{m=1}^{M} \mathbb{E}\|v_l^m - \bar{v}_l\|^2 + \frac{9C_f^2}{\rho^2} \sum_{m=1}^{M} \mathbb{E}\|u_l^m - \bar{u}_l\|^2 \Big) \Big), \tag{57}$$

where the last inequality holds by the above inequality (45).

By summing the above inequalities (50) and (56), we have

$$\sum_{m=1}^{M} \left( \mathbb{E}\|u_t^m - \bar{u}_t\|^2 + \mathbb{E}\|v_t^m - \bar{v}_t\|^2 \right) \tag{58}$$

$$\leq (1+\nu)(1-\beta_t)^2 \sum_{m=1}^{M} \mathbb{E}\|u_{t-1}^m - \bar{u}_{t-1})\|^2 + (1+\frac{1}{\nu})\left( \frac{36C_g^2 L_g^2 \eta_{t-1}^2 \gamma^2}{\rho^2} \sum_{m=1}^{M} \mathbb{E}\|v_{t-1}^m - \bar{v}_{t-1}\|^2 \right.$$

$$+ \frac{36C_f^2 L_g^2 \eta_{t-1}^2 \gamma^2}{\rho^2} \sum_{m=1}^{M} \mathbb{E}\|u_{t-1}^m - \bar{u}_{t-1}\|^2 + 4L_g^2 \eta_{t-1}^2 \gamma^2 \sum_{m=1}^{M} \mathbb{E}\|\bar{d}_{t-1}\|^2 + 4M\sigma^2\beta_t^2 + 12M\delta_g^2\beta_t^2$$

$$+ 24(q-1)L_g^2\beta_t^2 \sum_{l=s_t}^{t-2} \gamma^2\eta_l^2 \Big( \frac{9C_g^2}{\rho^2} \sum_{m=1}^{M} \mathbb{E}\|v_l^m - \bar{v}_l\|^2 + \frac{9C_f^2}{\rho^2} \sum_{m=1}^{M} \mathbb{E}\|u_l^m - \bar{u}_l\|^2 \Big) \Big)$$

$$+ (1+\nu)(1-\varrho_t)^2 \sum_{m=1}^{M} \mathbb{E}\|v_{t-1}^m - \bar{v}_{t-1})\|^2 + (1+\frac{1}{\nu})\left( \frac{72L_f^2 C_g^4 \eta_{t-1}^2 \gamma^2}{\rho^2} \sum_{m=1}^{M} \mathbb{E}\|v_{t-1}^m - \bar{v}_{t-1}\|^2 \right.$$

$$+ \frac{72C_f^2 L_f^2 C_g^2 \eta_{t-1}^2 \gamma^2}{\rho^2} \sum_{m=1}^{M} \mathbb{E}\|u_{t-1}^m - \bar{u}_{t-1}\|^2 + 8L_f^2 C_g^2 \eta_{t-1}^2 \gamma^2 \sum_{m=1}^{M} \mathbb{E}\|\bar{d}_{t-1}\|^2$$

$$+ 16(q-1)L_f^2 C_g^2 \alpha_t^2 \sum_{l=s_t}^{t-1} \gamma^2\eta_l^2 \Big( \frac{9C_g^2}{\rho^2} \sum_{m=1}^{M} \mathbb{E}\|v_l^m - \bar{v}_l\|^2 + \frac{9C_f^2}{\rho^2} \sum_{m=1}^{M} \mathbb{E}\|u_l^m - \bar{u}_l\|^2 \Big)$$

$$+ 32(\varrho_t^2 + \alpha_t^2)L_f^2 \sum_{m=1}^{M} \mathbb{E}\|h_{t-1}^m - g^m(\bar{x}_{t-1})\|^2 + 8ML_f^2\alpha_t^2\sigma^2 + 4M\varrho_t^2\sigma^2 + 16M\varrho_t^2\delta_f^2 + 16ML_f^2\varrho_t^2\delta_g^2 \Big)$$

$$\leq \max\left( (1+\nu)(1-\beta_t)^2 + (1+\frac{1}{\nu})\frac{72C_f^2(L_f^2 C_g^2 + L_g^2)\eta_{t-1}^2\gamma^2}{\rho^2}, (1+\nu)(1-\varrho_t)^2 + (1+\frac{1}{\nu})\frac{72C_g^2(C_g^2 L_f^2 + L_g^2)\eta_{t-1}^2\gamma^2}{\rho^2} \right)$$

$$\cdot \sum_{m=1}^{M} \left( \mathbb{E}\|u_{t-1}^m - \bar{u}_{t-1}\|^2 + \mathbb{E}\|v_{t-1}^m - \bar{v}_{t-1}\|^2 \right) + 8(1+\frac{1}{\nu})(L_f^2 C_g^2 + L_g^2)\eta_{t-1}^2\gamma^2 \sum_{m=1}^{M} \mathbb{E}\|\bar{d}_{t-1}\|^2$$

$$+ 24(1+\frac{1}{\nu})(q-1)\left(L_f^2 C_g^2\alpha_t^2 + L_g^2\beta_t^2\right) \sum_{l=s_t}^{t-1} \gamma^2\eta_l^2 \Big( \frac{9C_g^2}{\rho^2} \sum_{m=1}^{M} \mathbb{E}\|v_l^m - \bar{v}_l\|^2 + \frac{9C_f^2}{\rho^2} \sum_{m=1}^{M} \mathbb{E}\|u_l^m - \bar{u}_l\|^2 \Big)$$

$$+ 32(1+\frac{1}{\nu})(\varrho_t^2 + \alpha_t^2)L_f^2 \sum_{m=1}^{M} \mathbb{E}\|h_{t-1}^m - g^m(\bar{x}_{t-1})\|^2 + (1+\frac{1}{\nu})\Big( 8ML_f^2\alpha_t^2\sigma^2 + 4M\varrho_t^2\sigma^2 + 16M\varrho_t^2\delta_f^2$$

$$+ 16ML_f^2\varrho_t^2\delta_g^2 + 4M\sigma^2\beta_t^2 + 12M\delta_g^2\beta_t^2 \Big). \tag{59}$$

Let $C_{fg}^2 = \max(C_f^2, C_g^2)$, $L_{fg}^2 = L_f^2 C_g^2 + L_g^2$, $\nu = \frac{1}{q}$ and $\eta_t \leq \frac{\rho}{24\gamma q L_{fg}C_{fg}}$ for all $t \geq 0$. Since $\beta_t \in (0,1)$ for all $t \geq 0$, we have

$$(1+\nu)(1-\beta_t)^2 + (1+\frac{1}{\nu})\frac{72C_f^2(L_f^2 C_g^2 + L_g^2)\eta_{t-1}^2\gamma^2}{\rho^2}$$

$$\leq 1 + \frac{1}{q} + (1+q)\frac{72C_f^2(L_f^2 C_g^2 + L_g^2)\gamma^2}{\rho^2}\frac{\rho^2}{576\gamma^2 q^2 L_{fg}^2 C_{fg}^2}$$

$$\leq 1 + \frac{1}{q} + \frac{1+q}{8q^2} \leq 1 + \frac{5}{4q}. \tag{60}$$

Similarly, since $\varrho_t \in (0,1)$ for all $t \geq 0$, we have $(1+\nu)(1-\varrho_t)^2 + (1+\frac{1}{\nu})\frac{72C_g^2(C_g^2L_f^2+L_g^2)\eta_{t-1}^2\gamma^2}{\rho^2} \leq 1 + \frac{5}{4q}$. Based on the above inequality (58) and the parameters, then we have

$$\sum_{m=1}^{M}\left(\mathbb{E}\|u_t^m - \bar{u}_t\|^2 + \mathbb{E}\|v_t^m - \bar{v}_t\|^2\right) \tag{61}$$

$$\leq \left(1+\frac{5}{4q}\right)\sum_{m=1}^{M}\left(\mathbb{E}\|u_{t-1}^m - \bar{u}_{t-1}\|^2 + \mathbb{E}\|v_{t-1}^m - \bar{v}_{t-1}\|^2\right) + 8(q+1)L_{fg}^2\eta_{t-1}^2\gamma^2\sum_{m=1}^{M}\mathbb{E}\|\bar{d}_{t-1}\|^2$$

$$+ 216(q^2-1)\frac{C_{fg}^2L_{fg}^2\gamma^2}{\rho^2}(\alpha_t^2+\beta_t^2)\sum_{l=s_t}^{t-1}\eta_l^2\sum_{m=1}^{M}\left(\mathbb{E}\|v_l^m - \bar{v}_l\|^2 + \mathbb{E}\|u_l^m - \bar{u}_l\|^2\right)$$

$$+ 32(1+q)(\varrho_t^2+\alpha_t^2)L_f^2\sum_{m=1}^{M}\mathbb{E}\|h_{t-1}^m - g^m(\bar{x}_{t-1})\|^2$$

$$+ 4M(q+1)\left(2L_f^2\alpha_t^2\sigma^2 + \varrho_t^2\sigma^2 + 4\varrho_t^2\delta_f^2 + 4L_f^2\varrho_t^2\delta_g^2 + \sigma^2\beta_t^2 + 3\delta_g^2\beta_t^2\right)$$

$$\leq \left(1+\frac{5}{4q}\right)\sum_{m=1}^{M}\left(\mathbb{E}\|u_{t-1}^m - \bar{u}_{t-1}\|^2 + \mathbb{E}\|v_{t-1}^m - \bar{v}_{t-1}\|^2\right) + \frac{\rho^2}{36qC_{fg}^2}\sum_{m=1}^{M}\mathbb{E}\|\bar{d}_{t-1}\|^2$$

$$+ \frac{3(c_1^2+c_2^2)}{8}\eta_{t-1}^2\sum_{l=s_t}^{t-2}\eta_l^2\sum_{m=1}^{M}\left(\mathbb{E}\|v_l^m - \bar{v}_l\|^2 + \mathbb{E}\|u_l^m - \bar{u}_l\|^2\right) + \frac{\rho^2(c_1^2+c_3^2)}{9q\gamma^2C_{fg}^2C_g^2}\eta_{t-1}^2\sum_{m=1}^{M}\mathbb{E}\|h_{t-1}^m - g^m(\bar{x}_{t-1})\|^2$$

$$+ \frac{M\rho}{3\gamma L_{fg}C_{fg}}\left(2c_1^2L_f^2\sigma^2 + c_3^2\sigma^2 + 4c_3^2\delta_f^2 + 4c_3^2L_f^2\delta_g^2 + c_2^2\sigma^2 + 3c_2^2\delta_g^2\right)\eta_{t-1}^3, \tag{62}$$

where the first inequality holds by the above inequality (52) and $\nu = \frac{1}{q}$, and the last inequality holds by $\alpha_t = c_1\eta_{t-1}^2$, $\beta_t = c_2\eta_{t-1}^2$, $\varrho_t = c_3\eta_{t-1}^2$ and $\eta_t \leq \frac{\rho}{24\gamma q L_{fg}C_{fg}}$ for all $t \geq 0$, and $\frac{L_f^2}{L_{fg}^2} \leq \frac{1}{C_g^2}$.

According to the above inequality (61), we have

$$
\sum_{m=1}^{M} \left( \mathbb{E} \|u_t^m - \bar{u}_t\|^2 + \mathbb{E} \|v_t^m - \bar{v}_t\|^2 \right)
$$

$$
\leq \frac{\rho^2}{36qC_{fg}^2} \sum_{s=s_t}^{t-1} \left(1 + \frac{5}{4q}\right)^{t-1-s} \sum_{m=1}^{M} \mathbb{E}\|\bar{d}_s\|^2
$$

$$
+ \frac{3(c_1^2 + c_2^2)}{8} \sum_{s=s_t}^{t-1} \left(1 + \frac{5}{4q}\right)^{t-1-s} \eta_s^2 \sum_{l=s_t}^{s-2} \eta_l^2 \sum_{m=1}^{M} \left( \mathbb{E}\|v_l^m - \bar{v}_l\|^2 + \mathbb{E}\|u_l^m - \bar{u}_l\|^2 \right)
$$

$$
+ \frac{\rho^2(c_1^2 + c_3^2)}{9q\gamma^2 C_{fg}^2 C_g^2} \sum_{s=s_t}^{t-1} \left(1 + \frac{5}{4q}\right)^{t-1-s} \eta_s^2 \sum_{m=1}^{M} \mathbb{E}\|h_{s-1}^m - g^m(\bar{x}_{s-1})\|^2
$$

$$
+ \frac{M\rho}{3\gamma L_{fg} C_{fg}} \left(2c_1^2 L_f^2 \sigma^2 + c_3^2 \sigma^2 + 4c_3^2 \delta_f^2 + 4c_3^2 L_f^2 \delta_g^2 + c_2^2 \sigma^2 + 3c_2^2 \delta_g^2\right) \sum_{s=s_t}^{t-1} \left(1 + \frac{5}{4q}\right)^{t-1-s} \eta_s^3
$$

$$
\leq \frac{\rho^2}{9qC_{fg}^2} \sum_{s=s_t}^{t-1} \sum_{m=1}^{M} \mathbb{E}\|\bar{d}_s\|^2 + \frac{3(c_1^2 + c_2^2)}{2} \sum_{s=s_t}^{t-1} \eta_s^2 \sum_{l=s_t}^{s-2} \eta_l^2 \sum_{m=1}^{M} \left( \mathbb{E}\|v_l^m - \bar{v}_l\|^2 + \mathbb{E}\|u_l^m - \bar{u}_l\|^2 \right)
$$

$$
+ \frac{4\rho^2(c_1^2 + c_3^2)}{9q\gamma^2 C_{fg}^2 C_g^2} \sum_{s=s_t}^{t-1} \eta_s^2 \sum_{m=1}^{M} \mathbb{E}\|h_{s-1}^m - g^m(\bar{x}_{s-1})\|^2
$$

$$
+ \frac{4M\rho}{3\gamma L_{fg} C_{fg}} \left(2c_1^2 L_f^2 \sigma^2 + c_3^2 \sigma^2 + 4c_3^2 \delta_f^2 + 4c_3^2 L_f^2 \delta_g^2 + c_2^2 \sigma^2 + 3c_2^2 \delta_g^2\right) \sum_{s=s_t}^{t-1} \eta_s^3
$$

$$
\leq \frac{M\rho^2}{9qC_{fg}^2} \sum_{s=s_t}^{t-1} \mathbb{E}\|\bar{d}_s\|^2 + \frac{\rho^2(c_1^2 + c_2^2)}{24 * 16\gamma^2 q L_{fg}^2 C_{fg}^2} \sum_{s=s_t}^{t-1} \eta_s^2 \sum_{m=1}^{M} \left( \mathbb{E}\|v_s^m - \bar{v}_s\|^2 + \mathbb{E}\|u_s^m - \bar{u}_s\|^2 \right)
$$

$$
+ \frac{4\rho^2(c_1^2 + c_3^2)}{9q\gamma^2 C_{fg}^2 C_g^2} \sum_{s=s_t}^{t-1} \eta_s^2 \sum_{m=1}^{M} \mathbb{E}\|h_{s-1}^m - g^m(\bar{x}_{s-1})\|^2
$$

$$
+ \frac{4M\rho}{3\gamma L_{fg} C_{fg}} \left(2c_1^2 L_f^2 \sigma^2 + c_3^2 \sigma^2 + 4c_3^2 \delta_f^2 + 4c_3^2 L_f^2 \delta_g^2 + c_2^2 \sigma^2 + 3c_2^2 \delta_g^2\right) \sum_{s=s_t}^{t-1} \eta_s^3, \tag{63}
$$

where the second inequality holds by $\left(1 + \frac{5}{4q}\right)^{t-1-s} \leq \left(1 + \frac{5}{4q}\right)^q \leq e^{5/4} \leq 4$ and the last inequality holds by $\eta_t \leq \frac{\rho}{24\gamma q L_{fg} C_{fg}}$ for all $t \geq 0$.

By multiplying both sides of (63) by $\eta_t$ and summing over $t = s_t$ to $s_t + q - 1$, we have

$$
\sum_{t=s_t}^{s_t+q-1} \eta_t \sum_{m=1}^{M} \left( \mathbb{E}\|u_t^m - \bar{u}_t\|^2 + \mathbb{E}\|v_t^m - \bar{v}_t\|^2 \right)
$$

$$
\leq \frac{M\rho^2}{9C_{fg}^2} \sum_{t=s_t}^{s_t+q-1} \eta_t \mathbb{E}\|\bar{d}_t\|^2 + \frac{\rho^4(c_1^2 + c_2^2)}{24^3 * 16\gamma^4 q^2 L_{fg}^4 C_{fg}^4} \sum_{t=s_t}^{s_t+q-1} \eta_t \sum_{m=1}^{M} \left( \mathbb{E}\|v_t^m - \bar{v}_t\|^2 + \mathbb{E}\|u_t^m - \bar{u}_t\|^2 \right)
$$

$$
+ \frac{\rho^4(c_1^2 + c_3^2)}{24 * 54q^2\gamma^4 C_{fg}^4 L_{fg}^2 C_g^2} \sum_{t=s_t}^{s_t+q-1} \eta_t \sum_{m=1}^{M} \mathbb{E}\|h_t^m - g^m(\bar{x}_t)\|^2
$$

$$
+ \frac{M\rho^2}{18\gamma^2 L_{fg}^2 C_{fg}^2} \left(2c_1^2 L_f^2 \sigma^2 + c_3^2 \sigma^2 + 4c_3^2 \delta_f^2 + 4c_3^2 L_f^2 \delta_g^2 + c_2^2 \sigma^2 + 3c_2^2 \delta_g^2\right) \sum_{t=s_t}^{s_t+q-1} \eta_t^3, \tag{64}
$$

Given $c_1^2 + c_2^2 \leq \frac{(24)^4 q^2 \gamma^4 L_{fg}^4 C_{fg}^4}{9\rho^4}$, we have $\frac{60}{72} \leq 1 - \frac{\rho^4(c_1^2+c_2^2)}{24^3 * 16 \gamma^4 q^2 L_{fg}^4 C_{fg}^4}$, we have

$$
\sum_{t=s_t}^{s_t+q-1} \eta_t \sum_{m=1}^{M} \left( \mathbb{E} \|u_t^m - \bar{u}_t\|^2 + \mathbb{E} \|v_t^m - \bar{v}_t\|^2 \right)
$$

$$
\leq \frac{2M\rho^2}{15C_{fg}^2} \sum_{t=s_t}^{s_t+q-1} \eta_t \mathbb{E}\|\bar{d}_t\|^2 + \frac{\rho^4(c_1^2+c_3^2)}{1080 q^2 \gamma^4 C_{fg}^4 L_{fg}^2 C_g^2} \sum_{t=s_t}^{s_t+q-1} \eta_t \sum_{m=1}^{M} \mathbb{E}\|h_t^m - g^m(\bar{x}_t)\|^2
$$

$$
+ \frac{M\rho^2}{15\gamma^2 L_{fg}^2 C_{fg}^2} \left( 2c_1^2 L_f^2 \sigma^2 + c_3^2 \sigma^2 + 4c_3^2 \delta_f^2 + 4c_3^2 L_f^2 \delta_g^2 + c_2^2 \sigma^2 + 3c_2^2 \delta_g^2 \right) \sum_{t=s_t}^{s_t+q-1} \eta_t^3. \quad (65)
$$

According to the above inequality (45) and $C_{fg}^2 = \max(C_f^2, C_g^2)$, we have

$$
\sum_{m=1}^{M} \mathbb{E}\|d_t^m - \bar{d}_t\|^2 \leq \frac{9C_g^2}{\rho^2} \sum_{m=1}^{M} \mathbb{E}\|v_t^m - \bar{v}_t\|^2 + \frac{9C_f^2}{\rho^2} \sum_{m=1}^{M} \mathbb{E}\|u_t^m - \bar{u}_t\|^2 \leq \frac{9C_{fg}^2}{\rho^2} \sum_{m=1}^{M} \left( \mathbb{E}\|u_t^m - \bar{u}_t\|^2 + \mathbb{E}\|v_t^m - \bar{v}_t\|^2 \right).
$$
$$(66)$$

Thus we have

$$
\sum_{t=s_t}^{s_t+q-1} \eta_t \sum_{m=1}^{M} \mathbb{E}\|d_t^m - \bar{d}_t\|^2
$$

$$
\leq \frac{9C_{fg}^2}{\rho^2} \sum_{t=s_t}^{s_t+q-1} \eta_t \sum_{m=1}^{M} \left( \mathbb{E}\|u_t^m - \bar{u}_t\|^2 + \mathbb{E}\|v_t^m - \bar{v}_t\|^2 \right)
$$

$$
\leq \frac{6M}{5} \sum_{t=s_t}^{s_t+q-1} \eta_t \mathbb{E}\|\bar{d}_t\|^2 + \frac{\rho^2(c_1^2+c_3^2)}{120 q^2 \gamma^4 C_{fg}^2 L_{fg}^2 C_g^2} \sum_{t=s_t}^{s_t+q-1} \eta_t \sum_{m=1}^{M} \mathbb{E}\|h_t^m - g^m(\bar{x}_t)\|^2
$$

$$
+ \frac{3M}{5\gamma^2 L_{fg}^2} \left( 2c_1^2 L_f^2 \sigma^2 + c_3^2 \sigma^2 + 4c_3^2 \delta_f^2 + 4c_3^2 L_f^2 \delta_g^2 + c_2^2 \sigma^2 + 3c_2^2 \delta_g^2 \right) \sum_{t=s_t}^{s_t+q-1} \eta_t^3. \quad (67)
$$

$\square$

**Theorem 3.** *(Restatement of Theorem 1) Assume the sequence $\{\bar{x}_t\}_{t=1}^{T}$ be generated from **AdaMFCGD** algorithm. Under the above Assumptions, and let $\eta_t = \frac{k}{(n+t)^{1/3}}$ for all $t \geq 0$, $\alpha_{t+1} = c_1 \eta_t^2$, $\beta_{t+1} = c_2 \eta_t^2$, $\varrho_{t+1} = c_3 \eta_t^2$, $n \geq \max\left(2, k^3, (c_1 k)^3, (c_2 k)^3, (c_3 k)^3, \frac{(24k\gamma q L_{fg} C_{fg})^3}{\rho^3}\right)$, $k > 0$, $c_1 \geq \frac{2}{3k^3} + B$, $c_2 \geq \frac{2}{3k^3} + 5C_f^2$, $c_1^2 + c_2^2 \leq \frac{(24)^4 q^2 \gamma^4 L_{fg}^4 C_{fg}^4}{9\rho^4}$, $c_3 \geq \frac{2}{3k^3} + 5C_g^2$, $\frac{\rho(c_1^2+c_3^2)^{1/4}}{12\sqrt{5}q L_{fg} C_{fg}} \leq \gamma \leq \min\left(\frac{3\rho q L_{fg} C_{fg}}{4(C_g^2+L_g^2+2L_f^2 C_g^2)}, \frac{n^{1/3}\rho}{2Lk}\right)$, $B \geq 20C_g^2 L_f^2 + \frac{c_2^2 C_g^2 L_f^2}{216 q^3 \gamma^3 L_{fg}^3 C_{fg}^3} + \frac{\Theta \rho^2 (c_1^2+c_3^2)}{30 q^2 \gamma^4 C_{fg}^2 L_{fg}^2 C_g^2}$, $\Theta = \left(5C_f^2 L_g^2 + \frac{c_2^2 C_g^2 L_f^2}{864 q^3 \gamma^3 L_{fg}^3 C_{fg}^3}\right) \frac{\rho^2}{(24)^2 L_{fg}^2 C_{fg}^2} + \frac{\gamma\rho}{6q L_{fg} C_{fg}}\left(C_g^2 + L_g^2 + 2L_f^2 C_g^2\right)$ and $\Theta + \frac{BC_g^2 \rho^2}{(24)^2 L_{fg}^2 C_{fg}^2} \leq \frac{5\rho^2}{48}$, we have*

$$
\frac{1}{T} \sum_{t=1}^{T} \mathbb{E}\|\nabla F(\bar{x}_t)\| \leq \left( \frac{\sqrt{2G} n^{1/6}}{T^{1/2}} + \frac{\sqrt{2G}}{T^{1/3}} \right) \sqrt{\frac{1}{T} \sum_{t=1}^{T} \mathbb{E}\|A_t\|^2}, \quad (68)
$$

*where $C_{fg}^2 = \max(C_f^2, C_g^2)$, $L_{fg}^2 = L_f^2 C_g^2 + L_g^2$, $G = \frac{4(F(\bar{x}_1) - F^*)}{k\rho\gamma} + \frac{12 n^{1/3}\sigma^2}{qk^2\rho^2} + 4k^2\left(\frac{\hat{\delta}^2}{4\gamma^2 L_{fg}^2} + \frac{(c_1^2+c_2^2+c_3^2)\sigma^2}{3\rho\gamma q L_{fg} C_{fg}}\right) \ln(n+T)$ and $\hat{\delta}^2 = 2c_1^2 L_f^2 \sigma^2 + c_3^2 \sigma^2 + 4c_3^2 \delta_f^2 + 4c_3^2 L_f^2 \delta_g^2 + c_2^2 \sigma^2 + 3c_2^2 \delta_g^2$.*

*Proof.* Since $\eta_t = \frac{k}{(n+t)^{1/3}}$ on $t$ is decreasing and $n \geq k^3$, we have $\eta_t \leq \eta_0 = \frac{k}{n^{1/3}} \leq 1$ and $\gamma \leq \frac{n^{1/3}\rho}{2Lk} \leq \frac{\rho}{2L\eta_0} \leq \frac{\rho}{2L\eta_t}$ for any $t \geq 0$. Since $\eta_t \leq \frac{\rho}{24\gamma q L_{fg} C_{fg}}$ for all $t \geq 0$, we have

$\frac{k}{n^{1/3}} = \eta_0 \leq \eta_t \leq \frac{\rho}{24\gamma q L_{fg} C_{fg}}$, then we have $n \geq \frac{(24k\gamma q L_{fg} C_{fg})^3}{\rho^3}$. Due to $0 < \eta_t \leq 1$ and $n \geq (c_1 k)^3$, we have $\alpha_{t+1} = c_1\eta_t^2 \leq c_1\eta_t \leq \frac{c_1 k}{n^{1/3}} \leq 1$. Similarly, due to $n \geq (c_2 k)^3$ and $n \geq (c_3 k)^3$, we have $\beta_{t+1} \leq 1$ and $\varrho_{t+1} \leq 1$.

According to Lemma 7, for any $m \in [M]$, we have

$$\frac{1}{\eta_t}\mathbb{E}\|h_{t+1}^m - g^m(x_{t+1}^m)\|^2 - \frac{1}{\eta_{t-1}}\mathbb{E}\|h_t^m - g^m(x_t^m)\|^2 \tag{69}$$

$$\leq \left(\frac{1 - \alpha_{t+1}}{\eta_t} - \frac{1}{\eta_{t-1}}\right)\mathbb{E}\|h_t^m - g^m(x_t^m)\|^2 + 2C_g^2\mathbb{E}\|x_{t+1}^m - x_t^m\|^2 + 2\alpha_{t+1}^2\sigma^2$$

$$= \left(\frac{1}{\eta_t} - \frac{1}{\eta_{t-1}} - c_1\eta_t\right)\mathbb{E}\|h_t^m - g^m(x_t^m)\|^2 + 2C_g^2\mathbb{E}\|x_{t+1}^m - x_t^m\|^2 + 2\alpha_{t+1}^2\sigma^2,$$

where the second equality is due to $\alpha_{t+1} = c_1\eta_t^2$. Similarly, since $\beta_{t+1} = c_2\eta_t^2$, we have

$$\frac{1}{\eta_t}\mathbb{E}\|u_{t+1}^m - \nabla g^m(x_{t+1}^m)\|^2 - \frac{1}{\eta_{t-1}}\mathbb{E}\|u_t^m - \nabla g^m(x_t^m)\|^2 \tag{70}$$

$$\leq \left(\frac{1 - \beta_{t+1}}{\eta_t} - \frac{1}{\eta_{t-1}}\right)\mathbb{E}\|u_t^m - \nabla g^m(x_t^m)\|^2 + 2L_g^2\mathbb{E}\|x_{t+1}^m - x_t^m\|^2 + 2\beta_{t+1}^2\sigma^2$$

$$= \left(\frac{1}{\eta_t} - \frac{1}{\eta_{t-1}} - c_2\eta_t\right)\mathbb{E}\|u_t^m - \nabla g^m(x_t^m)\|^2 + 2L_g^2\mathbb{E}\|x_{t+1}^m - x_t^m\|^2 + 2\beta_{t+1}^2\sigma^2.$$

And we have

$$\frac{1}{\eta_t}\mathbb{E}\|v_{t+1}^m - \nabla f^m(h_{t+1}^m)\|^2 - \frac{1}{\eta_{t-1}}\mathbb{E}\|v_t^m - \nabla f^m(h_t^m)\|^2 \tag{71}$$

$$\leq \left(\frac{1 - \varrho_{t+1}}{\eta_t} - \frac{1}{\eta_{t-1}}\right)\mathbb{E}\|v_t^m - \nabla f^m(h_t^m)\|^2 + 4L_f^2 C_g^2\mathbb{E}\|x_{t+1}^m - x_t^m\|^2 + 2\varrho_{t+1}^2\sigma^2$$

$$+ 8\alpha_{t+1}^2 L_f^2\mathbb{E}\|h_t^m - g^m(x_t^m)\|^2 + 8L_f^2\alpha_{t+1}^2\sigma^2$$

$$= \left(\frac{1}{\eta_t} - \frac{1}{\eta_{t-1}} - c_3\eta_t\right)\mathbb{E}\|v_t^m - \nabla f^m(h_t^m)\|^2 + 4L_f^2 C_g^2\mathbb{E}\|x_{t+1}^m - x_t^m\|^2 + 2\varrho_{t+1}^2\sigma^2$$

$$+ 8\alpha_{t+1}^2 L_f^2\mathbb{E}\|h_t^m - g^m(x_t^m)\|^2 + 8L_f^2\alpha_{t+1}^2\sigma^2,$$

where the second equality is due to $\varrho_{t+1} = c_2\eta_t^2$.

By $\eta_t = \frac{k}{(n+t)^{1/3}}$, we have

$$\frac{1}{\eta_t} - \frac{1}{\eta_{t-1}} = \frac{1}{k}\left((n+t)^{\frac{1}{3}} - (n+t-1)^{\frac{1}{3}}\right) \leq \frac{1}{3k(n+t-1)^{2/3}} \leq \frac{1}{3k(n/2+t)^{2/3}}$$

$$\leq \frac{2^{2/3}}{3k(n+t)^{2/3}} = \frac{2^{2/3}}{3k^3}\frac{k^2}{(n+t)^{2/3}} = \frac{2^{2/3}}{3k^3}\eta_t^2 \leq \frac{2}{3k^3}\eta_t, \tag{72}$$

where the first inequality holds by the concavity of function $f(x) = x^{1/3}$, i.e., $(x+y)^{1/3} \leq x^{1/3} + \frac{y}{3x^{2/3}}$; the second inequality is due to $n \geq 2$, and the last inequality is due to $0 < \eta_t \leq 1$.

Let $c_1 \geq \frac{2}{3k^3} + B$, for any $m \in [M]$, we have

$$\frac{1}{\eta_t}\mathbb{E}\|h_{t+1}^m - g^m(x_{t+1}^m)\|^2 - \frac{1}{\eta_{t-1}}\mathbb{E}\|h_t^m - g^m(x_t^m)\|^2 \tag{73}$$

$$\leq -B\eta_t\mathbb{E}\|h_t^m - g^m(x_t^m)\|^2 + 2C_g^2\mathbb{E}\|x_{t+1}^m - x_t^m\|^2 + 2\alpha_{t+1}^2\sigma^2$$

$$= -B\eta_t\mathbb{E}\|h_t^m - g^m(x_t^m)\|^2 + 2C_g^2\eta_t^2\gamma^2\mathbb{E}\|d_t^m - \bar{d}_t + \bar{d}_t\|^2 + 2\alpha_{t+1}^2\sigma^2$$

$$\leq -B\eta_t\mathbb{E}\|h_t^m - g^m(x_t^m)\|^2 + 4C_g^2\eta_t^2\gamma^2\mathbb{E}\|d_t^m - \bar{d}_t\|^2 + 4C_g^2\eta_t^2\gamma^2\mathbb{E}\|\bar{d}_t\|^2 + 2\alpha_{t+1}^2\sigma^2$$

$$\leq -\frac{B}{2}\eta_t\|h_t^m - g^m(\bar{x}_t)\|^2 + BC_g^2\eta_t\|x_t^m - \bar{x}_t\|^2 + 4C_g^2\eta_t^2\gamma^2\mathbb{E}\|d_t^m - \bar{d}_t\|^2 + 4C_g^2\eta_t^2\gamma^2\mathbb{E}\|\bar{d}_t\|^2 + 2\alpha_{t+1}^2\sigma^2,$$

where the last inequality holds by $-\|h_t^m - g^m(x_t^m)\|^2 \leq -\frac{1}{2}\|h_t^m - g^m(\bar{x}_t)\|^2 + \|g^m(x_t^m) - g^m(\bar{x}_t)\|^2 \leq -\frac{1}{2}\|h_t^m - g^m(\bar{x}_t)\|^2 + C_g^2\|x_t^m - \bar{x}_t\|^2$.

Let $c_2 \geq \frac{2}{3k^3} + 5C_f^2$, for any $m \in [M]$, we have

$$\frac{1}{\eta_t}\mathbb{E}\|u_{t+1}^m - \nabla g^m(x_{t+1}^m)\|^2 - \frac{1}{\eta_{t-1}}\mathbb{E}\|u_t^m - \nabla g^m(x_t^m)\|^2 \tag{74}$$

$$\leq -5C_f^2\eta_t\mathbb{E}\|u_t^m - \nabla g^m(x_t^m)\|^2 + 2L_g^2\mathbb{E}\|x_{t+1}^m - x_t^m\|^2 + 2\beta_{t+1}^2\sigma^2$$

$$= -5C_f^2\eta_t\mathbb{E}\|u_t^m - \nabla g^m(x_t^m)\|^2 + 2L_g^2\eta_t^2\gamma^2\mathbb{E}\|d_t^m - \bar{d}_t + \bar{d}_t\|^2 + 2\beta_{t+1}^2\sigma^2$$

$$\leq -5C_f^2\eta_t\mathbb{E}\|u_t^m - \nabla g^m(x_t^m)\|^2 + 4L_g^2\eta_t^2\gamma^2\mathbb{E}\|d_t^m - \bar{d}_t\|^2 + 4L_g^2\eta_t^2\gamma^2\mathbb{E}\|\bar{d}_t\|^2 + 2\beta_{t+1}^2\sigma^2$$

$$\leq -\frac{5C_f^2\eta_t}{2}\mathbb{E}\|u_t^m - \nabla g^m(\bar{x}_t)\|^2 + 5C_f^2L_g^2\eta_t\|x_t^m - \bar{x}_t\|^2 + 4L_g^2\eta_t^2\gamma^2\mathbb{E}\|d_t^m - \bar{d}_t\|^2 + 4L_g^2\eta_t^2\gamma^2\mathbb{E}\|\bar{d}_t\|^2 + 2\beta_{t+1}^2\sigma^2,$$

where the last inequality holds by $-\|u_t^m - \nabla g^m(x_t^m)\|^2 \leq -\frac{1}{2}\|u_t^m - \nabla g^m(\bar{x}_t)\|^2 + \|\nabla g(x_t^m) - \nabla g^m(\bar{x}_t)\|^2 \leq -\frac{1}{2}\|u_t^m - \nabla g^m(\bar{x}_t)\|^2 + L_g^2\|x_t^m - \bar{x}_t\|^2$.

Let $c_3 \geq \frac{2}{3k^3} + 5C_g^2$, for any $m \in [M]$, we have

$$\frac{1}{\eta_t}\mathbb{E}\|v_{t+1}^m - \nabla f^m(h_{t+1}^m)\|^2 - \frac{1}{\eta_{t-1}}\mathbb{E}\|v_t^m - \nabla f^m(h_t^m)\|^2 \tag{75}$$

$$\leq -5C_g^2\eta_t\mathbb{E}\|v_t^m - \nabla f^m(h_t^m)\|^2 + 4L_f^2C_g^2\mathbb{E}\|x_{t+1}^m - x_t^m\|^2 + 2\varrho_{t+1}^2\sigma^2 + 8\alpha_{t+1}^2L_f^2\mathbb{E}\|h_t^m - g^m(x_t^m)\|^2 + 8L_f^2\alpha_{t+1}^2\sigma^2$$

$$= -5C_g^2\eta_t\mathbb{E}\|v_t^m - \nabla f^m(h_t^m)\|^2 + 4L_f^2C_g^2\eta_t^2\gamma^2\mathbb{E}\|d_t^m - \bar{d}_t + \bar{d}_t\|^2 + 2\varrho_{t+1}^2\sigma^2$$
$$+ 8\alpha_{t+1}^2L_f^2\mathbb{E}\|h_t^m - g^m(x_t^m)\|^2 + 8L_f^2\alpha_{t+1}^2\sigma^2$$

$$\leq -5C_g^2\eta_t\mathbb{E}\|v_t^m - \nabla f^m(h_t^m)\|^2 + 8L_f^2C_g^2\eta_t^2\gamma^2\mathbb{E}\|d_t^m - \bar{d}_t\|^2 + 8L_f^2C_g^2\eta_t^2\gamma^2\mathbb{E}\|\bar{d}_t\|^2 + 2\varrho_{t+1}^2\sigma^2$$
$$+ 8\alpha_{t+1}^2L_f^2\mathbb{E}\|h_t^m - g^m(x_t^m)\|^2 + 8L_f^2\alpha_{t+1}^2\sigma^2$$

$$\leq -5C_g^2\eta_t\mathbb{E}\|v_t^m - \nabla f^m(h_t^m)\|^2 + 8L_f^2C_g^2\eta_t^2\gamma^2\mathbb{E}\|d_t^m - \bar{d}_t\|^2 + 8L_f^2C_g^2\eta_t^2\gamma^2\mathbb{E}\|\bar{d}_t\|^2 + 2\varrho_{t+1}^2\sigma^2$$
$$+ \frac{c_2^2L_f^2}{864q^3\gamma^3L_{fg}^3C_{fg}^3}\eta_t\mathbb{E}\|h_t^m - g^m(\bar{x}_t)\|^2 + \frac{c_2^2C_g^2L_f^2}{864q^3\gamma^3L_{fg}^3C_{fg}^3}\eta_t\mathbb{E}\|x_t^m - \bar{x}_t\|^2 + 8L_f^2\alpha_{t+1}^2\sigma^2,$$

where the last inequality holds by Assumption , $\alpha_{t+1} = c_2\eta_t^2$ and $\eta_t \leq \frac{\rho}{24q\gamma L_{fg}C_{fg}}$ for all $t \geq 0$.

According to Lemma 4, we have

$$F(\bar{x}_{t+1}) - F(\bar{x}_t) \leq \frac{1}{M}\sum_{m=1}^M \Big(\frac{2C_f^2\eta_t\gamma}{\rho}\|u_t^m - \nabla g^m(\bar{x}_t)\|^2 + \frac{4C_g^2\eta_t\gamma}{\rho}\|v_t^m - \nabla f^m(h_t^m)\|^2$$

$$+ \frac{4C_g^2L_f^2\eta_t\gamma}{\rho}\|h_t^m - g^m(\bar{x}_t)\|^2\Big) - \frac{\rho}{2\eta_t\gamma}\|\bar{x}_{t+1} - \bar{x}_t\|^2$$

$$= \frac{1}{M}\sum_{m=1}^M \Big(\frac{2C_f^2\eta_t\gamma}{\rho}\|u_t^m - \nabla g^m(\bar{x}_t)\|^2 + \frac{4C_g^2\eta_t\gamma}{\rho}\|v_t^m - \nabla f^m(h_t^m)\|^2$$

$$+ \frac{4C_g^2L_f^2\eta_t\gamma}{\rho}\|h_t^m - g^m(\bar{x}_t)\|^2\Big) - \frac{\rho\eta_t\gamma}{2}\|\bar{d}_t\|^2. \tag{76}$$

$$\sum_{t=s_t}^{s_t+q-1} \eta_t \sum_{m=1}^M \mathbb{E}\|d_t^m - \bar{d}_t\|^2$$

$$\leq \frac{6M}{5}\sum_{t=s_t}^{s_t+q-1}\eta_t\mathbb{E}\|\bar{d}_t\|^2 + \frac{\rho^2(c_1^2+c_3^2)}{120q^2\gamma^4C_{fg}^2L_{fg}^2C_g^2}\sum_{t=s_t}^{s_t+q-1}\eta_t\sum_{m=1}^M\mathbb{E}\|h_t^m - g^m(\bar{x}_t)\|^2 + \frac{3M\hat{\delta}^2}{5\gamma^2L_{fg}^2}\sum_{t=s_t}^{s_t+q-1}\eta_t^3, \tag{77}$$

Next, we define a *potential* function, for any $t \geq 1$

$$\Omega_t = \mathbb{E}\Big[F(\bar{x}_t) + \frac{\gamma}{\rho\eta_{t-1}}\frac{1}{M}\sum_{m=1}^M \Big(\|h_t^m - g^m(x_t^m)\|^2 + \|u_t^m - \nabla g^m(x_t^m)\|^2 + \|v_t^m - \nabla f^m(h_t^m)\|^2\Big)\Big].$$

Then we have

$$\Omega_{t+1} - \Omega_t$$

$$= F(\bar{x}_{t+1}) - F(\bar{x}_t) + \frac{\gamma}{M\rho} \sum_{m=1}^{M} \left( \frac{1}{\eta_t} \mathbb{E}\|h_{t+1}^m - g^m(x_{t+1}^m)\|^2 - \frac{1}{\eta_{t-1}} \mathbb{E}\|h_t^m - g^m(x_t^m)\|^2 + \frac{1}{\eta_t}\|u_{t+1}^m - \nabla g^m(x_{t+1}^m)\|^2 \right.$$

$$\left. - \frac{1}{\eta_{t-1}}\|u_t^m - \nabla g^m(x_t^m)\|^2 + \frac{1}{\eta_t}\|v_{t+1}^m - \nabla f^m(h_{t+1}^m)\|^2 - \frac{1}{\eta_{t-1}}\|v_t^m - \nabla f^m(h_t^m)\|^2 \right)$$

$$\leq \frac{1}{M} \sum_{m=1}^{M} \left( \frac{2C_f^2\eta_t\gamma}{\rho}\|u_t^m - \nabla g^m(\bar{x}_t)\|^2 + \frac{4C_g^2\eta_t\gamma}{\rho}\|v_t^m - \nabla f^m(h_t^m)\|^2 + \frac{4C_g^2 L_f^2\eta_t\gamma}{\rho}\|h_t^m - g^m(\bar{x}_t)\|^2 \right) - \frac{\rho\eta_t\gamma}{2}\|\bar{d}_t\|^2$$

$$+ \frac{\gamma}{M\rho} \sum_{m=1}^{M} \left( -\frac{B}{2}\eta_t\|h_t^m - g^m(\bar{x}_t)\|^2 + BC_g^2\eta_t\|x_t^m - \bar{x}_t\|^2 + 4C_g^2\eta_t^2\gamma^2 \mathbb{E}\|d_t^m - \bar{d}_t\|^2 + 4C_g^2\eta_t^2\gamma^2 \mathbb{E}\|\bar{d}_t\|^2 + 2\alpha_{t+1}^2\sigma^2 \right.$$

$$- \frac{5C_f^2\eta_t}{2}\mathbb{E}\|u_t^m - \nabla g^m(\bar{x}_t)\|^2 + 5C_f^2 L_g^2\eta_t\|x_t^m - \bar{x}_t\|^2 + 4L_g^2\eta_t^2\gamma^2 \mathbb{E}\|d_t^m - \bar{d}_t\|^2 + 4L_g^2\eta_t^2\gamma^2 \mathbb{E}\|\bar{d}_t\|^2 + 2\beta_{t+1}^2\sigma^2$$

$$- 5C_g^2\eta_t\mathbb{E}\|v_t^m - \nabla f^m(h_t^m)\|^2 + 8L_f^2 C_g^2\eta_t^2\gamma^2 \mathbb{E}\|d_t^m - \bar{d}_t\|^2 + 8L_f^2 C_g^2\eta_t^2\gamma^2 \mathbb{E}\|\bar{d}_t\|^2 + 2\varrho_{t+1}^2\sigma^2$$

$$\left. + \frac{c_2^2 L_f^2}{864q^3\gamma^3 L_{fg}^3 C_{fg}^3}\eta_t\mathbb{E}\|h_t^m - g^m(\bar{x}_t)\|^2 + \frac{c_2^2 C_g^2 L_f^2}{864q^3\gamma^3 L_{fg}^3 C_{fg}^3}\eta_t\mathbb{E}\|x_t^m - \bar{x}_t\|^2 + 8L_f^2\alpha_{t+1}^2\sigma^2 \right)$$

$$\leq \frac{1}{M} \sum_{m=1}^{M} \left( -\frac{C_f^2\gamma}{2\rho}\eta_t\|u_t^m - \nabla g^m(\bar{x}_t)\|^2 - \frac{C_g^2\gamma}{\rho}\eta_t\|v_t^m - \nabla f^m(h_t^m)\|^2 \right.$$

$$\left. - \frac{\gamma}{\rho}\left(\frac{B}{2} - 4C_g^2 L_f^2 - \frac{c_2^2 C_g^2 L_f^2}{864q^3\gamma^3 L_{fg}^3 C_{fg}^3}\right)\eta_t\|h_t^m - g^m(\bar{x}_t)\|^2 \right) - \left(\frac{\rho\gamma\eta_t}{2} - \frac{4C_g^2\eta_t^2\gamma^3}{\rho} - \frac{4L_g^2\eta_t^2\gamma^3}{\rho} - \frac{8L_f^2 C_g^2\eta_t^2\gamma^3}{\rho}\right)\|\bar{d}_t\|^2$$

$$+ \frac{\gamma}{M\rho}\left(BC_g^2 + 5C_f^2 L_g^2 + \frac{c_2^2 C_g^2 L_f^2}{864q^3\gamma^3 L_{fg}^3 C_{fg}^3}\right)\eta_t(q-1)\sum_{l=s_t}^{t-1}\gamma^2\eta_l^2 \sum_{m=1}^{M} \mathbb{E}\|d_l^m - \bar{d}_l\|^2$$

$$+ \frac{\gamma}{M\rho}\left(4C_g^2\eta_t^2\gamma^2 + 4L_g^2\eta_t^2\gamma^2 + 8L_f^2 C_g^2\eta_t^2\gamma^2\right)\sum_{m=1}^{M}\mathbb{E}\|d_t^m - \bar{d}_t\|^2 + \frac{2\sigma^2\gamma}{\rho}\left(\alpha_{t+1}^2 + \beta_{t+1}^2 + \varrho_{t+1}^2\right),$$

$$(78)$$

where the first inequality holds by the above inequalities (73), (74), (75) and (76), and the last inequality is due to Lemma 9.

Let $s_t = q\lfloor t/q \rfloor + 1$, summing the above inequality (78) over $t = s_t$ to $s_t + q - 1$, we have

$$
\sum_{t=s_t}^{s_t+q-1} \left( \Omega_{t+1} - \Omega_t \right)
$$

$$
\leq \sum_{t=s_t}^{s_t+q-1} \frac{1}{M} \sum_{m=1}^{M} \left( - \frac{C_f^2 \gamma}{2\rho} \eta_t \|u_t^m - \nabla g^m(\bar{x}_t)\|^2 - \frac{C_g^2 \gamma}{\rho} \eta_t \|v_t^m - \nabla f^m(h_t^m)\|^2 \right.
$$

$$
\left. - \frac{\gamma}{\rho} \left( \frac{B}{2} - 4C_g^2 L_f^2 - \frac{c_2^2 C_g^2 L_f^2}{864 q^3 \gamma^3 L_{fg}^3 C_{fg}^3} \right) \eta_t \|h_t^m - g^m(\bar{x}_t)\|^2 \right)
$$

$$
- \sum_{t=s_t}^{s_t+q-1} \left( \frac{\rho \gamma \eta_t}{2} - \frac{4C_g^2 \eta_t^2 \gamma^3}{\rho} - \frac{4L_g^2 \eta_t^2 \gamma^3}{\rho} - \frac{8L_f^2 C_g^2 \eta_t^2 \gamma^3}{\rho} \right) \|\bar{d}_t\|^2
$$

$$
+ \frac{\gamma}{M\rho} \left( BC_g^2 + 5C_f^2 L_g^2 + \frac{c_2^2 C_g^2 L_f^2}{864 q^3 \gamma^3 L_{fg}^3 C_{fg}^3} \right) \sum_{t=s_t}^{s_t+q-1} \eta_t (q-1) \sum_{l=s_t}^{t-1} \gamma^2 \eta_l^2 \sum_{m=1}^{M} \mathbb{E}\|d_l^m - \bar{d}_l\|^2
$$

$$
+ \sum_{t=s_t}^{s_t+q-1} \frac{\gamma}{M\rho} \left( 4C_g^2 \eta_t^2 \gamma^2 + 4L_g^2 \eta_t^2 \gamma^2 + 8L_f^2 C_g^2 \eta_t^2 \gamma^2 \right) \sum_{m=1}^{M} \mathbb{E}\|d_t^m - \bar{d}_t\|^2 + \sum_{t=s_t}^{s_t+q-1} \frac{2\sigma^2 \gamma}{\rho} \left( \alpha_{t+1}^2 + \beta_{t+1}^2 + \varrho_{t+1}^2 \right)
$$

$$
\leq \sum_{t=s_t}^{s_t+q-1} \frac{1}{M} \sum_{m=1}^{M} \left( - \frac{C_f^2 \gamma}{2\rho} \eta_t \|u_t^m - \nabla g^m(\bar{x}_t)\|^2 - \frac{C_g^2 \gamma}{\rho} \eta_t \|v_t^m - \nabla f^m(h_t^m)\|^2 \right.
$$

$$
\left. - \frac{\gamma}{\rho} \left( \frac{B}{2} - 4C_g^2 L_f^2 - \frac{c_2^2 C_g^2 L_f^2}{864 q^3 \gamma^3 L_{fg}^3 C_{fg}^3} \right) \eta_t \|h_t^m - g^m(\bar{x}_t)\|^2 \right)
$$

$$
- \sum_{t=s_t}^{s_t+q-1} \left( \frac{\rho \gamma \eta_t}{2} - \frac{4C_g^2 \eta_t^2 \gamma^3}{\rho} - \frac{4L_g^2 \eta_t^2 \gamma^3}{\rho} - \frac{8L_f^2 C_g^2 \eta_t^2 \gamma^3}{\rho} \right) \|\bar{d}_t\|^2
$$

$$
+ \frac{\gamma}{M\rho} \left( BC_g^2 + 5C_f^2 L_g^2 + \frac{c_2^2 C_g^2 L_f^2}{864 q^3 \gamma^3 L_{fg}^3 C_{fg}^3} \right) \frac{\rho^2}{(24)^2 L_{fg}^2 C_{fg}^2} \sum_{t=s_t}^{s_t+q-1} \eta_t \sum_{m=1}^{M} \mathbb{E}\|d_t^m - \bar{d}_t\|^2
$$

$$
+ \frac{\gamma}{M\rho} \frac{\gamma \rho}{6q L_{fg} C_{fg}} \left( C_g^2 + L_g^2 + 2L_f^2 C_g^2 \right) \sum_{t=s_t}^{s_t+q-1} \eta_t \sum_{m=1}^{M} \mathbb{E}\|d_t^m - \bar{d}_t\|^2
$$

$$
+ \frac{\sigma^2}{12q L_{fg} C_{fg}} \left( c_1^2 + c_2^2 + c_3^2 \right) \sum_{t=s_t}^{s_t+q-1} \eta_t^3, \tag{79}
$$

where the second inequality is due to $\eta_t \leq \frac{\rho}{24q\gamma L_{fg} C_{fg}}$ for all $t \geq 0$.

Let $\gamma^2 \geq \frac{\rho^2 \sqrt{c_1^2 + c_3^2}}{24\sqrt{30}q L_{fg}^2 C_{fg}^2}$, we have

$$
\frac{\rho^2 C_g^2}{(24)^2 L_{fg}^2 C_{fg}^2} \frac{\rho^2 (c_1^2 + c_3^2)}{120 q^2 \gamma^4 C_{fg}^2 L_{fg}^2 C_g^2} \leq \frac{1}{4}. \tag{80}
$$

Set $\Theta = \left(5C_f^2 L_g^2 + \frac{c_2^2 C_g^2 L_f^2}{864 q^3 \gamma^3 L_{fg}^3 C_{fg}^3}\right) \frac{\rho^2}{(24)^2 L_{fg}^2 C_{fg}^2} + \frac{\gamma\rho}{6qL_{fg}C_{fg}}\left(C_g^2 + L_g^2 + 2L_f^2 C_g^2\right)$. Based on the above Lemma 10, then we have

$$\sum_{t=s_t}^{s_t+q-1} \left(\Omega_{t+1} - \Omega_t\right)$$

$$\leq \sum_{t=s_t}^{s_t+q-1} \frac{1}{M} \sum_{m=1}^{M} \left( -\frac{C_f^2\gamma}{2\rho}\eta_t \mathbb{E}\|u_t^m - \nabla g^m(\bar{x}_t)\|^2 - \frac{C_g^2\gamma}{\rho}\eta_t \mathbb{E}\|v_t^m - \nabla f^m(h_t^m)\|^2 \right.$$

$$\left. - \frac{\gamma}{\rho}\left(\frac{B}{2} - 4C_g^2 L_f^2 - \frac{c_2^2 C_g^2 L_f^2}{864 q^3 \gamma^3 L_{fg}^3 C_{fg}^3}\right)\eta_t \mathbb{E}\|h_t^m - g^m(\bar{x}_t)\|^2 \right)$$

$$- \sum_{t=s_t}^{s_t+q-1} \left(\frac{\rho\gamma\eta_t}{2} - \frac{4C_g^2\eta_t^2\gamma^3}{\rho} - \frac{4L_g^2\eta_t^2\gamma^3}{\rho} - \frac{8L_f^2 C_g^2\eta_t^2\gamma^3}{\rho}\right)\mathbb{E}\|\bar{d}_t\|^2$$

$$+ \frac{\gamma}{M\rho}\left(BC_g^2 + 5C_f^2 L_g^2 + \frac{c_2^2 C_g^2 L_f^2}{864 q^3 \gamma^3 L_{fg}^3 C_{fg}^3}\right) \frac{\rho^2}{(24)^2 L_{fg}^2 C_{fg}^2}$$

$$\cdot \left(\frac{6M}{5}\sum_{t=s_t}^{s_t+q-1}\eta_t \mathbb{E}\|\bar{d}_t\|^2 + \frac{\rho^2(c_1^2+c_3^2)}{120 q^2 \gamma^4 C_{fg}^2 L_{fg}^2 C_g^2}\sum_{t=s_t}^{s_t+q-1}\eta_t \sum_{m=1}^{M}\mathbb{E}\|h_t^m - g^m(\bar{x}_t)\|^2 + \frac{3M\hat{\delta}^2}{5\gamma^2 L_{fg}^2}\sum_{t=s_t}^{s_t+q-1}\eta_t^3\right)$$

$$+ \frac{\gamma}{M\rho}\frac{\gamma\rho}{6qL_{fg}C_{fg}}\left(C_g^2 + L_g^2 + 2L_f^2 C_g^2\right)$$

$$\cdot \left(\frac{6M}{5}\sum_{t=s_t}^{s_t+q-1}\eta_t \mathbb{E}\|\bar{d}_t\|^2 + \frac{\rho^2(c_1^2+c_3^2)}{120 q^2 \gamma^4 C_{fg}^2 L_{fg}^2 C_g^2}\sum_{t=s_t}^{s_t+q-1}\eta_t \sum_{m=1}^{M}\mathbb{E}\|h_t^m - g^m(\bar{x}_t)\|^2 + \frac{3M\hat{\delta}^2}{5\gamma^2 L_{fg}^2}\sum_{t=s_t}^{s_t+q-1}\eta_t^3\right)$$

$$+ \frac{\sigma^2}{12qL_{fg}C_{fg}}\left(c_1^2 + c_2^2 + c_3^2\right)\sum_{t=s_t}^{s_t+q-1}\eta_t^3$$

$$\leq \sum_{t=s_t}^{s_t+q-1} \frac{1}{M} \sum_{m=1}^{M} \left( -\frac{C_f^2\gamma}{2\rho}\eta_t \mathbb{E}\|u_t^m - \nabla g^m(\bar{x}_t)\|^2 - \frac{C_g^2\gamma}{\rho}\eta_t \mathbb{E}\|v_t^m - \nabla f^m(h_t^m)\|^2 \right.$$

$$\left. - \frac{\gamma}{\rho}\left(\frac{B}{4} - 4C_g^2 L_f^2 - \frac{c_2^2 C_g^2 L_f^2}{864 q^3 \gamma^3 L_{fg}^3 C_{fg}^3} - \frac{\Theta\rho^2(c_1^2+c_3^2)}{120 q^2 \gamma^4 C_{fg}^2 L_{fg}^2 C_g^2}\right)\eta_t \mathbb{E}\|h_t^m - g^m(\bar{x}_t)\|^2 \right)$$

$$- \sum_{t=s_t}^{s_t+q-1} \left(\frac{\rho\gamma\eta_t}{2} - \frac{4C_g^2\eta_t^2\gamma^3}{\rho} - \frac{4L_g^2\eta_t^2\gamma^3}{\rho} - \frac{8L_f^2 C_g^2\eta_t^2\gamma^3}{\rho} - \frac{6\gamma}{5\rho}\left(\Theta + \frac{BC_g^2\rho^2}{(24)^2 L_{fg}^2 C_{fg}^2}\right)\eta_t\right)\mathbb{E}\|\bar{d}_t\|^2$$

$$+ \left(\Theta + \frac{BC_g^2\rho^2}{(24)^2 L_{fg}^2 C_{fg}^2}\right)\frac{3\hat{\delta}^2}{5\rho\gamma L_{fg}^2}\sum_{t=s_t}^{s_t+q-1}\eta_t^3 + \frac{\sigma^2}{12qL_{fg}C_{fg}}\left(c_1^2 + c_2^2 + c_3^2\right)\sum_{t=s_t}^{s_t+q-1}\eta_t^3$$

$$\leq \sum_{t=s_t}^{s_t+q-1} \frac{1}{M} \sum_{m=1}^{M} \left( -\frac{C_f^2\gamma}{2\rho}\eta_t \mathbb{E}\|u_t^m - \nabla g^m(\bar{x}_t)\|^2 - \frac{C_g^2\gamma}{\rho}\eta_t \mathbb{E}\|v_t^m - \nabla f^m(h_t^m)\|^2 - \frac{\gamma C_g^2 L_f^2}{\rho}\eta_t \mathbb{E}\|h_t^m - g^m(\bar{x}_t)\|^2 \right)$$

$$- \sum_{t=s_t}^{s_t+q-1} \frac{\rho\gamma\eta_t}{4}\mathbb{E}\|\bar{d}_t\|^2 + \frac{\rho\hat{\delta}^2}{16\gamma L_{fg}^2}\sum_{t=s_t}^{s_t+q-1}\eta_t^3 + \frac{\sigma^2}{12qL_{fg}C_{fg}}\left(c_1^2 + c_2^2 + c_3^2\right)\sum_{t=s_t}^{s_t+q-1}\eta_t^3, \qquad (81)$$

where the second inequality holds by the above inequality (80), and the last inequality holds by $B \geq 20C_g^2 L_f^2 + \frac{c_2^2 C_g^2 L_f^2}{216 q^3 \gamma^3 L_{fg}^3 C_{fg}^3} + \frac{\Theta\rho^2(c_1^2+c_3^2)}{30 q^2 \gamma^4 C_{fg}^2 L_{fg}^2 C_g^2}$, $\gamma \leq \frac{3\rho q L_{fg}C_{fg}}{4(C_g^2 + L_g^2 + 2L_f^2 C_g^2)}$ (i.e., the following inequality (82) and $\Theta + \frac{BC_g^2\rho^2}{(24)^2 L_{fg}^2 C_{fg}^2} \leq \frac{5\rho^2}{48}$.

Since $\eta_t \leq \frac{\rho}{24q\gamma L_{fg}C_{fg}}$ and $\gamma \leq \frac{3\rho q L_{fg}C_{fg}}{4(C_g^2 + L_g^2 + 2L_f^2 C_g^2)}$, we have

$$\gamma^2 \leq \frac{\rho^2}{32(C_g^2 + L_g^2 + 2L_f^2 C_g^2)} \frac{24q\gamma L_{fg}C_{fg}}{\rho} \leq \frac{\rho^2}{32\eta_t(C_g^2 + L_g^2 + 2L_f^2 C_g^2)}. \tag{82}$$

Summing the above inequality (81) from $t = 1$ to $T$, then we have

$$\sum_{t=1}^{T} (\Omega_{t+1} - \Omega_t)$$

$$\leq \sum_{t=1}^{T} \frac{1}{M} \sum_{m=1}^{M} \left( -\frac{C_f^2 \gamma}{2\rho} \eta_t \mathbb{E}\|u_t^m - \nabla g^m(\bar{x}_t)\|^2 - \frac{C_g^2 \gamma}{\rho} \eta_t \mathbb{E}\|v_t^m - \nabla f^m(h_t^m)\|^2 - \frac{\gamma C_g^2 L_f^2}{\rho} \eta_t \mathbb{E}\|h_t^m - g^m(\bar{x}_t)\|^2 \right)$$

$$- \sum_{t=1}^{T} \frac{\rho\gamma\eta_t}{4} \mathbb{E}\|\bar{d}_t\|^2 + \frac{\rho\hat{\delta}^2}{16\gamma L_{fg}^2} \sum_{t=1}^{T} \eta_t^3 + \frac{\sigma^2}{12q L_{fg}C_{fg}}(c_1^2 + c_2^2 + c_3^2) \sum_{t=1}^{T} \eta_t^3. \tag{83}$$

Since $h_1^m = \frac{1}{q} \sum_{j=1}^{q} g^m(x_1^m; \zeta_{1,j}^m)$, $u_1^m = \frac{1}{q} \sum_{j=1}^{q} \nabla g^m(x_1^m; \zeta_{1,j}^m)$ and $v_1^m = \frac{1}{q} \sum_{j=1}^{q} \nabla f(h_1^m; \xi_{1,j}^m)$ for all $m \in [M]$, we have

$$\Omega_1 = \mathbb{E}\left[ F(\bar{x}_1) + \frac{\gamma}{\rho\eta_0} \frac{1}{M} \sum_{m=1}^{M} \left( \|h_1^m - g^m(x_1^m)\|^2 + \|u_1^m - \nabla g^m(x_1^m)\|^2 + \|v_1^m - \nabla f^m(h_1^m)\|^2 \right) \right]$$

$$\leq F(\bar{x}_1) + \frac{3\gamma\sigma^2}{q\rho\eta_0}, \tag{84}$$

where the last inequality holds by Assumption 2.

Since $\eta_t = \frac{k}{(n+t)^{1/3}}$ is decreasing, i.e., $\eta_T^{-1} \geq \eta_t^{-1}$ for any $0 \leq t \leq T$, we have

$$\frac{1}{T} \sum_{t=1}^{T} \mathbb{E}\left[ \frac{1}{M} \sum_{m=1}^{M} \frac{1}{\rho^2} \left( 2C_f^2 \|u_t^m - \nabla g^m(\bar{x}_t)\|^2 + 4C_g^2 \|v_t^m - \nabla f^m(h_t^m)\|^2 + 4C_g^2 L_f^2 \|h_t^m - g^m(\bar{x}_t)\|^2 \right) + \|\bar{d}_t\|^2 \right]$$

$$\leq \frac{4}{T\rho\gamma\eta_T} \sum_{t=1}^{T} (\Omega_t - \Omega_{t+1}) + \frac{\hat{\delta}^2}{4T\gamma^2 \eta_T L_{fg}^2} \sum_{t=1}^{T} \eta_t^3 + \frac{(c_1^2 + c_2^2 + c_3^2)\sigma^2}{3T\rho\gamma\eta_T q L_{fg}C_{fg}} \sum_{t=1}^{T} \eta_t^3$$

$$\leq \frac{4}{T\rho\gamma\eta_T} \left( F(\bar{x}_1) + \frac{3\gamma\sigma^2}{\rho\eta_0} - F^* \right) + \left( \frac{\hat{\delta}^2}{4T\gamma^2 \eta_T L_{fg}^2} + \frac{(c_1^2 + c_2^2 + c_3^2)\sigma^2}{3T\rho\gamma\eta_T q L_{fg}C_{fg}} \right) \sum_{t=1}^{T} \eta_t^3$$

$$\leq \frac{4}{T\rho\gamma\eta_T} \left( F(\bar{x}_1) + \frac{3\gamma\sigma^2}{q\rho\eta_0} - F^* \right) + \left( \frac{\hat{\delta}^2}{4T\gamma^2 \eta_T L_{fg}^2} + \frac{(c_1^2 + c_2^2 + c_3^2)\sigma^2}{3T\rho\gamma\eta_T q L_{fg}C_{fg}} \right) \int_1^T \frac{k^3}{n+t} dt$$

$$\leq \frac{4}{T\rho\gamma\eta_T} \left( F(\bar{x}_1) + \frac{3\gamma\sigma^2}{q\rho\eta_0} - F^* \right) + \frac{1}{T\eta_T} \left( \frac{\hat{\delta}^2}{4\gamma^2 L_{fg}^2} + \frac{(c_1^2 + c_2^2 + c_3^2)\sigma^2}{3\rho\gamma q L_{fg}C_{fg}} \right) \ln(n+T)$$

$$= \left( \frac{4(F(\bar{x}_1) - F^*)}{k\rho\gamma} + \frac{12n^{1/3}\sigma^2}{qk^2\rho^2} + 4k^2 \left( \frac{\hat{\delta}^2}{4\gamma^2 L_{fg}^2} + \frac{(c_1^2 + c_2^2 + c_3^2)\sigma^2}{3\rho\gamma q L_{fg}C_{fg}} \right) \ln(n+T) \right) \frac{(n+T)^{1/3}}{T}, \tag{85}$$

where the second inequality holds by the above inequality (84). Let $G = \frac{4(F(\bar{x}_1) - F^*)}{k\rho\gamma} + \frac{12n^{1/3}\sigma^2}{qk^2\rho^2} + 4k^2 \left( \frac{\hat{\delta}^2}{4\rho\gamma^2 L_{fg}^2} + \frac{(c_1^2 + c_2^2 + c_3^2)\sigma^2}{3\rho\gamma q L_{fg}C_{fg}} \right) \ln(n+T)$. According to the above Lemma 5, then we have

$$\frac{1}{T} \sum_{t=1}^{T} \mathbb{E}\left[ \frac{1}{\rho^2} \|\bar{w}_t - \nabla F(\bar{x}_t)\|^2 + \|\bar{d}_t\|^2 \right]$$

$$\leq \frac{1}{T} \sum_{t=1}^{T} \mathbb{E}\left[ \frac{1}{M} \sum_{m=1}^{M} \frac{1}{\rho^2} \left( 2C_f^2 \|u_t^m - \nabla g^m(\bar{x}_t)\|^2 + 4C_g^2 \|v_t^m - \nabla f^m(h_t^m)\|^2 + 4C_g^2 L_f^2 \|h_t^m - g^m(\bar{x}_t)\|^2 \right) + \|\bar{d}_t\|^2 \right]$$

$$\leq \frac{G}{T}(n+T)^{1/3}, \tag{86}$$

where the first inequality holds by Lemma 5, and the last inequality holds by (85).

Since $\bar{d}_t = \frac{\bar{x}_t - \bar{x}_{t+1}}{\eta_t \gamma} = \frac{\eta_t \gamma A_t^{-1} \bar{w}_t}{\eta_t \gamma} = A_t^{-1} \bar{w}_t$, and let $\mathcal{G}_t = \frac{1}{\rho} \|\bar{w}_t - \nabla F(\bar{x}_t)\| + \|\bar{d}_t\|$, we have

$$
\begin{aligned}
\mathcal{G}_t &= \frac{1}{\rho} \|\bar{w}_t - \nabla F(\bar{x}_t)\| + \|\bar{d}_t\| = \frac{1}{\rho} \|\bar{w}_t - \nabla F(\bar{x}_t)\| + \|A_t^{-1} \bar{w}_t\| \\
&\geq \|A_t^{-1} \bar{w}_t\| + \frac{1}{\rho} \|\bar{w}_t - \nabla F(\bar{x}_t)\| \\
&= \frac{1}{\|A_t\|} \|A_t\| \|A_t^{-1} \bar{w}_t\| + \frac{1}{\rho} \|\bar{w}_t - \nabla F(\bar{x}_t)\| \\
&\geq \frac{1}{\|A_t\|} \|\bar{w}_t\| + \frac{1}{\rho} \|\bar{w}_t - \nabla F(\bar{x}_t)\| \\
&\overset{(i)}{\geq} \frac{1}{\|A_t\|} \|\bar{w}_t\| + \frac{1}{\|A_t\|} \|\nabla F(\bar{x}_t) - \bar{w}_t\| \\
&\geq \frac{1}{\|A_t\|} \|\nabla F(\bar{x}_t)\|,
\end{aligned}
\tag{87}
$$

where the inequality $(i)$ holds by $\|A_t\| \geq \rho$ for all $t \geq 1$ due to Assumption 6. Then we have

$$
\|\nabla F(\bar{x}_t)\| \leq \|A_t\| \mathcal{G}_t.
\tag{88}
$$

According to Cauchy-Schwarz inequality, we have

$$
\frac{1}{T} \sum_{t=1}^{T} \mathbb{E}\|\nabla F(\bar{x}_t)\| \leq \frac{1}{T} \sum_{t=1}^{T} \mathbb{E}[\mathcal{G}_t \|A_t\|] \leq \sqrt{\frac{1}{T} \sum_{t=1}^{T} \mathbb{E}[\mathcal{G}_t^2]} \sqrt{\frac{1}{T} \sum_{t=1}^{T} \mathbb{E}\|A_t\|^2}.
\tag{89}
$$

According to the above inequality (86), we have

$$
\begin{aligned}
\frac{1}{T} \sum_{t=1}^{T} \mathbb{E}[\mathcal{G}_t^2] &\leq \frac{1}{T} \sum_{t=1}^{T} \mathbb{E}\Big[\frac{2}{\rho^2} \|\bar{w}_t - \nabla F(\bar{x}_t)\|^2 + 2\|\bar{d}_t\|^2\Big] \\
&\leq \frac{2G}{T} (n+T)^{1/3}.
\end{aligned}
\tag{90}
$$

Combining the above inequalities (89) with (90), we have

$$
\begin{aligned}
\frac{1}{T} \sum_{t=1}^{T} \mathbb{E}\|\nabla F(\bar{x}_t)\| &\leq \sqrt{\frac{1}{T} \sum_{t=1}^{T} \mathbb{E}[\mathcal{G}_t^2]} \sqrt{\frac{1}{T} \sum_{t=1}^{T} \mathbb{E}\|A_t\|^2} \\
&\leq \Big(\frac{\sqrt{2G} n^{1/6}}{T^{1/2}} + \frac{\sqrt{2G}}{T^{1/3}}\Big) \sqrt{\frac{1}{T} \sum_{t=1}^{T} \mathbb{E}\|A_t\|^2}.
\end{aligned}
\tag{91}
$$

$\square$

# B    ADDITIONAL EXPERIMENTS

## B.1    TASK-DISTRIBUTED META LEARNING

In this subsection, we evaluate the effectiveness of our proposed algorithms for personalized federated learning, which can be described a task-distributed meta learning Huang et al. (2021a). From the above (2), the task-distributed meta learning problem can be rewritten as a distributed composition optimization problem, defined as

$$
\min_{x \in \mathbb{R}^d} \frac{1}{M} \sum_{m=1}^{M} \exp\Big(f^m\big(x - \eta \nabla f^m(x)\big)/\lambda\Big),
\tag{92}
$$

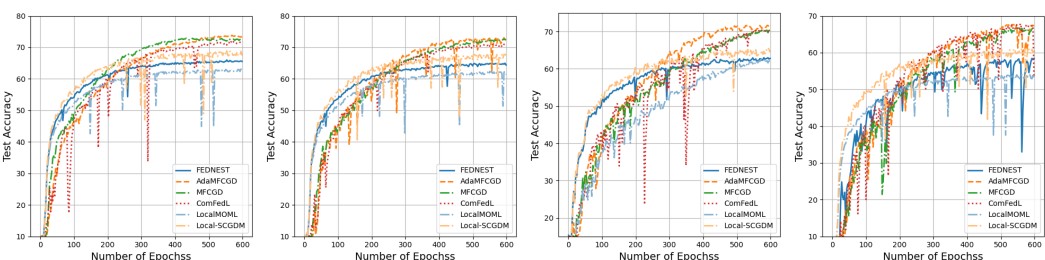

Figure 3: We evaluated the test accuracy (%) of different FCO methods for solving the distributed problem (92) under various settings using a heterogeneous CIFAR-10 dataset. We varied the hyper-parameter $\chi$ to control the percentage of samples from the dominant class, with values of 0.3, 0.5, 0.7, and 0.9 from left to right.

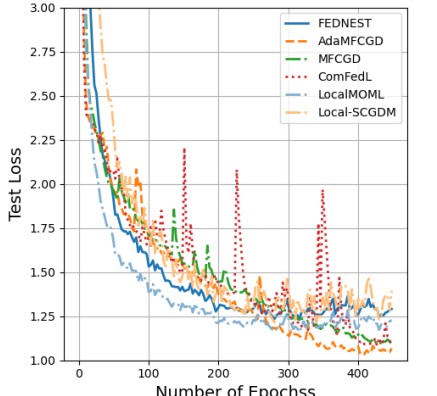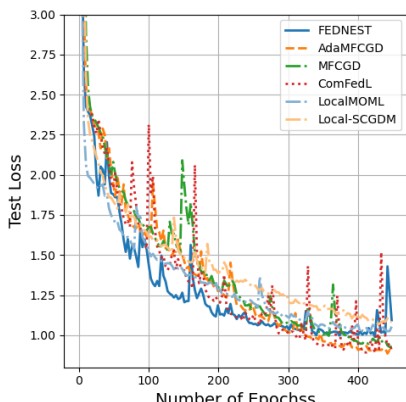

Figure 4: We computed the cross entropy on the test sets of various FCO methods for solving the distributed problem (92) using a heterogeneous CIFAR-10 dataset under different settings. The results were obtained for $\chi = 0.7$ on the left and $\chi = 0.9$ on the right.

where $\lambda > 0$ is a regularization parameter. The Problem (92) is also a special case of the above Problem (1).

In the experiments, we consider a multi-class classification task over the CIFAR10 Krizhevsky et al. (2009) dataset, using a 7-layer CNN. We create a heterogeneous training (validation) dataset consisting of 10 clients and 1 server, where each client has images from a dominant class and a small percentage of images from other classes. Specifically, the $m$-th client owns $\chi$ percentage of images from the $m$-th class and $(1 - \chi)/9$ for the other classes. For $\chi > 0.1$, the images of each client are dominated by a different class, which is referred to as the dominant class. In our experiments, each client has a different dominant class, for example, one client has 60% samples from the airplane class and the remaining 40% samples from other classes. The hyper-parameter $\chi$ controls the percentage of samples from the dominant class over each client. The dominant class is different for different clients.

In the experiments, we utilize a grid search approach to determine the optimal hyper-parameters for all methods, and the search space is described in subsection B.2. As each client's data distribution is constructed to be heterogeneous, tuning a personalized model for each client can offer additional benefits. For selecting the learning rate, we typically set the learning rate to 0.05 at inner loops and the learning rate to 0.1 at outer loops. Additionally, we randomly select five clients to participate in training per epoch, and we set the asynchronization step $q$ to 5 if it is not specified. The total number of training iteration steps is set to 600.

From Figures 3 and 4, it is apparent that the convergence difficulty increases as the hyperparameter $\chi$ increases. Among all of the heterogeneous ratios, our MFGCD and AdaMFGCD algorithms outperform other baselines, particularly when the data distribution is significantly heterogeneous. The other composition federated optimization methods such as FEDNEST Tarzanagh et al. (2022), Local-MOML Wang et al. (2021), and Local-SCGDM Gao et al. (2022) exhibit inferior performance in both test accuracy and loss under these circumstances. Meanwhile, although ComFedL Huang et al. (2021a) obtains a better performance, the loss (accuracy) curve is quite noisy. Our MFCGD and AdaMFCGD algorithms adaptively adjust the weight of clients based on their performance on the task (training loss). In other words, if a client's data distribution is challenging to learn (i.e., higher training loss), the algorithm increases its learning rate, while for clients with simpler distributions, the learning rate is reduced. Overall, the results indicate that our MFCGD and AdaMFCGD approaches can also enhance personalized FL.

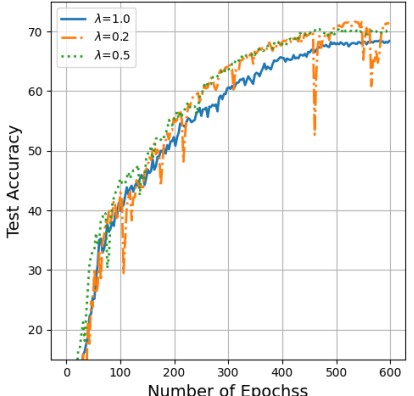 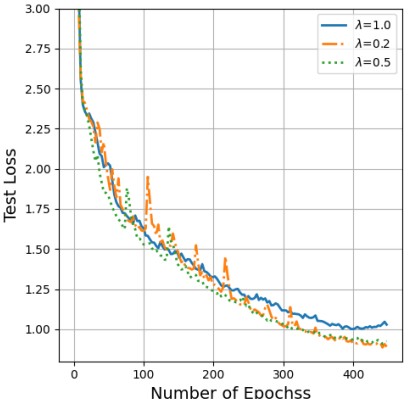

Figure 5: Comparing the accuracy(%) (left) and cross-entropy loss (right) on different regularization parameter $\lambda$ on our AdaMFCGD algorithm

Figure 5 demonstrates the robustness of our AdaMFCGD algorithm by varying the regularization parameter $\lambda$. The results show that our AdaMFCGD algorithm achieves good test accuracy and low test loss for different values of $\lambda$. Additionally, decreasing $\lambda$ leads to faster convergence of the algorithm, particularly when running multiple local epochs. Figure 6 illustrates the robustness of our algorithm when varying the asynchronization step $q$. It is noteworthy that our AdaMFCGD achieves its optimal performance when $q = 1$, as it enables the momentum-based variance reduction technique to fully demonstrate its potential by calculating an adaptive matrix in each iteration. In comparison to varying $q$ in Robust FL, $q$ shows a less significant influence in Task-Distributed Meta Learning due to the heterogeneity of the data, which can result in significant changes in the gradient across iterations, further impacting the stability of the convergence curve. In Task-Distributed Meta Learning, asynchronization step $q$ also relatively controls the degree of heterogeneity.

| Layer Type | Output Size | Kernel Size | Stride | Activation |
|---|---|---|---|---|
| Input | 28 x 28 x 1 | - | - | - |
| Convolution | 24 x 24 x 6 | 5 x 5 | 1 | ReLU |
| Max Pooling | 12 x 12 x 6 | 2 x 2 | 2 | - |
| Convolution | 8 x 8 x 16 | 5 x 5 | 1 | ReLU |
| Max Pooling | 4 x 4 x 16 | 2 x 2 | 2 | - |
| Convolution | 120 | 5 x 5 | 1 | ReLU |
| Dense | 360 | - | - | ReLU |
| Output | 10 | - | - | Softmax |

Table 2: Structure of a 4-layer CNN for MNIST

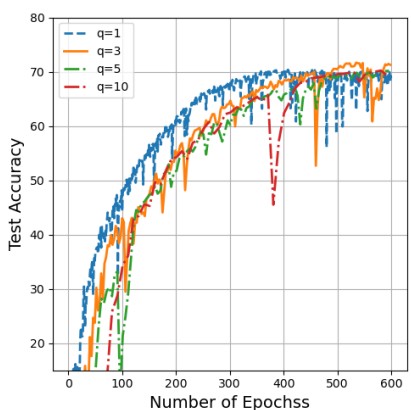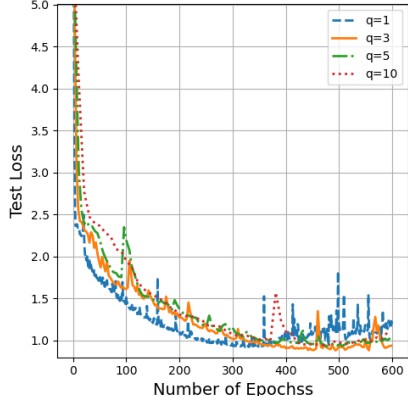

Figure 6: Comparing the accuracy(%) (left) and cross-entropy loss (right) on different synchronization step $q$ on our AdaMFCGD algorithm.

| Layer Type | Output Size | Kernel Size | Stride | Activation |
|---|---|---|---|---|
| Input | 32 x 32 x 3 | - | - | - |
| Convolution | 30 x 30 x 96 | 3 x 3 | 1 | ReLU |
| Convolution | 14 x 14 x 96 | 3 x 3 | 2 | ReLU |
| Convolution | 14 x 14 x 196 | 1 x 1 | 1 | ReLU |
| Convolution | 14 x 14 x 10 | 1 x 1 | 1 | ReLU |
| Flatten | 1,960 | - | - | - |
| Dense | 1000 | - | - | ReLU |
| Dense | 1000 | - | - | ReLU |
| Output | 10 | - | - | Softmax |

Table 3: Structure of a 7-layer CNN for CIFAR-10

## B.2 IMPLEMENTATION DETAILS

In this subsection, we provide the specific backbone networks of the above two tasks, which are described in Table 2 and Table 3, respectively.

In the above **Robust Federated Learning** experiments, we conduct a search for the learning rate within the range [0.001, 0.01, 0.05, 0.1, 0.2, 0.5, 1]. We observe that for most methods, a learning rate greater than 0.5 caused divergence. In the case of ComFedL Huang et al. (2021a), we conduct a search for the regularization parameter within the range [0.1, 0.5, 1.5, 2] and 0.5 is the best. For FEDNEST Tarzanagh et al. (2022), we directly use the hyperparameters as it reports. We also conduct a search for the hyperparameter of Local-SCGDM Gao et al. (2022) within the range [0.1, 0.5, 1, 1.5, 2] and 1 is the best.

In the above **Task-Distributed Meta Learning** experiments: for the learning rate (both inner and outer if two types of learning rates are needed), we search from [0.001, 0.01, 0.05, 0.1, 0.2, 0.5, 1]. For our method, we search the regularization parameter from [0.1, 0.5, 1, 5]; For FEDNEST Tarzanagh et al. (2022), we directly use the hyperparameters as it reports. We also conduct a search for the hyperparameter of Local-SCGDM Gao et al. (2022) within the range [0.1, 0.5, 1, 1.5, 2] and 0.5 is the best. For Local-MOML Wang et al. (2021), we search its weight parameter $\beta$ within the range [0.1, 0.3, 0.5, 0.7, 0.9] with the fixed inner and outer learning rates for a fair comparison and $\beta = 0.7$ is the best.

