# OpenReview forum: "Faster Adaptive Momentum-Based Federated Methods for Distributed Composition Optimization"
_ICLR.cc/2025/Conference — ICLR 2025 Conference Withdrawn Submission_

### Official Review · Reviewer_EseG · 2024-10-26

**Soundness:** 4
**Presentation:** 4
**Contribution:** 3
**Rating:** 6
**Confidence:** 3

**Summary:**

In this paper, the authors considered the distributed composition optimization problem. They proposed MFCGD and AdaMFCGD, which leverage adaptive step size on both the server and clients, and momentum-based variance reduction on clients. The authors derived convergence results for the proposed algorithms. Finally, they conducted experiments to show that the proposed algorithms achieve better performance than existing methods.

**Strengths:**

- The proposed algorithms are novel and well-motivated.
- The theoretical results are thorough. The authors provided convergence results for a general adaptive step size framework.
- The experimental results are promising. The proposed algorithms achieve better performance than existing methods.

**Weaknesses:**

- The literature review is not up-to-date. It seems the authors only considered works up to 2022. As far as I know, there are more recent works on Distributed/ Federated Composition Optimization (e.g., [1], [2], [3]). The authors should conduct a more thorough literature review.
  [1] Hongchang Gao. A Doubly Recursive Stochastic Compositional Gradient Descent Method for Federated Multi-Level Compositional Optimization. ICML 2024.
  [2] Xidong Wu et al. Federated Conditional Stochastic Optimization. NeurIPS 2023.
  [3] Xinwen Zhang et al. Federated Compositional Deep AUC Maximization. NeurIPS 2023.

**Questions:**

- What are some of the recent works in this area (see also Weaknesses)?
- Typos:
  - Line 897: "Assumption ,."

---

### Official Review · Reviewer_4oDV · 2024-10-26

**Soundness:** 2
**Presentation:** 3
**Contribution:** 2
**Rating:** 3
**Confidence:** 4

**Summary:**

This paper proposes a class of faster adaptive federated compositional optimization algorithms to address non-convex distributed compositional problems. By employing a unified adaptive matrix, the adaptive algorithm achieves a lower sample and lower communication complexities.

**Strengths:**

1. The proposed adaptive algorithms effectively reduce sample and communication complexities for non-convex distributed compositional optimization problems.

**Weaknesses:**

1. Some highly relevant works are not cited. For instance, [1] and [2] discuss storm-like algorithms for compositional optimization problems, while [3] addresses momentum-based federated methods for distributed compositional optimization.
2. The improvement presented in the paper appears to be incremental. The use of an adaptive matrix is not novel within adaptive algorithms, as similar approaches have been explored, such as in [4]. In the distributed setting, the adaptive matrix operates independently on each client without communication. While communicating this matrix would be computationally expensive and unnecessary, this choice makes the theoretical analysis closely resemble that of the non-federated setting. As a result, the contribution of this paper is trivial, limiting its overall novelty.
3. There is a typo in Eq.(4), a_t should have a squared root.

[1] Huizhuo Yuan, Wenqing Hu. Stochastic Recursive Momentum Method for Non-Convex Compositional Optimization, arXiv preprint arXiv: 2006.01688, 2020.
[2] Liu, Jin, et al. Faster Stochastic Variance Reduction Methods for Compositional MiniMax Optimization. Proceedings of the AAAI Conference on Artificial Intelligence. Vol. 38. No. 12. 2024.
[3] Zhang, Xinwen, et al. Federated compositional deep auc maximization. Advances in Neural Information Processing Systems 36 (2024).
[4] Feihu Huang, et al. Super-adam: faster and universal framework of adaptive gradients. Advances in Neural Information Processing Systems, 34:9074–9085, 2021b.

**Questions:**

1. See the Weaknesses above.

---

### Official Review · Reviewer_VADD · 2024-11-03

**Soundness:** 1
**Presentation:** 2
**Contribution:** 1
**Rating:** 1
**Confidence:** 4

**Summary:**

This paper proposes a variance-reduced local stochastic algorithm based on STORM [1] estimators for improving sample and communication complexities in distributed compositional optimization. The algorithm incorporates adaptivity within the SUPER-ADAM [2] framework. They also performed numerical experiments on the robust federated learning problem to testify the proposed algorithm.

[1] Cutkosky, Ashok, and Francesco Orabona. "Momentum-based variance reduction in non-convex sgd." Advances in neural information processing systems 32 (2019).

[2] Huang, Feihu, Junyi Li, and Heng Huang. "Super-adam: faster and universal framework of adaptive gradients." Advances in Neural Information Processing Systems 34 (2021): 9074-9085.

**Strengths:**

Whether the sample and communication complexities for the distributed composition optimization problem can be further improved is indeed an open problem.  Additionally, adaptivity in stochastic algorithms is an interesting area of research, even for single-level optimization problems.

**Weaknesses:**

- The proof of Lemma 6 seems to be problematic. How can you get (27) from (26)? To be specific, how can you get $<w\_t^m,\frac{1}{\eta\_t \gamma}(\bar{x}\_t - \bar{x}\_{t+1})>$ from $\frac{1}{M}\sum_{m=1}^M<w\_t^m,\frac{1}{\eta\_t \gamma}(x\_t^m - x\_{t+1}^m)>$? Note that $w\_t^m$ also depends on $m$.
- The title "Federated Methods for Distributed Compositional Optimization" is potentially misleading. Federated learning and distributed learning are fundamentally distinct paradigms, each with different settings, assumptions (on data and clients), and requirements. See, e.g., Table 1.1 in [3]. Using these terms interchangeably is quite confusing.
- In lines 18 and 19 of Algorithm 1, the proposed algorithms require projection onto balls with unknown problem-specific constants $C_g$ and $C_f$. This limits the applicability of the proposed algorithm.
- Client sampling is highly desirable in the cross-device federated learning regime since the total number of clients M is quite large. Unfortunately, the proposed algorithms in this paper do not support this feature.
- The function g in (10) is an exponential function. It makes no sense to assume its stochastic gradient is Lipschitz and bounded (as in Assumption 2 and Assumption 3), considering the model w in this paper is unconstrained.
- The numerical experiments cannot fully verify the robustness of the proposed AdaMFCGD algorithm without comparing it to standard local algorithms such as FedAvg and SCAFFOLD. This can justify why the composition optimization problem in (10) is meaningful and whether it can improve the robustness compared to Federated ERM.


[3] Kairouz, Peter, H. Brendan McMahan, Brendan Avent, Aurélien Bellet, Mehdi Bennis, Arjun Nitin Bhagoji, Kallista Bonawitz et al. "Advances and open problems in federated learning." Foundations and trends® in machine learning 14, no. 1–2 (2021): 1-210.

**Questions:**

- What are the technical contributions of this paper? I skimmed through the proof and it looks like a combination of several existing results [2, 4, 5, 6]. Can you highlight the technical difficulties and novelties?
- Can you explain how you handled the unknown constant  $C_g$ and $C_f$ (lines 18 and 19 of Algorithm 1) in your experiments?

[4] Gao, Hongchang, Junyi Li, and Heng Huang. "On the convergence of local stochastic compositional gradient descent with momentum." In International Conference on Machine Learning, pp. 7017-7035. PMLR, 2022.

[5] Chen, Tianyi, Yuejiao Sun, and Wotao Yin. "Solving stochastic compositional optimization is nearly as easy as solving stochastic optimization." IEEE Transactions on Signal Processing 69 (2021): 4937-4948.

[6] Jiang, Wei, Bokun Wang, Yibo Wang, Lijun Zhang, and Tianbao Yang. "Optimal algorithms for stochastic multi-level compositional optimization." In International Conference on Machine Learning, pp. 10195-10216. PMLR, 2022.

---

### Official Review · Reviewer_Ua4z · 2024-11-03

**Soundness:** 2
**Presentation:** 2
**Contribution:** 2
**Rating:** 3
**Confidence:** 2

**Summary:**

The paper introduces MFCGD and AdaMFCGD, two algorithms designed for federated compositional optimization in nonconvex settings with heterogeneous data. By leveraging variance reduction techniques, these algorithms achieve efficient sample and communication complexities of $O(\epsilon^{-3})$ and $O(\epsilon^{-2})$, respectively. The authors support their findings with theoretical analysis and empirical validation using the MNIST and CIFAR-10 datasets.

**Strengths:**

The paper tackles a challenging compositional optimization problem in nonconvex settings with heterogeneous data, achieving sample complexity of $O(\epsilon^{-3})$ and communication complexity of $O(\epsilon^{-2})$, comparable to the best current methods.

**Weaknesses:**

* *Novelty of this Approach*: Upon review, there appears to be an overlap with recent advances, specifically with the Fed-DR-SCGD algorithm published at the ICML proceedings in early May ([Fed-DR-SCGD Paper](https://openreview.net/pdf?id=GentO2E4ID)). This algorithm demonstrates comparable sample complexity of $O\left(\frac{1}{\epsilon^3}\right)$ and communication complexity of $O\left(\frac{1}{\epsilon^2}\right)$. Given this context, comparing Fed-DR-SCGD would strengthen the paper’s contribution and clarify its unique value. Could the authors highlight any distinctive advantages of MFCGD / AdaMFCGD over Fed-DR-SCGD?

* *Lack of Explicit Dependence on the Number of Machines $M$*: The convergence rates in *Theorem 1.* and *Theorem 2.* do not explicitly depend on the number of machines $M$, which is atypical in distributed stochastic optimization. In many distributed settings, the convergence rate often scales with the number of machines, reflecting the collective contribution to reducing variance and accelerating convergence.

* *Unavailable Code*: The absence of available code for the experiments reduces the transparency of the results, making it challenging to validate the findings and restricting the community's ability to build upon this work.

**Questions:**

Regarding sample complexity, shouldn't the total number of samples be $ 2qM + TM$? This would include the $2q$ samples drawn during the initialization phase for each worker, as well as the two samples drawn at each iteration $t \in [T] $ and for each worker $m \in [M]$. By this reasoning, the sample complexity would be $2qM + TM = \tilde{O}(M \epsilon^{-3})$. Could the authors provide clarification on this calculation?

---

### Note · Authors · 2024-11-12

I have read and agree with the venue's withdrawal policy on behalf of myself and my co-authors.